# SynTSBench: Rethinking Temporal Pattern Learning in Deep Learning Models for Time Series

**Qitai Tan**[*] , **Yiyun Chen**[*] , **Mo Li**[*] , **Ruiwen Gu** , **Yilin Su** , and **Xiao-Ping Zhang**[†]

Shenzhen Key Laboratory of Ubiquitous Data Enabling,
Shenzhen International Graduate School, Tsinghua University
{tqt24, yy-chen22, li-m24, grw23, suyl24}@mails.tsinghua.edu.cn
xpzhang@ieee.org

## Abstract

Recent advances in deep learning have driven rapid progress in time series forecasting, yet many state-of-the-art models continue to struggle with robust performance in real-world applications, even when they achieve strong results on standard benchmark datasets. This persistent gap can be attributed to the blackbox nature of deep learning architectures and the inherent limitations of current evaluation frameworks, which frequently lack the capacity to provide clear, quantitative insights into the specific strengths and weaknesses of different models, thereby complicating the selection of appropriate models for particular forecasting scenarios. To address these issues, we propose a synthetic data-driven evaluation paradigm, **SynTSBench**, that systematically assesses fundamental modeling capabilities of time series forecasting models through programmable feature configuration. Our framework isolates confounding factors and establishes an interpretable evaluation system with three core analytical dimensions: (1) temporal feature decomposition and capability mapping, which enables systematic evaluation of model capacities to learn specific pattern types; (2) robustness analysis under data irregularities, which quantifies noise tolerance thresholds and anomaly recovery capabilities; and (3) theoretical optimum benchmarking, which establishes performance boundaries for each pattern type—enabling direct comparison between model predictions and mathematical optima. Our experiments show that current deep learning models do not universally approach optimal baselines across all types of temporal features. The code is available at https://github.com/TanQitai/SynTSBench.

## 1 Introduction

In complex decision-support systems such as financial market trend analysis, energy system dispatch optimization, and climate pattern forecasting, time series prediction remains a cornerstone technology due to its exceptional capacity for modeling temporal dependencies and delivering high-precision trend forecasting. Traditional statistical approaches such as ARIMA [1], ETS [2], VAR [3], and other classical methods have historically dominated the field, offering interpretable frameworks based on explicit statistical assumptions and decomposition principles. Recent advancements in deep learning-driven forecasting models have achieved systematic breakthroughs over these conventional methods in both prediction accuracy and computational efficiency. For example, Aut-

---

[*]Equal contribution.

[†]Corresponding author.

39th Conference on Neural Information Processing Systems (NeurIPS 2025) Track on Datasets and Benchmarks.

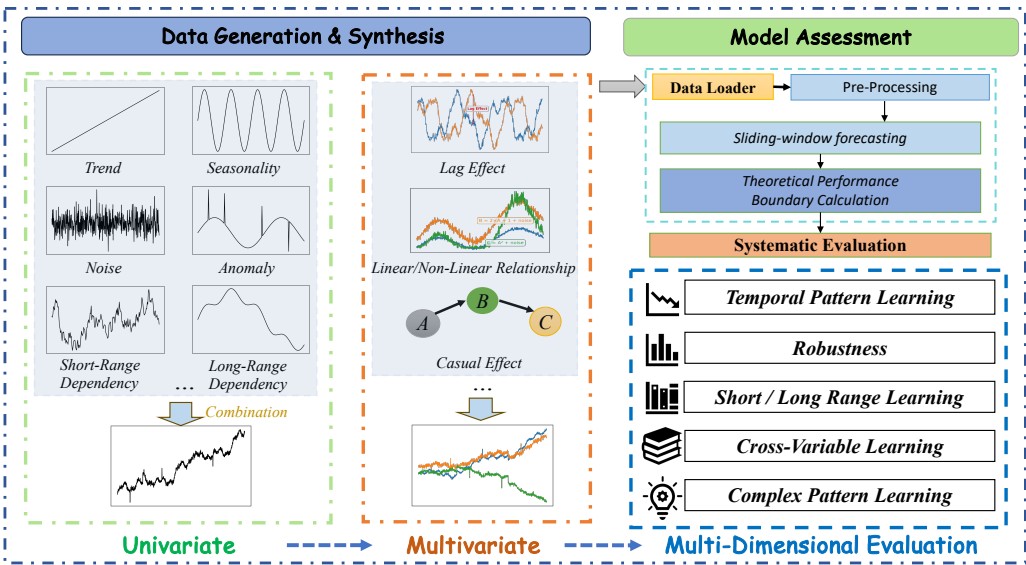

Figure 1: Overview of SynTSBench, a synthetic data-based evaluation framework for time series forecasting models. The framework generates controlled time series data from basic univariate components to complex multivariate patterns with defined relationships. This approach enables systematic assessment of model capabilities across temporal pattern learning, robustness, and dependency modeling, complex pattern recognition, and so on.

oformer [4] enhances long-term temporal modeling through autocorrelation mechanisms and sequence decomposition; DLinear [5] achieves robust predictions in resource-constrained scenarios via lightweight linear architecture; PatchTST [6] reduces computational complexity for long sequences through patch-based processing; SegRNN [7] optimizes long-range dependency modeling via slice encoding and parallel decoding; TSMixer [8] enables joint modeling of temporal and attribute features using multilayer perceptrons; TimeKAN [9] improves nonlinear mapping through learnable activation functions; TimeMixer [10] adapts to complex fluctuation patterns via multi-scale decomposition; TimesNet [11] focuses on deep periodic component modeling; and iTransformer [12] enhances parameter efficiency through improved attention mechanisms. The integration of RevIN dynamic normalization [13] effectively mitigates data distribution shifts in non-stationary scenarios. These innovations collectively advance prediction accuracy and deployment efficiency through sequence decomposition, computational optimization, and adaptive modeling, establishing a robust technical foundation for industrial-grade forecasting systems.

The time series forecasting field has developed several comprehensive evaluation frameworks to standardize model assessment. ProbTS [14] unifies various forecasting paradigms including point and probabilistic forecasting, short and long-range prediction, and autoregressive versus non-autoregressive settings. TFB [15] provides consistent evaluation through diverse domain coverage and standardized pipelines. Other frameworks such as BasicTS [16], BasicTS+ [17], TSlib [18], and similar benchmarks have also contributed to the field. These frameworks have significantly advanced model development, yet they face two critical limitations for industrial applications. First, current frameworks lack effective feature isolation capabilities. Real-world time series contain intertwined components (trends, seasonal patterns, dependencies of varying lengths) that cannot be separated for targeted evaluation. For example, urban traffic data simultaneously exhibits daily commuting patterns (24h), weekend effects (7d), and seasonal variations, with complex interactions between them (such as morning peak hours being affected by seasonal factors like summer heat). Without isolating these components, it remains unclear whether a model's improved performance comes from genuinely capturing specific patterns or from exploiting coincidental correlations in the training data. Second, these frameworks provide no theoretical performance boundaries. Evaluations rely on observational data without establishing what optimal performance should be for a given pattern type. This absence of theoretical baselines creates two challenges: distinguishing between meaningful generalization improvements versus mere noise overfitting, and lacking clear

criteria for determining when a sequence is fundamentally predictable. Real-world data frequently contains outliers (from equipment failures to sensor noise), non-stationarity, and abrupt changes that further complicate evaluation. Without knowing the theoretical optimal solution for a dataset, model optimization resembles a "blind-box" parameter tuning process rather than structured improvement against established boundaries. These limitations reflect a fundamental methodological gap: current approaches use data complexity as a primary validation criterion while neglecting systematic characterization of time series properties and their theoretical solution spaces. A more principled approach would involve controlled generation of temporal patterns with known optimal predictions, enabling precise assessment of model capabilities under varying conditions.

To address these challenges, we propose a synthetic data-driven evaluation paradigm, **SynTSBench**, that systematically assesses fundamental modeling capabilities of time series forecasting models through programmable feature configuration. As illustrated in Figure 1, this framework isolates confounding factors to establish an interpretable evaluation system with three core analytical dimensions:

- **Temporal Feature Decomposition and Capability Mapping:** Constructing synthetic datasets with four key temporal characteristics (trend, seasonality, short/long-range dependencies, multivariate correlation) and complex datasets based on traditional stochastic time series and signal processing models that simulate real-world scenarios. This design enables systematic evaluation of model capacities to learn specific pattern types without interference from confounding factors, quantifying how accurately models capture time series patterns while revealing comparative strengths between different architectural approaches.

- **Robustness Analysis Under Data Irregularities:** Implementing a comprehensive resilience assessment through controlled modification of clean synthetic signals with progressive noise injection and different types of anomalies. This framework facilitates quantification of both noise tolerance thresholds and recovery capabilities.

- **Theoretical Optimum Benchmarking:** Leveraging synthetic data generation to establish theoretical performance boundaries (optimal solutions) for each type of pattern - measures previously unattainable with observational datasets. This innovation enables direct comparison between model predictions and mathematical optima, providing reference points to analyze performance gaps and improvement potential.

## 2   Problem Statement

**Time Series Forecasting**   A time series $X \in \mathbb{R}^{T \times N}$ is a time-oriented sequence of $N$-dimensional time points, where $T$ is the number of time points, and $N$ is the number of variables. When $N = 1$, a time series is called univariate. When $N > 1$, it is called multivariate. Given a historical time series $X \in \mathbb{R}^{H \times N}$ of $H$ time points, time series forecasting aims to predict the next $F$ future time points, i.e., $Y \in \mathbb{R}^{F \times N}$, where $F$ is called the forecasting horizon.

**Time Series Decomposition**   The Wold decomposition theorem [19] states that any stationary time series can be represented as the sum of a deterministic component and a stochastic component. Based on this, time series data, consisting of observed values $y(t)$, can be expressed as: $y(t) = x(t) + n(t)$, where $x(t)$ represents the latent structure of the data (e.g., trends, periodicity), and $n(t)$ accounts for the stochastic noise. The goal of time series forecasting is to model the true signal $x(t)$ using a predictive model $M(\theta, t)$.

## 3   Dataset Construction

**Dataset Design**   Our dataset framework is specifically constructed to serve as a comprehensive evaluation suite for time series forecasting models. Rather than relying solely on real-world data, we design synthetic datasets that systematically cover a wide range of temporal characteristics, enabling targeted assessment of model capabilities. As summarized in Table 1, the dataset includes tests for trend, seasonality, noise robustness, anomaly resilience, dependency modeling, cross-variable relationships, and complex real-world pattern simulation. Each component is carefully parameterized to isolate specific forecasting challenges, ensuring that model performance can be directly attributed to the ability to capture particular temporal features.

Table 1: Dataset Design and Targeted Evaluation Capabilities

| Dataset Part | Construction Details | Targeted Modeling Capability |
|---|---|---|
| **Temporal Pattern Learning** | Implement 11 trend functions and 10 periodic patterns with varying complexities | Assess capability to learn and extrapolate both functional trends and cyclical patterns |
| **Robustness Against Noise & Anomalies** | Inject Gaussian noise at multiple SNR levels; test diverse noise distributions (uniform, Laplace, t-distribution, Lévy stable); introduce point anomalies, pulse anomalies, mean shifts, and trend shifts at varying densities | Evaluate signal recovery, denoising capability, resilience to various noise characteristics, and robustness to irregular disturbances and distribution shifts |
| **Short/Long-range Dependencies** | Generate ARMA processes with short/long-range dependencies; random walks; white noise without dependency | Measure ability to capture varying temporal dependencies from immediate to distant lags |
| **Cross-Variable Learning** | Create lagged relationships with varying time steps; sine-noise composition for linear additive relationships; conditional relationships with threshold-dependent interactions; nonlinear transformations; complex multi-variable systems with feedback mechanisms | Test ability to detect and leverage diverse dependencies between multiple variables, including temporal lags, linear combinations, conditional interactions, and nonlinear relationships |
| **Dataset Scale Sensitivity** | Construct time series with varying lengths to assess model performance at different data scales | Analyze how the amount of available data points affects forecasting accuracy and compare learning efficiency of different models |
| **Complex Real-World Pattern Simulation** | Develop synthetic datasets mimicking real-world scenarios (economic indicators, coupled systems, weather-sales relationships) | Evaluate performance on realistic complexity across both single and multi-variable scenarios |

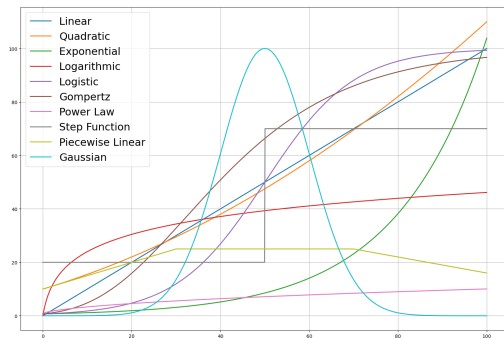

Figure 2: Visualization of diverse trend functions used in our benchmark to systematically evaluate model performance across a wide range of temporal patterns.

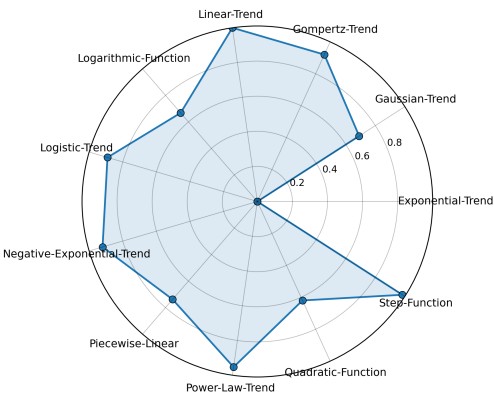

Figure 3: Relative prediction difficulty by trend type. Higher values indicate easier prediction (log scale)

**Experimental Setup** All experiments are driven by a single sliding-window protocol with input length 96 and forecast horizons {10, 24, 48, 96, 192}, using train/validation/test splits of 7:1:2 for deep learning models and 8:2 for traditional methods. Our comprehensive comparison includes 12 fine-tuning models spanning diverse architectural paradigms: Transformer-based models (Autoformer [4], PatchTST [6], iTransformer [12], CATS [20],TimeLLM [21]), MLP-based architectures (TimeMixer [10], TSMixer [8], PaiFilter [22], TexFilter [22], DLinear [5]), N-BEATS [23]), N-HiTS [24]), CNN-based approach (TimesNet [11]), RNN variant (SegRNN [7]), KAN-based model (TimeKAN [9]). Additionally, we evaluate three zero-shot forecasting models (Chronos [25], TimeMoE [26], and Moirai [27]) to assess their generalization capabilities without task-specific fine-tuning.

## 4 Experiments

### 4.1 Evaluation on Trend Datasets and Period Datasets

In real-world scenarios, time series data typically consist of both trend and periodic components, so it is crucial to evaluate model performance on each type individually. In our study, we construct the trend dataset using eleven diverse functions, as illustrated in Figure 2. This design allows for a comprehensive assessment of model generalization across a broad spectrum of realistic trend patterns. For periodic patterns, we relied on the Fourier series theorem, which states that any periodic

Table 2: Comprehensive comparison of MSE and MAE for trend signals (All results are averaged from four different forecasting horizons: $H \in \{24, 48, 96, 192\}$). **Red bold** indicates the best result, blue underlined indicates the second-best, and **green bold** indicates the theoretical optimum. The same notation applies in the following tables.

| Signal Type | Metric | Autoformer | CATS | DLinear | N-BEATS | N-HiTS | PaiFilter | PatchTST | SegRNN | TSMixer | TexFilter | TimeKAN | TimeLLM | TimeMixer | TimesNet | iTransformer | Optimal |
|---|---|---|---|---|---|---|---|---|---|---|---|---|---|---|---|---|---|
| Exponential-Trend | MSE | 8.4545 | 8.18e-02 | **3.40e-08** | 3.63e-07 | 2.32e-03 | 1.52e-07 | 5.42e-02 | 1.9642 | 1.3157 | 1.85e-06 | 8.85e-04 | 9.79e-02 | 2.39e-04 | 6.17e-04 | 8.72e-04 | 0 |
| | MAE | 2.3403 | 1.44e-01 | **8.51e-05** | 4.03e-04 | 3.21e-02 | 3.03e-04 | 1.87e-01 | 0.8362 | 0.9468 | 8.12e-04 | 1.98e-02 | 2.64e-01 | 1.11e-02 | 8.47e-03 | 2.31e-02 | 0 |
| Gaussian-Trend | MSE | 2.00e-02 | 2.01e-04 | 3.09e-04 | 4.39e-05 | 3.44e-03 | 4.09e-06 | 3.39e-05 | 5.24e-04 | 7.61e-03 | 3.21e-06 | 2.34e-04 | 5.45e-05 | 2.17e-05 | 1.32e-04 | 2.93e-04 | 0 |
| | MAE | 1.22e-01 | 8.67e-03 | 9.56e-03 | 3.69e-03 | 3.95e-02 | 9.42e-04 | 3.76e-03 | 1.60e-02 | 6.49e-02 | 1.41e-03 | 9.61e-03 | 4.00e-03 | 2.77e-03 | 7.44e-03 | 9.32e-03 | 0 |
| Gompertz-Trend | MSE | 3.56e-03 | 1.01e-05 | 1.51e-04 | 1.12e-06 | 2.43e-05 | 1.40e-06 | 7.03e-06 | 1.45e-03 | 4.21e-04 | 8.16e-06 | 1.03e-05 | 4.42e-06 | 2.85e-06 | 1.36e-05 | 1.98e-05 | 0 |
| | MAE | 5.16e-02 | 2.65e-03 | 9.25e-03 | 7.79e-04 | 2.95e-03 | 1.06e-03 | 2.06e-03 | 2.56e-02 | 1.67e-02 | 2.48e-03 | 2.83e-03 | 1.53e-03 | 1.21e-03 | 3.17e-03 | 3.71e-03 | 0 |
| Linear-Trend | MSE | 2.02e-03 | 2.13e-04 | 1.10e-09 | 6.44e-10 | 2.13e-05 | 1.34e-12 | 6.26e-05 | 9.43e-07 | 6.64e-04 | 2.71e-12 | 2.71e-06 | 1.05e-04 | 2.95e-07 | 1.96e-07 | 7.93e-07 | 0 |
| | MAE | 3.44e-02 | 8.71e-03 | 1.44e-05 | 1.71e-05 | 3.42e-03 | 7.71e-07 | 6.98e-03 | 7.72e-04 | 2.10e-02 | 1.19e-06 | 1.14e-03 | 9.19e-03 | 4.34e-04 | 1.46e-04 | 7.40e-04 | 0 |
| Logarithmic-Function | MSE | 4.50e-02 | 6.32e-05 | 1.45e-04 | 5.90e-07 | 4.93e-06 | 4.42e-07 | 5.02e-06 | 1.56e-04 | 8.32e-04 | 1.45e-06 | 1.08e-05 | 6.06e-05 | 1.32e-06 | 4.67e-06 | 3.05e-05 | 0 |
| | MAE | 1.42e-01 | 5.16e-03 | 7.20e-03 | 5.39e-04 | 1.70e-03 | 5.63e-04 | 1.86e-03 | 9.85e-03 | 2.58e-02 | 1.05e-03 | 1.78e-03 | 5.73e-03 | 7.06e-04 | 1.33e-03 | 2.93e-03 | 0 |
| Logistic-Trend | MSE | 1.75e-03 | 6.96e-05 | 3.85e-03 | 1.14e-05 | 2.47e-04 | 5.57e-06 | 3.09e-06 | 5.04e-04 | 1.01e-03 | 1.31e-05 | 2.05e-05 | 6.16e-06 | 9.61e-06 | 1.20e-05 | 1.95e-05 | 0 |
| | MAE | 3.40e-02 | 6.06e-03 | 5.63e-02 | 2.12e-03 | 1.06e-02 | 1.53e-03 | 1.44e-03 | 1.47e-02 | 2.13e-02 | 2.53e-03 | 3.03e-03 | 2.26e-03 | 1.76e-03 | 2.57e-03 | 3.19e-03 | 0 |
| Negative-Exponential-Trend | MSE | 3.85e-03 | 1.00e-05 | 6.65e-11 | 5.03e-08 | 1.14e-05 | 8.10e-07 | 7.04e-07 | 1.70e-03 | 6.49e-05 | 5.69e-06 | 5.37e-06 | 4.14e-07 | 2.06e-07 | 5.35e-06 | 6.98e-06 | 0 |
| | MAE | 5.30e-02 | 2.45e-03 | 3.98e-06 | 1.91e-04 | 2.09e-03 | 8.54e-04 | 6.51e-04 | 2.67e-02 | 6.43e-03 | 2.15e-03 | 1.96e-03 | 4.97e-04 | 3.59e-04 | 2.09e-03 | 2.38e-03 | 0 |
| Piecewise-Linear | MSE | 2.40e-02 | 3.74e-04 | 7.16e-05 | 2.65e-05 | 8.67e-03 | 5.53e-06 | 5.01e-05 | 1.81e-04 | 1.17e-03 | 9.68e-06 | 1.53e-05 | 2.33e-04 | 1.49e-05 | 8.14e-05 | 3.14e-05 | 0 |
| | MAE | 1.32e-01 | 1.00e-02 | 5.07e-03 | 2.76e-03 | 6.50e-02 | 1.47e-03 | 4.24e-03 | 1.06e-02 | 2.83e-02 | 1.57e-03 | 1.99e-03 | 1.09e-02 | 2.28e-03 | 3.91e-03 | 2.98e-03 | 0 |
| Power-Law-Trend | MSE | 1.77e-03 | 2.74e-04 | 1.72e-04 | 2.73e-07 | 1.15e-05 | 9.01e-08 | 2.26e-05 | 3.40e-04 | 1.86e-03 | 4.12e-08 | 1.93e-06 | 9.68e-05 | 2.76e-07 | 1.12e-05 | 4.70e-06 | 0 |
| | MAE | 2.83e-02 | 8.10e-03 | 1.05e-02 | 3.68e-04 | 2.34e-03 | 1.79e-04 | 4.13e-03 | 1.25e-02 | 3.89e-02 | 1.75e-04 | 7.63e-04 | 7.94e-03 | 4.14e-04 | 1.80e-03 | 1.29e-03 | 0 |
| Quadratic-Function | MSE | 3.81e-02 | 4.73e-03 | 2.33e-03 | 1.26e-05 | 1.50e-04 | 2.97e-06 | 2.88e-04 | 2.14e-03 | 2.54e-02 | 3.19e-06 | 2.41e-05 | 5.32e-04 | 3.80e-06 | 9.06e-06 | 2.24e-05 | 0 |
| | MAE | 1.61e-01 | 4.28e-02 | 4.44e-02 | 2.14e-03 | 8.32e-03 | 1.01e-03 | 1.53e-02 | 3.33e-02 | 1.42e-01 | 9.21e-04 | 3.28e-03 | 2.21e-02 | 1.33e-03 | 1.59e-03 | 3.24e-03 | 0 |
| Step-Function | MSE | 1.75e-03 | 9.59e-06 | 3.67e-04 | 2.46e-06 | 1.45e-06 | 1.37e-05 | 8.69e-06 | 4.31e-04 | 1.30e-04 | 5.70e-04 | 1.24e-05 | 2.87e-05 | 6.02e-06 | 2.79e-05 | 1.80e-05 | 0 |
| | MAE | 3.43e-02 | 2.83e-03 | 1.45e-02 | 1.36e-03 | 9.84e-04 | 3.64e-03 | 2.66e-03 | 1.56e-02 | 8.94e-03 | 1.69e-02 | 3.36e-03 | 4.72e-03 | 2.32e-03 | 5.14e-03 | 3.52e-03 | 0 |

Table 3: Comprehensive Comparison of MSE and MAE for Periodic Signals (All results are averaged from four different forecasting horizons: $H \in \{24, 48, 96, 192\}$). Signal types are organized by category: Double-Sin, Triple-Sin, Five-Sin, and Ten-Sin represent the superposition of 2, 3, 5, and 10 sine waves with different frequencies, respectively; Exp-Sine-Wave is an exponentially modulated sine wave.

| Signal Type | Metric | Autoformer | CATS | DLinear | N-BEATS | N-HiTS | PaiFilter | PatchTST | SegRNN | TSMixer | TexFilter | TimeKAN | TimeLLM | TimeMixer | TimesNet | iTransformer | Optimal |
|---|---|---|---|---|---|---|---|---|---|---|---|---|---|---|---|---|---|
| Sin(fre=0.005) | MSE | 6.45e-01 | 7.89e-02 | 3.85e-11 | 9.33e-05 | 1.20e-05 | 2.97e-06 | 1.35e-02 | 5.33e-04 | 1.63e-03 | 3.33e-06 | 2.72e-04 | 2.35e-02 | 1.26e-04 | 3.98e-04 | 1.14e-02 | 0 |
| | MAE | 5.53e-03 | 1.95e-01 | 1.61e-06 | 7.56e-03 | 1.67e-03 | 1.33e-03 | 1.00e-01 | 1.67e-02 | 1.63e-03 | 1.38e-03 | 1.30e-02 | 1.31e-01 | 8.87e-03 | 1.14e-02 | 2.64e-02 | 0 |
| Sin(fre=0.05) | MSE | 1.05e-02 | 1.50e-03 | 2.51e-09 | 3.83e-05 | 6.99e-06 | 7.70e-07 | 8.85e-05 | 3.95e-04 | 2.72e-06 | 9.70e-07 | 4.26e-04 | 1.10e-02 | 1.37e-04 | 1.33e-05 | 1.79e-04 | 0 |
| | MAE | 6.91e-02 | 2.76e-02 | 2.27e-05 | 4.53e-03 | 1.24e-03 | 6.48e-04 | 8.11e-02 | 1.63e-02 | 1.22e-03 | 6.55e-04 | 1.76e-02 | 9.23e-02 | 9.56e-03 | 2.86e-03 | 1.15e-02 | 0 |
| Double-Sin | MSE | 3.70e-01 | 1.07e-03 | 9.22e-11 | 4.04e-05 | 6.48e-06 | 1.01e-07 | 8.46e-03 | 1.61e-03 | 3.70e-06 | 2.23e-07 | 3.84e-04 | 2.18e-02 | 2.88e-04 | 6.97e-05 | 4.05e-04 | 0 |
| | MAE | 3.59e-01 | 2.38e-02 | 3.13e-06 | 4.37e-03 | 1.17e-03 | 2.02e-04 | 7.47e-02 | 3.06e-02 | 1.47e-03 | 2.36e-04 | 1.60e-02 | 1.11e-01 | 1.37e-02 | 6.20e-03 | 1.62e-02 | 0 |
| Triple-Sin | MSE | 3.43e-02 | 2.15e-03 | 2.11e-10 | 3.09e-05 | 5.97e-06 | 2.17e-08 | 8.59e-03 | 1.75e-02 | 1.14e-05 | 9.63e-08 | 7.72e-04 | 4.93e-02 | 1.81e-04 | 4.01e-05 | 8.27e-04 | 0 |
| | MAE | 1.14e-01 | 3.18e-02 | 4.51e-06 | 3.39e-03 | 1.10e-03 | 9.01e-05 | 8.19e-02 | 9.85e-02 | 2.67e-03 | 1.64e-04 | 2.36e-02 | 1.64e-01 | 1.05e-02 | 4.83e-03 | 2.14e-02 | 0 |
| Five-Sin | MSE | 6.39e-01 | 3.38e-03 | 6.24e-11 | 3.74e-04 | 4.30e-05 | 5.17e-06 | 8.38e-03 | 3.23e-01 | 2.63e-05 | 5.27e-06 | 3.07e-04 | 4.76e-02 | 2.07e-04 | 1.38e-04 | 4.78e-04 | 0 |
| | MAE | 6.12e-01 | 4.52e-02 | 2.05e-06 | 1.45e-02 | 3.19e-03 | 1.76e-03 | 7.56e-02 | 4.56e-01 | 4.06e-03 | 1.56e-03 | 1.34e-02 | 1.54e-01 | 1.16e-02 | 8.96e-03 | 1.67e-02 | 0 |
| Ten-Sin | MSE | 7.72e-01 | 1.11e-02 | 2.40e-10 | 4.27e-04 | 2.29e-05 | 4.81e-06 | 8.00e-03 | 7.32e-01 | 9.35e-05 | 3.53e-06 | 5.34e-04 | 1.77e-01 | 6.62e-04 | 4.61e-04 | 4.53e-04 | 0 |
| | MAE | 5.27e-01 | 7.16e-02 | 5.22e-06 | 1.58e-02 | 2.35e-03 | 1.64e-03 | 5.09e-02 | 4.30e-01 | 7.41e-03 | 1.24e-03 | 1.52e-02 | 2.09e-01 | 1.87e-02 | 1.22e-02 | 1.26e-02 | 0 |
| Exp-Sine-Wave | MSE | 1.75e-02 | 1.99e-03 | 3.69e-05 | 4.17e-05 | 7.51e-06 | 6.83e-07 | 8.83e-03 | 1.62e-03 | 3.70e-06 | 6.70e-07 | 5.28e-04 | 1.31e-02 | 2.47e-04 | 2.69e-05 | 2.80e-04 | 0 |
| | MAE | 9.05e-02 | 3.21e-02 | 5.28e-03 | 4.50e-03 | 1.28e-03 | 6.03e-04 | 7.93e-02 | 2.89e-02 | 3.27e-03 | 4.84e-04 | 1.94e-02 | 9.39e-02 | 1.23e-02 | 3.91e-03 | 1.32e-02 | 0 |
| Triangle-Wave | MSE | 3.03e-02 | 8.97e-03 | 7.89e-07 | 1.39e-03 | 4.50e-04 | 3.86e-07 | 8.37e-03 | 9.33e-03 | 9.36e-05 | 6.88e-08 | 1.50e-03 | 1.69e-02 | 1.01e-03 | 4.79e-05 | 2.64e-04 | 0 |
| | MAE | 1.10e-01 | 6.83e-02 | 5.37e-04 | 2.60e-02 | 7.90e-03 | 4.55e-04 | 7.75e-02 | 6.50e-02 | 6.51e-03 | 1.58e-04 | 2.74e-02 | 9.08e-02 | 2.25e-02 | 5.33e-03 | 1.30e-02 | 0 |
| Sawtooth-Wave | MSE | 8.22e-02 | 4.15e-02 | 5.14e-10 | 9.23e-07 | 1.70e-06 | 6.58e-09 | 7.93e-03 | 8.55e-02 | 3.43e-05 | 6.74e-09 | 8.46e-04 | 4.18e-02 | 2.23e-04 | 3.22e-05 | 3.31e-04 | 0 |
| | MAE | 1.30e-01 | 1.16e-01 | 6.11e-06 | 5.77e-04 | 6.70e-04 | 5.85e-05 | 7.67e-02 | 1.73e-01 | 4.52e-03 | 6.36e-05 | 2.44e-02 | 1.42e-01 | 1.14e-02 | 4.26e-03 | 1.42e-02 | 0 |
| Square-Wave | MSE | 2.92e-01 | 1.48e-01 | 1.07e-01 | 1.23e-01 | 8.53e-02 | 1.15e-01 | 1.14e-01 | 1.57e-01 | 1.09e-01 | 1.14e-01 | 1.17e-01 | 1.45e-01 | 1.15e-01 | 1.05e-01 | 1.05e-01 | 0 |
| | MAE | 3.24e-01 | 2.44e-01 | 1.46e-01 | 1.99e-01 | 1.19e-01 | 1.65e-01 | 1.81e-01 | 2.56e-01 | 1.64e-01 | 1.77e-01 | 1.79e-01 | 2.43e-01 | 1.76e-01 | 1.48e-01 | 1.46e-01 | 0 |

function can be represented as an infinite sum of sine and cosine functions. This guided our dataset design to include simple sine waves with varying frequencies, superimposed sine waves.

For trend forecasting, Table 2 shows that patterns with accelerating growth are the most challenging. The trend radar chart in Figure 3 visualizes the relative difficulty of each trend type, which is computed based on the MSE of each pattern (with normalization and log transformation applied; see Appendix B.5 for detailed calculation steps). MLP-based architectures clearly dominate trend forecasting, with PaiFilter achieving top performance on 5 out of 11 patterns, followed by TexFilter, N-BEATS and DLinear. Among non-MLP models, only PatchTST (transformer-based) achieved first place once, confirming the effectiveness of MLP approaches for capturing functional trend relationships.

For periodic patterns (Table 3), DLinear demonstrates exceptional capabilities in modeling pure sinusoidal signals, achieving best performance on 7 out of 10 functions with remarkably low errors. All models struggle with Square-Wave patterns characterized by discontinuous jumps.

## 4.2 Robustness Analysis: Model Performance Under Noise and Anomalies

To evaluate the resilience of time series forecasting models under realistic data conditions, we conducted comprehensive robustness experiments with two types of data irregularities: continuous noise at varying intensities and discrete anomalies of different patterns and frequencies.

**Noise and Anomaly Injection.** To mimic measurement errors and irregular disturbances in real-world time series, we corrupted clean synthetic signals with controlled noise and anomalies. For Gaussian noise injection, given a clean series $x_t$, we added white Gaussian noise $\epsilon_t \sim \mathcal{N}(0, \sigma_{\text{noise}}^2)$

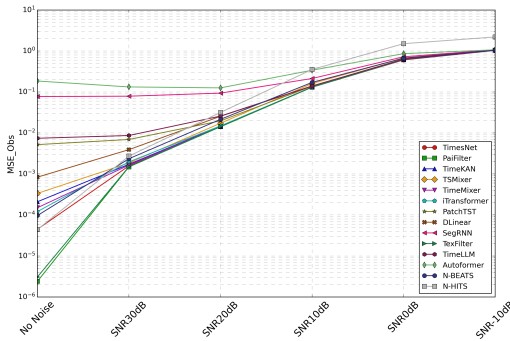
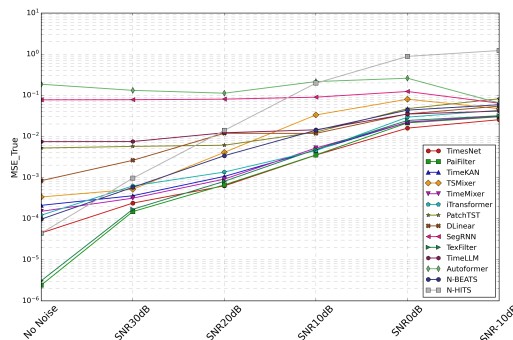

Figure 4: Comparison of $\text{MSE}_{Obs}$ across noise levels for twelve forecasting models.

Figure 5: Comparison of $\text{MSE}_{True}$ across noise levels for twelve forecasting models.

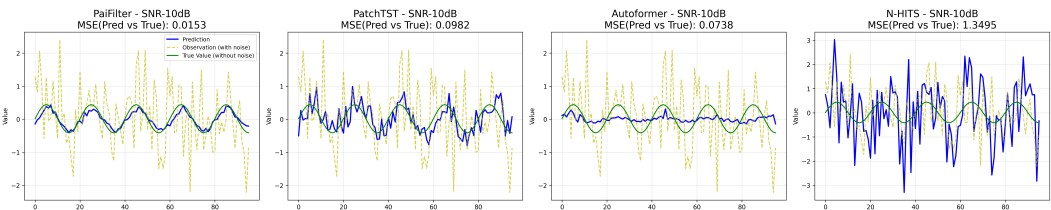

Figure 6: Comparison of model predictions under SNR-10dB noise levels for PaiFilter, PatchTST, and Autoformer.

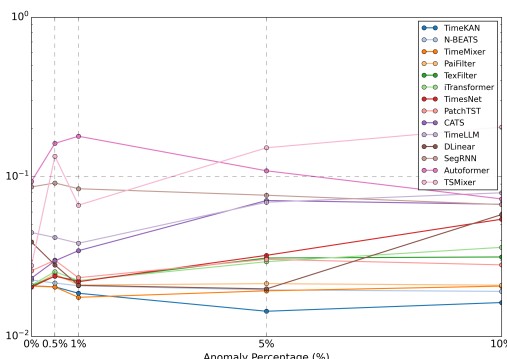

Figure 7: Impact of increasing point anomaly percentage on model performance.

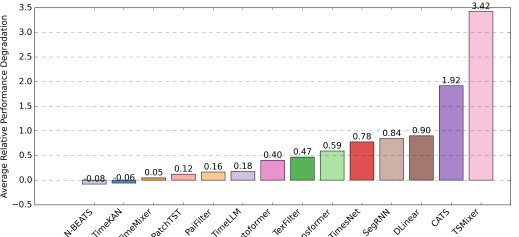

Figure 8: Average relative performance degradation across anomaly scenarios, combining both point and pulse anomalies. Values represent $(MSE_{anomaly} - MSE_{clean})/MSE_{clean}$, indicating how much worse a model performs when anomalies are introduced compared to clean data.

to obtain:

$$y_t = x_t + \epsilon_t, \qquad \text{SNR} = 10\log_{10}\left(\frac{\text{Var}(x_t)}{\sigma_{\text{noise}}^2}\right) \tag{1}$$

where the noise variance $\sigma_{\text{noise}}^2$ was set to achieve a target Signal-to-Noise Ratio (SNR).

For noise evaluation, we tested models across multiple SNR levels from clean data to extreme noise (SNR = -10dB). Beyond Gaussian noise, we also evaluated model robustness under diverse noise distributions including uniform noise, Laplace noise, t-distributions, and heavy-tailed Lévy stable distributions to assess model performance under various realistic noise characteristics (detailed results in Table 12 in Appendix B.5). We measured both the error between predictions and noisy observations ($\text{MSE}_{Obs}$) and between predictions and the underlying clean signal ($\text{MSE}_{True}$):

$$\text{MSE}_{Obs} = \frac{1}{n}\sum_{t=1}^{n}(y_t - \hat{y}_t)^2, \quad \text{MSE}_{True} = \frac{1}{n}\sum_{t=1}^{n}(x_t - \hat{y}_t)^2 \tag{2}$$

Table 4: Comparison of MSE for Different Noise Levels (Forecasting Horizons: $H = 96$)

| Noise Level(SNR) | Metric | Autoformer | CATS | DLinear | N-BEATS | N-HiTS | PaiFilter | PatchTST | SegRNN | TSMixer | TexFilter | TimeKAN | TimeLLM | TimeMixer | TimesNet | iTransformer | Optimal |
|---|---|---|---|---|---|---|---|---|---|---|---|---|---|---|---|---|---|
| -10dB | $MSE_{Obs}$ | 1.0697 | 1.0431 | 1.0629 | 1.0500 | 2.1988 | 1.0383 | 1.0861 | 1.0681 | 1.0537 | 1.0383 | 1.0402 | 1.0407 | 1.0383 | 1.0328 | 1.0509 | 1.0045 |
|  | $MSE_{True}$ | 0.0648 | 0.0533 | 0.0542 | 0.0588 | 1.2221 | 0.0300 | 0.0820 | 0.0640 | 0.0471 | 0.0318 | 0.0314 | 0.0411 | 0.0309 | 0.0254 | 0.0432 | 0.0000 |
| 0dB | $MSE_{Obs}$ | 0.8610 | 0.6451 | 0.6248 | 0.6683 | 1.5074 | 0.6105 | 0.6396 | 0.7221 | 0.6765 | 0.6138 | 0.6112 | 0.6256 | 0.6102 | 0.6046 | 0.6182 | 0.5886 |
|  | $MSE_{True}$ | 0.2595 | 0.0199 | 0.0352 | 0.0435 | 0.8817 | 0.0213 | 0.0472 | 0.1234 | 0.0801 | 0.0244 | 0.0222 | 0.0349 | 0.0220 | 0.0159 | 0.0293 | 0.0000 |
| 10dB | $MSE_{Obs}$ | 0.3395 | 0.1651 | 0.1386 | 0.1712 | 0.3535 | 0.1306 | 0.1389 | 0.2131 | 0.1605 | 0.1317 | 0.1314 | 0.1405 | 0.1323 | 0.1306 | 0.1314 | 0.1267 |
|  | $MSE_{True}$ | 0.2149 | 0.0075 | 0.0118 | 0.0141 | 0.1970 | 0.0035 | 0.0127 | 0.0898 | 0.0332 | 0.0048 | 0.0046 | 0.0144 | 0.0052 | 0.0035 | 0.0045 | 0.0000 |
| 20dB | $MSE_{Obs}$ | 0.1260 | 0.0230 | 0.0256 | 0.0215 | 0.0317 | 0.0142 | 0.0198 | 0.0939 | 0.0173 | 0.0143 | 0.0146 | 0.0258 | 0.0145 | 0.0142 | 0.0149 | 0.0134 |
|  | $MSE_{True}$ | 0.1121 | 0.0048 | 0.0119 | 0.0034 | 0.0140 | 0.0007 | 0.0061 | 0.0804 | 0.0041 | 0.0008 | 0.0011 | 0.0122 | 0.0009 | 0.0006 | 0.0014 | 0.0000 |
| 30dB | $MSE_{Obs}$ | 0.1328 | 0.0051 | 0.0039 | 0.0024 | 0.0027 | 0.0015 | 0.0069 | 0.0789 | 0.0019 | 0.0015 | 0.0017 | 0.0087 | 0.0016 | 0.0016 | 0.0020 | 0.0013 |
|  | $MSE_{True}$ | 0.1315 | 0.0033 | 0.0026 | 0.0006 | 0.0010 | 0.0001 | 0.0057 | 0.0778 | 0.0005 | 0.0002 | 0.0004 | 0.0075 | 0.0003 | 0.0002 | 0.0006 | 0.0000 |
| No Noise | $MSE_{Obs}$ | 0.1852 | 0.0015 | 0.0008 | 9.68e-5 | 4.42e-5 | 2.41e-6 | 0.0052 | 0.0773 | 0.0003 | 3.08e-6 | 0.0002 | 0.0074 | 0.0002 | 4.47e-5 | 0.0001 | 0.0000 |
|  | $MSE_{True}$ | 0.1852 | 0.0015 | 0.0008 | 9.68e-5 | 4.42e-5 | 2.41e-6 | 0.0052 | 0.0773 | 0.0003 | 3.08e-6 | 0.0002 | 0.0074 | 0.0002 | 4.47e-5 | 0.0001 | 0.0000 |

Table 5: Comparison of MSE, MAE for Different Signal Types (Forecasting Horizons: $H = 10$)

| Signal Type | Metric | Autoformer | CATS | DLinear | N-BEATS | N-HiTS | PaiFilter | PatchTST | SegRNN | TSMixer | TexFilter | TimeKAN | TimeLLM | TimeMixer | TimesNet | iTransformer | arima | mean | naive |
|---|---|---|---|---|---|---|---|---|---|---|---|---|---|---|---|---|---|---|---|
| ARMA(1,1) | MSE | 8.6800 | 7.2728 | 6.4687 | 7.6396 | 9.9578 | 7.8367 | 7.5958 | 6.6061 | 7.1438 | 7.5022 | 6.6782 | 7.1420 | 7.1104 | 7.7908 | 8.2393 | 6.0889 | 8.7105 | 9.2425 |
|  | MAE | 2.3520 | 2.1275 | 1.9787 | 2.1576 | 2.4638 | 2.2158 | 2.1505 | 2.0009 | 2.1045 | 2.1454 | 2.0159 | 2.1025 | 2.0988 | 2.1974 | 2.2521 | 1.9151 | 2.3522 | 2.3342 |
| ARMA(2,2) | MSE | 4.8409 | 4.5999 | 4.4971 | 4.8195 | 6.5592 | 4.7068 | 4.9648 | 4.4977 | 4.5485 | 4.7868 | 4.4556 | 4.6613 | 4.6155 | 4.8728 | 4.8555 | 4.1972 | 4.8037 | 7.9638 |
|  | MAE | 1.7509 | 1.7050 | 1.6690 | 1.7383 | 2.0147 | 1.7243 | 1.7556 | 1.6697 | 1.6915 | 1.7354 | 1.6590 | 1.7158 | 1.7040 | 1.7533 | 1.7481 | 1.6131 | 1.7544 | 2.2134 |
| ARMA-Long-Dependency | MSE | 2.0083 | 1.9572 | 1.0402 | 1.1443 | 1.8921 | 1.1692 | 1.1431 | 1.9756 | 1.1343 | 1.1881 | 1.0581 | 1.5247 | 1.0933 | 1.1133 | 1.1950 | 1.0070 | 1.9769 | 3.9410 |
|  | MAE | 1.1375 | 1.1240 | 0.8156 | 0.8553 | 1.0987 | 0.8638 | 0.8533 | 1.1280 | 0.8524 | 0.8708 | 0.8227 | 0.9855 | 0.8357 | 0.8434 | 0.8722 | 0.8020 | 1.1293 | 1.5873 |
| Random-Walk | MSE | 25.9527 | 9.2639 | 11.2441 | 6.7003 | 21.8189 | 6.9310 | 6.6035 | 84.4943 | 6.7319 | 5.7561 | 6.3726 | 6.1647 | 7.0258 | 7.9997 | - | - | - | 5.3786 |
|  | MAE | 3.9874 | 2.3754 | 2.5298 | 1.9929 | 3.4250 | 2.0318 | 1.9795 | 7.4527 | 1.9970 | 1.8419 | 1.9414 | 1.9150 | 2.0552 | 2.1885 | - | - | - | 1.7765 |
| White-Noise | MSE | 1.0078 | 0.9965 | 1.0261 | 1.0561 | 1.6940 | 1.0013 | 1.0845 | 1.0075 | 0.9880 | 1.0067 | 0.9969 | 1.0003 | 1.0015 | 1.0083 | 1.0304 | - | 0.9861 | - |
|  | MAE | 0.7989 | 0.7953 | 0.8048 | 0.8178 | 1.0378 | 0.7970 | 0.8282 | 0.7997 | 0.7917 | 0.7989 | 0.7955 | 0.7967 | 0.7971 | 0.8003 | 0.8088 | - | 0.7910 | - |

where $y_t$ represents the noisy observed value, $x_t$ represents the clean signal, and $\hat{y}_t$ represents the model's prediction.

For anomaly testing, we introduced two types of irregularities: point anomalies (isolated outliers at random positions with deviations $\Delta_t \sim \mathcal{N}(0, \sigma^2_{\text{anomaly}})$) at rates from 0.5% to 10%, and pulse anomalies (clustered disturbances spanning multiple consecutive timestamps using deterministic pulses $\gamma_t$) with varying numbers (1, 3, and 5). Additionally, we evaluated model resilience under distribution shift scenarios including mean shifts and trend shifts (comprehensive results in Table 11 in Appendix B.5).

### 4.2.1 Performance Under Noise Conditions

As shown in Figures 4 and 5, when noise intensity increases to extreme levels (SNR = -10dB), $MSE_{Obs}$ values across most of models converge to approximately 1, making this metric ineffective for distinguishing model performance. In contrast, $MSE_{True}$ maintains its discriminative power even under severe noise conditions, revealing substantial performance differences between architectures. Figure 6 illustrates this phenomenon, where under intense noise, PaiFilter predictions closely align with the true underlying signal, while Autoformer fails to extract meaningful patterns beneath the noise. N-HiTS performs even worse, with predictions clearly overfitting to the noise, resulting in significantly degraded forecasting quality. This highlights that despite similar $MSE_{Obs}$ values, different models may be learning fundamentally different patterns from the data.

Table 4 provides quantitative evidence that TimesNet achieves the best performance under extreme noise conditions, closely followed by FilterNet models (PaiFilter, TexFilter). Notably, transformer-based architectures show the most significant performance degradation as noise intensifies.

### 4.2.2 Resilience to Anomalies

Figure 7 illustrates the distinct behavioral patterns of various architectures as point anomaly percentages increase. FilterNet models (PaiFilter and TexFilter) and TimeKAN maintain relatively stable error rates even with 10% anomalies. Figure 8 quantifies these observations through relative performance degradation metrics across all anomaly types. N-BEATS and TimeKAN demonstrate exceptional resilience with negligible degradation even under significant anomaly presence, while TimeMixer follows closely behind with only 5% performance deterioration. Most concerning are N-HiTS and TSMixer's vulnerabilities: N-HiTS exhibits the worst robustness with catastrophic performance collapse under anomalies (its extreme degradation is excluded from visualization for clarity; detailed results in Table 11 in Appendix B.5), while TSMixer shows an alarming 342% error increase as anomaly density rises, revealing critical weaknesses in their architectures when handling corrupted inputs.

Table 6: Comparison of MSE, MAE for Multivariate Time Series (Forecasting Horizons: $H = 96$)

| Signal Type | Metric | Model | | | | | | | | | | | | | | | Optimal |
|---|---|---|---|---|---|---|---|---|---|---|---|---|---|---|---|---|---|
| | | Autoformer | CATS | DLinear | N-BEATS | N-HiTS | PaiFilter | PatchTST | SegRNN | TSMixer | TexFilter | TimeKAN | TimeLLM | TimeMixer | TimesNet | iTransformer | |
| Feature Lag (5 steps) | MSE | 1.0930 | 1.0797 | 1.0925 | 1.1202 | 2.3238 | 1.0777 | 1.1430 | 1.0644 | 1.0450 | 1.0889 | 1.0682 | 1.0634 | 1.0780 | 1.0452 | 1.0793 | 1.0174 |
| | MAE | 0.8240 | 0.8168 | 0.8215 | 0.8318 | 1.2116 | 0.8141 | 0.8397 | 0.8113 | 0.8088 | 0.8190 | 0.8126 | 0.8105 | 0.8117 | 0.7969 | 0.8175 | 0.7877 |
| Feature Lag (10 steps) | MSE | 1.0257 | 1.0626 | 1.0533 | 1.1099 | 2.2863 | 1.0346 | 1.0949 | 1.0269 | 0.9993 | 1.0483 | 1.0267 | 1.0256 | 1.0331 | 0.9820 | 1.0180 | 0.9572 |
| | MAE | 0.8058 | 0.8203 | 0.8159 | 0.8395 | 1.2050 | 0.8096 | 0.8330 | 0.8062 | 0.7894 | 0.8149 | 0.8063 | 0.8059 | 0.7956 | 0.7763 | 0.7953 | 0.7568 |
| Feature Lag (24 steps) | MSE | 1.0213 | 1.0597 | 1.0409 | 1.0921 | 2.2911 | 1.0266 | 1.0757 | 1.0227 | 0.9004 | 1.0307 | 1.0165 | 1.0162 | 0.9448 | 0.9007 | 0.9281 | 0.8861 |
| | MAE | 0.8083 | 0.8237 | 0.8153 | 0.8336 | 1.2069 | 0.8108 | 0.8250 | 0.8094 | 0.7301 | 0.8108 | 0.8070 | 0.8070 | 0.7500 | 0.7272 | 0.7410 | 0.7056 |
| Feature Lag (48 steps) | MSE | 0.9152 | 0.9941 | 1.0033 | 1.0286 | 2.1798 | 0.9874 | 1.0348 | 0.9746 | 0.7527 | 0.9901 | 0.9743 | 0.9727 | 0.7646 | 0.7519 | 0.7740 | 0.7274 |
| | MAE | 0.7598 | 0.7965 | 0.7948 | 0.8105 | 1.1760 | 0.7946 | 0.8085 | 0.7879 | 0.6328 | 0.7944 | 0.7891 | 0.7882 | 0.6328 | 0.6335 | 0.6480 | 0.5875 |
| Sine-Noise Composition | MSE | 0.3302 | 0.3394 | 0.3333 | 0.3427 | 0.6785 | 0.3305 | 0.3402 | 0.3330 | 0.3218 | 0.3330 | 0.3297 | 0.3350 | 0.3300 | 0.3261 | 0.3325 | 0.3208 |
| | MAE | 0.3875 | 0.4721 | 0.3800 | 0.4634 | 0.6472 | 0.3827 | 0.4005 | 0.3841 | 0.3754 | 0.3904 | 0.3787 | 0.4009 | 0.3804 | 0.3786 | 0.3866 | 0.3684 |
| Conditional Relationship | MSE | 1.0149 | 1.0028 | 0.9972 | 1.0266 | 1.9346 | 1.0030 | 1.0241 | 1.0027 | 0.8332 | 1.0016 | 1.0049 | 1.0021 | 0.8533 | 0.8665 | 0.8955 | 0.7967 |
| | MAE | 0.9014 | 0.9005 | 0.8921 | 0.9022 | 1.1292 | 0.8997 | 0.8984 | 0.9026 | 0.7410 | 0.8981 | 0.9024 | 0.8985 | 0.7633 | 0.7724 | 0.8031 | 0.6736 |
| Nonlinear Relationship | MSE | 0.9331 | 0.9656 | 0.9869 | 1.0010 | 1.9523 | 0.9795 | 1.0205 | 0.9670 | 0.7302 | 0.9788 | 0.9656 | 0.9738 | 0.7481 | 0.7385 | 0.7617 | 0.7136 |
| | MAE | 0.7726 | 0.7925 | 0.8054 | 0.8058 | 1.1154 | 0.7981 | 0.8139 | 0.7930 | 0.6340 | 0.7973 | 0.7925 | 0.7955 | 0.6354 | 0.6333 | 0.6519 | 0.6090 |
| Multivariable Complex | MSE | 0.9880 | 1.0075 | 1.0279 | 1.0380 | 1.6438 | 1.0272 | 1.0557 | 1.0087 | 0.9374 | 1.0191 | 1.0069 | 1.0117 | 0.8901 | 0.9105 | 1.0066 | 0.8767 |
| | MAE | 0.8417 | 0.8613 | 0.8670 | 0.8680 | 1.0431 | 0.8674 | 0.8745 | 0.8621 | 0.8104 | 0.8620 | 0.8610 | 0.8619 | 0.7667 | 0.7847 | 0.8416 | 0.7531 |

## 4.3 Capturing Short-range and Long-range Dependencies in Time Series

To evaluate temporal dependency modeling capabilities, we tested models on five stochastic time series patterns with varying autocorrelation structures, as shown in Table 5. For standard ARMA processes, traditional statistical methods (ARIMA) maintain a slight edge, with DLinear and TimeKAN achieving the closest deep learning performance on ARMA(1,1) and ARMA(2,2) respectively, indicating their strong capability to capture short-range temporal dependencies in structured stochastic processes. In our specialized ARMA-Long-Dependency test with lag-50 temporal relationships, DLinear and TimeKAN approached ARIMA's benchmark, while Autoformer significantly underperformed despite its Auto-Correlation Mechanism being specifically designed for temporal pattern extraction. Similarly, SegRNN's slice encoding approach, which was developed to optimize long-range dependency modeling, failed to effectively capture these sparse long-range relationships. For Random-Walk processes, SegRNN and TimeKAN nearly matched the theoretical optimum (naive forecast), while TSMixer performed poorly with MSE 15 times higher than the best model.

## 4.4 Evaluation on Cross-Variable Learning

To evaluate models' ability to capture relationships between time series variables, we designed controlled experiments isolating specific types of variable dependencies, as shown in Table 6. Our evaluation framework encompasses five key scenarios with progressively increasing complexity: **Feature Lag Relationships** testing temporal dependencies where one variable is a lagged version of another (lag values: 5, 10, 24, and 48 steps), simulating delayed effects commonly observed in economic indicators and sensor networks; **Sine-Noise Composition** evaluating models' ability to decompose linear additive relationships where one variable represents the sum of a periodic signal and random noise; **Conditional Relationships** assessing capability to capture variable interactions

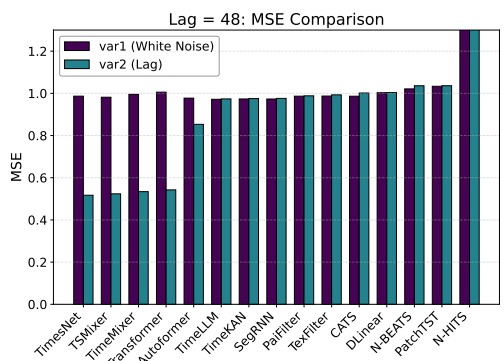

Figure 9: MSE comparison for lag relationship detection with lag value of 48.

that depend on specific conditions or thresholds, mimicking decision-driven processes in real-world systems; **Nonlinear Relationships** testing models on variables connected through nonlinear transformations, representing complex physical or economic relationships; and **Multivariable Complex** examining performance on five-variable systems with intricate interdependencies and feedback mechanisms.

For the lag relationship experiments, we generated two variables: var1 as white noise and var2 as var1 lagged by varying steps. A model that successfully learns the temporal dependency should perform significantly better on var2 prediction despite both signals having similar statistical properties. As illustrated in Figure 9, channel-dependent architectures such as TimesNet, TSMixer, and TimeMixer demonstrate substantially lower MSE when predicting var2 compared to var1, with reductions of approximately 50%, indicating successful recognition and leverage of temporal relationships between variables. In contrast, channel-independent models like PaiFilter, TexFilter, and

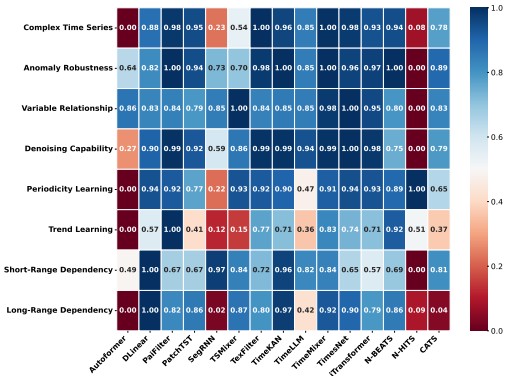

Figure 10: Performance heatmap of deep learning models across different forecasting capabilities. Values are normalized using logarithmic transformation. Larger values mean better performance.

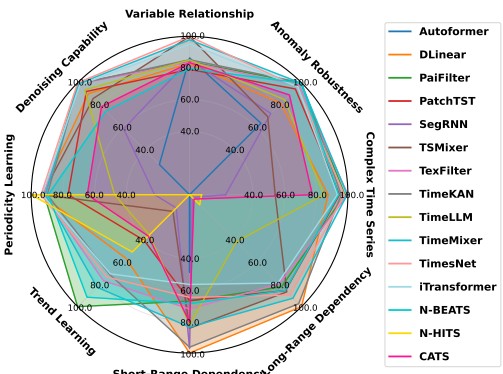

Figure 11: Radar chart visualization of model capabilities across different forecasting dimensions. Larger area coverage indicates broader forecasting competence. MSE values are log-transformed and normalized.

PatchTST show similar performance on both variables, confirming their inherent design limitation in utilizing cross-variable information.

Across all multivariate scenarios, TSMixer and TimesNet consistently outperform other architectures, demonstrating superior capability in capturing diverse cross-variable dependencies. Channel-dependent models show clear advantages on lag relationships, additive compositions, conditional interactions, and nonlinear transformations. The performance advantage becomes particularly pronounced in the complex five-variable scenario, where TimeMixer and TimesNet achieve near-optimal performance, highlighting the effectiveness of channel-mixing mechanisms for modeling intricate multivariate interactions.

## 4.5 Other experiments and brief analysis

We conducted several additional experiments to evaluate model performance across diverse aspects. We tested how varying dataset lengths impact learning efficiency and convergence rates(Appendix B.1), evaluated model performance on synthetic datasets that mimic real-world scenarios including economic indicators, coupled dynamic systems, and weather-sales relationships across both single and multi-variable settings(Appendix B.2). We also assessed the forecasting capabilities of contemporary zero-shot models, examining their ability to generalize to unseen time series patterns without task-specific fine-tuning(Appendix B.3). Additionally, we also investigated the limitations of batch normalization which is a widely adopted technique in time series forecasting(Appendix B.4). As illustrated in Figures 10 and 11, these experiments provide a comprehensive visualization of each model's strengths and weaknesses across multiple forecasting dimensions. Detailed experimental results, methodologies, and in-depth analyses for all these experiments can be found in the appendix.

As visualized in Figure 10, model performance varies considerably across different forecasting dimensions. TimeKAN, TimeMixer, PaiFilter, TexFilter and TimesNet demonstrate strong overall performance across most capabilities, though each exhibits specific weaknesses: TimeKAN, PaiFilter, TexFilter struggle with capturing variable relationships between time series, and TimesNet performs poorly in short distance dependency modeling. The radar chart in Figure 11 further illustrates these competence variations, with these three models covering relatively large areas despite their individual shortcomings. Models like N-HiTS, Autoformer and SegRNN show significant weaknesses in lots of areas. Notably, no single model achieves optimal performance across all dimensions, emphasizing the importance of task-specific model selection based on the dominant characteristics of the target time series. For a more comprehensive analysis of each model's strengths and weaknesses, please refer to Appendix B.7.

# 5    Conclusion and Future Work

We introduce a systematic evaluation framework for time series forecasting models, leveraging synthetic datasets based on traditional stochastic time series and signal processing models, with programmable feature configurations to rigorously isolate and quantify distinct modeling capabilities. By decoupling core temporal features and establishing theoretical performance boundaries, this framework overcomes key limitations of traditional evaluation protocols, enabling more precise attribution of model strengths and weaknesses than is possible with real-world benchmarks alone.

Through extensive experimentation, we demonstrate that, despite significant progress in the field, state-of-the-art forecasting models continue to struggle with extreme noise, complex multivariate dependencies, and zero-shot generalization. Our results highlight that no single architecture achieves optimal performance across all temporal patterns, underscoring inherent trade-offs in model design. The quantifiable performance boundaries established in this work lay the groundwork for future research aimed at targeted architectural improvements and more principled model selection tailored to the specific characteristics of time series forecasting tasks. Future work could explore methods to enhance model generalization across diverse temporal patterns, such as developing architectures with improved universality and adaptability to better address the complexity and variability inherent in real-world forecasting tasks.

## Acknowledgments and Disclosure of Funding

This work is supported by Shenzhen Ubiquitous Data Enabling Key Lab under grant ZDSYS20220527171406015, and by Tsinghua Shenzhen International Graduate School-Shenzhen Pengrui Endowed Professorship Scheme of Shenzhen Pengrui Foundation.

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

# A  Related Work

**Time Series Forecasting Models**   Time series forecasting has evolved through several paradigms, from classical statistical methods to advanced deep learning approaches. Traditional statistical methods like ARIMA [1], ETS [2], and VAR [3] provided interpretable frameworks based on explicit statistical assumptions but struggled with complex non-linear patterns. Machine learning approaches, including XGBoost [28, 29], Random Forests [30], and LightGBM [31], offered improved flexibility but often required manual feature engineering. The deep learning revolution brought a diverse array of architectures for time series modeling, progressing from MLP-based models (N-BEATS [32], DLinear [5], N-HiTS [24], NLinear [5]) to RNN architectures [33, 34, 35, 36] (LSTM [37, 38], GRU [39, 40, 41], DeepAR [42], SegRNN [7]) and CNN architectures (TCN [43], WaveNet [44], SCINet [45], TimesNet [11]), and then to transformer-based models (Informer [46], Autoformer [4], PatchTST [6], FEDformer [47], Pyraformer [48], iTransformer [12]). Recent advances include specialized architectures like TimesNet [11] for periodic pattern modeling, TSMixer [8] and TimeMixer [10] for efficient temporal mixing, and filter-based approaches (PaiFilter [22], TexFilter [22]). Meanwhile, foundation models for time series have emerged, including Chronos [25], Lag-Llama [49], TimesFM [50], Timer [51], MOIRAI [27], UniTS [52],and TimeMoE [26], offering promising zero-shot forecasting capabilities across diverse domains.

**Time Series Benchmarks and Evaluation**   A variety of benchmarks have been proposed for time series forecasting, such as M3 [53], M4 [54], Monash [55], LTSF-Linear [5], BasicTS [16], and BasicTS+ [17]. However, these benchmarks are not comprehensive: some focus only on univariate or multivariate forecasting, and many either omit statistical methods or deep learning methods, resulting in limited coverage of different modeling paradigms. Recently, more comprehensive benchmarks have emerged to carry out more detailed evaluations. ProbTS [14] advances probabilistic forecasting evaluation with improved uncertainty quantification, TSlib [18] covers multiple time series tasks including forecasting, anomaly detection, and missing value imputation within a unified pipeline, and TFB [15] provides fine-grained dataset categorization with scalable integration of diverse methods. Despite these improvements, existing benchmarks still struggle to systematically isolate specific temporal patterns, establish theoretical performance boundaries, or provide interpretable capability mapping. Our SynTSBench framework is designed to fill these gaps by enabling controllable synthetic data generation, systematic decomposition of temporal features, and rigorous, interpretable evaluation of model capabilities across a wide range of forecasting scenarios. As shown in Table 7, this table presents a comparison between SynTSBench and other benchmarks.

Table 7: Time series forecasting benchmark comparison. ✓ indicates present, ✗ indicates absent, – indicates incomplete.

| Benchmark | Univariate Forecasting | Multivariate Forecasting | Patterns-learning Assessment | Theoretical Performance Boundaries | Cross-variable Relationship Assessment | Robustness Testing | Foundation Models Zero-shot Testing |
|---|---|---|---|---|---|---|---|
| M3 [53] | ✓ | ✗ | ✗ | ✗ | ✗ | ✗ | ✗ |
| M4 [54] | ✓ | ✗ | ✗ | ✗ | ✗ | ✗ | ✗ |
| TSlib [18] | ✓ | ✓ | ✗ | ✗ | ✗ | – | ✗ |
| BasicTS [16] | ✗ | ✓ | ✗ | ✗ | ✗ | ✗ | ✗ |
| BasicTS+ [17] | ✗ | ✓ | – | ✗ | ✗ | ✗ | ✗ |
| Monash [55] | ✓ | ✗ | – | ✗ | ✗ | ✗ | ✗ |
| ProbTS [14] | ✓ | ✓ | – | ✗ | ✗ | ✗ | ✓ |
| TFB [15] | ✓ | ✓ | – | ✗ | ✗ | ✗ | ✗ |
| SynTSBench (Ours) | ✓ | ✓ | ✓ | ✓ | ✓ | ✓ | ✓ |

# B  Additional Experiments and Supplementary Results

We provide additional experiments and results to further validate SynTSBench. Appendix B.1 evaluates the effect of dataset length on model performance. Appendix B.2 presents results on complex real-world pattern simulations. Appendix B.3 examines zero-shot forecasting capabilities of foundation models. Appendix B.4 investigates the impact of normalization layers on model performance. Appendix B.5 provides supplementary visualizations and detailed experimental results. Appendix B.6 validates the benchmark through extensive experiments on real-world datasets. Appendix B.7 presents comprehensive model capability analysis and architectural trade-offs.

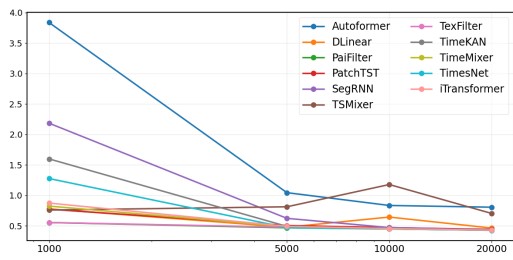
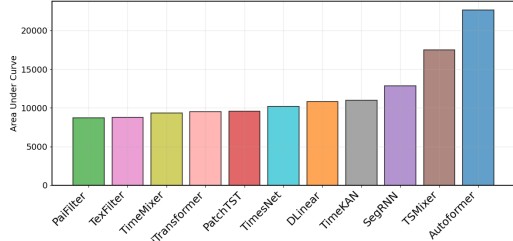

Figure 12: Average MSE by Dataset Length for Different Models

Figure 13: Model Efficiency Ranking (Lower AUC means better performance)

## B.1 Length of dataset

In this experiment, we evaluated the performance of various models on datasets of different lengths: 1000, 5000, 10000, and 20000. Figure 12 shows the average MSE for each model as the dataset length increases. It is evident that most models benefit from longer datasets, with their MSE values decreasing as more data becomes available.

To further quantify the efficiency of each model in learning from datasets of varying lengths, we calculated the area under the curve (AUC) formed by each model's MSE curve and the x-axis. A smaller AUC indicates better efficiency in learning from the data. Figure 13 ranks the models based on their AUC values, where lower values correspond to higher efficiency.

The results demonstrate that models based on FilterNet, specifically PaiFilter and TexFilter, achieved the best performance, as they consistently exhibited lower MSE values and smaller AUCs compared to other models. This highlights their superior ability to learn effectively from datasets of varying lengths.

## B.2 Performance on Complex Real-world Pattern Simulations

Table 8: MSE and MAE Results for Univariate Datasets. **Red bold** indicates best performance, blue underlined indicates second-best performance. (Forecasting Horizons: $H = 96$) The following tables use the same notation.

| Dataset | Metric | Autoformer | CATS | DLinear | N-BEATS | N-HiTS | PaiFilter | PatchTST | SegRNN | TSMixer | TexFilter | TimeKAN | TimeLLM | TimeMixer | TimesNet | iTransformer | Optimal |
|---|---|---|---|---|---|---|---|---|---|---|---|---|---|---|---|---|---|
| Economic-Indicator | MSE | 0.1017 | 0.0852 | 0.1017 | 0.0624 | 0.8073 | 0.0721 | 0.0603 | 0.0617 | 0.8370 | 0.0548 | **0.0534** | 0.0547 | 0.0545 | **0.0534** | 0.0607 | **0.0424** |
|  | MAE | 0.2564 | 0.2310 | 0.2453 | 0.2003 | 0.7568 | 0.2095 | 0.1989 | 0.1984 | 0.7999 | 0.1885 | **0.1836** | 0.1876 | 0.1878 | 0.1872 | 0.1992 | **0.1609** |
| Electricity-Consumption | MSE | 0.3593 | 0.1755 | 0.1921 | 0.1295 | 0.1309 | 0.1201 | 0.1269 | 0.1719 | 0.1298 | **0.1183** | 0.1523 | 0.1605 | 0.1324 | 0.1243 | _0.1189_ | 0.0833 |
|  | MAE | 0.4494 | 0.3259 | 0.3358 | 0.2585 | 0.2687 | _0.2426_ | 0.2571 | 0.3203 | 0.2630 | 0.2427 | 0.2897 | 0.3036 | 0.2592 | 0.2450 | 0.2414 | 0.1851 |
| Industrial-Sensor | MSE | 0.1027 | 0.0240 | 0.0227 | 0.0197 | 0.0464 | 0.0188 | 0.0289 | 0.2849 | 0.0270 | 0.0187 | 0.0191 | 0.0350 | 0.0189 | _0.0177_ | **0.0172** | 0.0151 |
|  | MAE | 0.2533 | 0.1229 | 0.1199 | 0.1115 | 0.1735 | 0.1085 | 0.1353 | 0.4099 | 0.1307 | 0.1083 | 0.1098 | 0.1487 | 0.1090 | _0.1049_ | **0.1037** | 0.0974 |
| Industrial-Sensor-Normal | MSE | 0.2441 | 0.0726 | 0.0608 | 0.0630 | 0.0638 | _0.0533_ | 0.0590 | 0.1643 | 0.0594 | _0.0542_ | 0.0556 | 0.0694 | 0.0547 | 0.0552 | 0.0544 | 0.0495 |
|  | MAE | 0.3165 | 0.2153 | 0.1963 | 0.1989 | 0.1995 | **0.1831** | 0.1921 | 0.3257 | 0.1931 | _0.1847_ | 0.1870 | 0.2092 | 0.1854 | 0.1864 | 0.1847 | 0.1772 |
| Network-Traffic | MSE | 1.3672 | 0.9323 | 1.1490 | 0.9158 | 1.0625 | 0.8882 | 0.9019 | 1.2382 | 1.1145 | _0.8804_ | 0.8855 | **0.8644** | 0.9037 | 0.8904 | 0.9277 | 0.8120 |
|  | MAE | 0.7985 | 0.5710 | 0.6810 | 0.5601 | 0.6223 | 0.5594 | 0.5416 | 0.7490 | 0.6427 | 0.5493 | _0.5083_ | **0.5288** | 0.5368 | 0.5555 | 0.5326 | 0.3413 |
| Retail-Sales | MSE | 0.7671 | 0.5642 | _0.4314_ | 0.4630 | 0.4645 | 0.4715 | 0.4549 | 0.6358 | 0.4423 | 0.4765 | **0.4005** | 0.5329 | 0.4379 | 0.4919 | 0.5001 | 0.2708 |
|  | MAE | 0.7087 | 0.5877 | 0.4539 | 0.5050 | 0.4734 | 0.5083 | 0.4861 | 0.6312 | 0.5096 | **0.4434** | 0.5599 | 0.4562 | 0.5145 | 0.5147 | | 0.2535 |
| Stock-Price | MSE | 0.1933 | 0.1853 | 0.1898 | 0.1802 | 0.1833 | 0.1812 | 0.1794 | 0.1774 | 0.1815 | 0.1779 | **0.1660** | 0.1742 | _0.1663_ | 0.1825 | 0.1797 | 0.1199 |
|  | MAE | 0.3460 | 0.3380 | 0.3424 | 0.3335 | 0.3360 | 0.3360 | 0.3327 | 0.3327 | 0.3308 | 0.3301 | _0.3213_ | 0.3271 | **0.3182** | 0.3347 | 0.3317 | 0.2708 |
| Temperature-Sensor | MSE | 0.1750 | 0.1079 | 0.1547 | 0.0675 | 0.1672 | **0.0628** | 0.0703 | 0.0772 | 0.1549 | 0.0653 | 0.0673 | 0.1101 | 0.0653 | 0.0641 | _0.0639_ | 0.0508 |
|  | MAE | 0.3357 | 0.2705 | 0.3186 | 0.2051 | 0.3404 | **0.1974** | 0.2110 | 0.2211 | 0.3245 | 0.2056 | 0.2718 | 0.2020 | 0.2001 | _0.1996_ | 0.1847 | 0.1768 |
| Website-Traffic | MSE | 0.4310 | 0.2840 | 0.0799 | 0.0688 | 0.0654 | 0.0531 | 0.0592 | 0.1344 | 0.0677 | _0.0509_ | 0.0782 | 0.0913 | 0.0682 | 0.0595 | **0.0445** | 0.0296 |
|  | MAE | 0.5519 | 0.4420 | 0.2255 | 0.2089 | 0.2048 | 0.1846 | 0.1959 | 0.2931 | 0.2073 | _0.1805_ | 0.2252 | 0.2451 | 0.2088 | 0.1951 | **0.1686** | 0.1402 |

Table 9: MSE and MAE results for multivariate datasets (Forecasting Horizons: $H = 96$)

| Dataset | Metric | Autoformer | CATS | DLinear | N-BEATS | N-HiTS | PaiFilter | PatchTST | SegRNN | TSMixer | TexFilter | TimeKAN | TimeLLM | TimeMixer | TimesNet | iTransformer |
|---|---|---|---|---|---|---|---|---|---|---|---|---|---|---|---|---|
| Ad-Sales | MSE | 1.4331 | 1.4469 | 1.2836 | 1.2894 | 1.2640 | 1.2681 | 1.2638 | 1.3543 | **1.2483** | 1.2834 | 1.3236 | 1.4684 | _1.2509_ | 1.3167 | 1.3274 |
|  | MAE | 0.8169 | 0.8146 | 0.7306 | 0.7271 | 0.7177 | 0.7194 | 0.7191 | 0.7681 | _0.7069_ | 0.7278 | 0.7486 | 0.8264 | **0.7050** | 0.7400 | 0.7395 |
| Intervention-Effect | MSE | 0.5306 | 0.1444 | 0.1174 | 0.1142 | 0.1017 | 0.1026 | 0.1072 | _0.0994_ | 0.1208 | 0.1064 | 0.1008 | 0.1126 | 0.1012 | **0.0980** | 0.1010 |
|  | MAE | 0.4202 | 0.2652 | 0.2509 | 0.2398 | 0.2339 | 0.2238 | 0.2286 | _0.2214_ | 0.2553 | 0.2287 | 0.2242 | 0.2368 | 0.2227 | **0.2198** | 0.2254 |
| Lotka-Volterra | MSE | 1.8547 | 0.2453 | 0.1660 | 0.0250 | 0.0101 | 0.0929 | 0.1073 | 0.0233 | **0.0068** | 0.0198 | 0.0978 | 0.2453 | _0.0158_ | 0.0404 | 0.0193 |
|  | MAE | 1.0714 | 0.3474 | 0.2982 | 0.1083 | 0.0778 | 0.1653 | 0.2138 | 0.1100 | **0.0652** | 0.0941 | 0.2107 | 0.3288 | _0.0927_ | 0.1105 | 0.0994 |
| Macro-Economy | MSE | 0.7312 | 0.7224 | 0.7519 | 0.7300 | 0.7479 | _0.7086_ | 0.7142 | 0.7104 | 0.7845 | 0.7092 | 0.7123 | 0.7224 | 0.7109 | **0.7022** | 0.7136 |
|  | MAE | 0.6537 | 0.6479 | 0.6728 | 0.6391 | 0.6699 | _0.6286_ | 0.6307 | 0.6291 | 0.7106 | 0.6317 | 0.6337 | 0.6481 | 0.6321 | **0.6260** | 0.6308 |
| SIR-Model | MSE | 0.0198 | 0.0030 | 0.0030 | 0.0024 | 0.0074 | **0.0019** | 0.0019 | 0.0026 | 0.5375 | 0.0020 | **0.0019** | 0.0019 | 0.0019 | 0.0034 | 0.0022 |
|  | MAE | 0.1156 | 0.0421 | 0.0407 | 0.0377 | 0.0678 | **0.0328** | 0.0328 | 0.0408 | 0.6662 | 0.0342 | 0.0333 | 0.0333 | 0.0330 | 0.0464 | 0.0361 |
| Supply-Demand-Price | MSE | 5.1740 | 0.2656 | 0.1594 | 0.1145 | 8.3423 | **0.0601** | 0.1346 | 5.3351 | 1.6242 | _0.1181_ | 0.1364 | 0.5075 | 0.2176 | 0.2048 | 0.3853 |
|  | MAE | 1.6923 | 0.3856 | 0.2762 | 0.2497 | 1.5176 | **0.1573** | 0.2607 | 1.5233 | 0.9403 | _0.2554_ | 0.2567 | 0.4820 | 0.3137 | 0.3288 | 0.3986 |
| Weather-Sales | MSE | 0.8457 | 0.7810 | 0.5425 | 0.7739 | 0.5326 | 0.6510 | 0.7205 | 0.6514 | **0.5084** | 0.5678 | 0.6791 | 0.7643 | 0.5285 | _0.5203_ | 0.5476 |
|  | MAE | 0.7195 | 0.6561 | 0.5301 | 0.6450 | 0.5224 | 0.5802 | 0.6260 | 0.5805 | **0.5094** | 0.5547 | 0.5998 | 0.6514 | 0.5243 | _0.5211_ | 0.5294 |

To evaluate the capacity of time series models to handle complex temporal patterns found in real-world applications, we generated synthetic datasets that simulate various real-world phenomena while maintaining controlled generation processes. These datasets include simulations of economic indicators, electricity consumption patterns, industrial sensor measurements, network traffic, stock prices, temperature readings, and website traffic for univariate testing, as well as advertising-sales relationships, intervention effects, Lotka-Volterra dynamics, macroeconomic variables, epidemiological models, supply-demand-price relationships, and weather-sales correlations for multivariate scenarios.

As shown in Table 8, filter-based models excel with sensor-related data, with PaiFilter achieving best performance on Industrial-Sensor-Normal and Temperature-Sensor datasets. TimeKAN demonstrates superior performance with economic and financial time series, leading in Stock-Price and Retail-Sales forecasting. The results in Table 9 reveal that channel-dependent models (such as TSMixer, TimesNet, and TimeMixer) generally perform well on complex multivariate datasets, which aligns with our previous findings from the cross-variable learning experiments. This suggests that models designed to leverage relationships between variables have an inherent advantage when handling multivariate time series with complex interdependencies and feedback mechanisms. A particularly notable observation is that channel-dependent models do not consistently outperform channel-independent models across all multivariate datasets. For instance, while TSMixer (channel-dependent) excels on Weather-Sales and Lotka-Volterra datasets, PaiFilter (channel-independent) achieves the best performance on Supply-Demand-Price and SIR-Model datasets. This suggests that when variable relationships become more complex and nuanced, channel-dependent models may sometimes struggle to effectively learn these patterns, occasionally performing worse than channel-independent models that treat each variable separately. These results also highlight that no single architecture dominates across all patterns, with performance advantages highly dependent on the specific temporal characteristics of the target domain.

### B.3 Evaluating Zero-Shot Capabilities of Time Series Foundation Models

Table 10: Zero-Shot Forecasting Results by Pattern Type.

| Pattern Type | Metric | Model | | |
|---|---|---|---|---|
| | | Chronos | Moirai | TimeMoE |
| Trend Patterns | MSE | **0.0002** | 0.9332 | 30.9859 |
| | MAE | **0.0047** | 0.3306 | 1.5138 |
| Periodic Patterns | MSE | 0.5293 | 1.3476 | **0.0594** |
| | MAE | 0.4885 | 0.9125 | **0.1353** |
| Complex Univariate | MSE | **0.4137** | 0.6437 | 0.5269 |
| | MAE | **0.3925** | 0.5707 | 0.4716 |
| Complex Multivariate | MSE | **0.7800** | 1.8564 | 2.0845 |
| | MAE | **0.5066** | 0.7266 | 0.7574 |

Our evaluation of zero-shot forecasting capabilities reveals distinct pattern specialization among time series foundation models. As shown in Figure 14 and Table 10, each model demonstrates unique strengths aligned with specific temporal patterns.

Chronos excels at trend forecasting with near-perfect accuracy (MSE: 0.0002), capturing growth patterns without fine-tuning. Conversely, TimeMoE demonstrates exceptional capabilities in modeling periodic patterns (MSE: 0.0594), precisely predicting oscillatory behaviors while struggling significantly with trends. Moirai shows moderate but consistent performance across pattern types without particular specialization.

For complex datasets containing mixed patterns, Chronos maintains the best overall performance on both univariate and multivariate data, showing that Chronos' approach of using Gaussian processes to generate synthetic data effectively enhances its generalization capabilities.

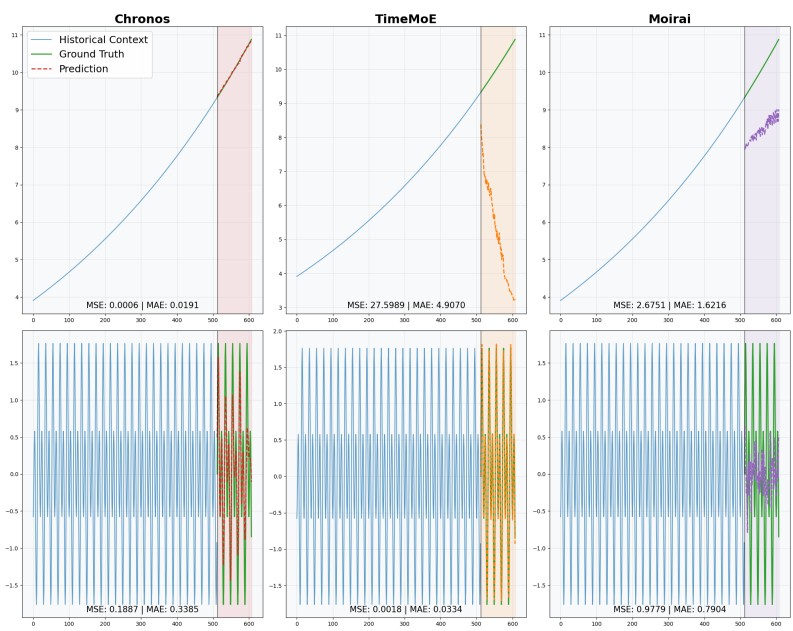

Figure 14: Zero-shot forecasting comparison between Chronos, TimeMoE, and Moirai models on different data types. The top row shows trend forecasting performance, while the bottom row displays periodic pattern forecasting capabilities.

## B.4  Limitations of Normalization Layer

To investigate how different architectures handle long-range dependencies and trend transitions, particularly the impact of normalization layers, we designed two challenging synthetic time series patterns, as shown in Figure 15. These patterns were specifically chosen because their numerical values carry critical predictive information—a characteristic that could expose potential limitations of normalization techniques. The first is a triangle wave with an extended period of 400 time steps, where reaching values of 1 or -1 indicates an imminent trend reversal. The second is a pulse pattern with spikes appearing at regular intervals of 96 time steps, requiring models to accurately predict both the timing and amplitude of subsequent pulses.

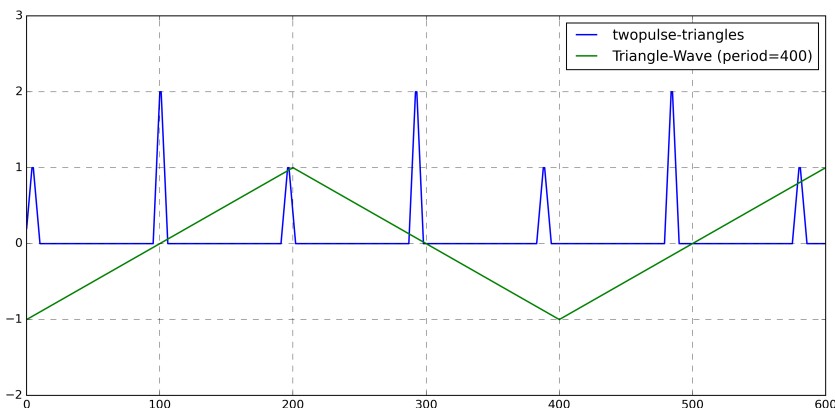

Figure 15: Synthetic time series patterns designed to test long-range dependencies: a triangle wave with period 400 (green) and a sequence with alternating pulses at regular intervals of 96 time steps (blue).

For both patterns, we used an input window of 96 time steps and asked models to predict the next 96 time steps. This experimental setup poses two specific challenges:

1. **Long-range dependency recognition:** For the pulse pattern, models must recognize that after observing one pulse, another will appear exactly 96 steps later, potentially with a different amplitude.

2. **Trend reversal prediction:** For the triangle wave, models must predict when the upward or downward trend will reverse, despite having seen only a fraction of the full 400-step cycle. This reversal is signaled by the series approaching its extreme values of 1 or -1.

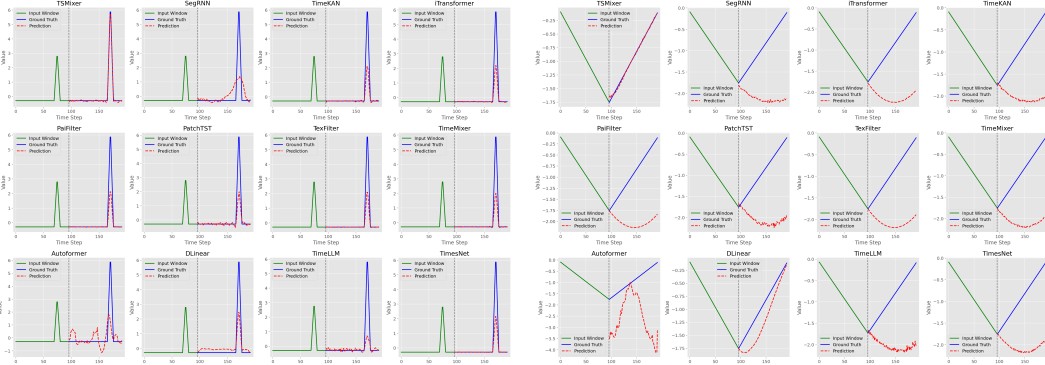

Figure 16: Model predictions on the pulse pattern. Each subplot shows a model's prediction (red dashed line) compared to ground truth (blue solid line) and input window (green solid line). TSMixer accurately predicts the correct pulse height, while most batch-normalized models struggle to preserve magnitude information.

Figure 17: Model predictions on the triangle wave pattern. Each subplot shows a model's prediction (red dashed line) compared to ground truth (blue solid line) and input window (green solid line). Models with batch normalization struggle to predict the correct trend reversal point.

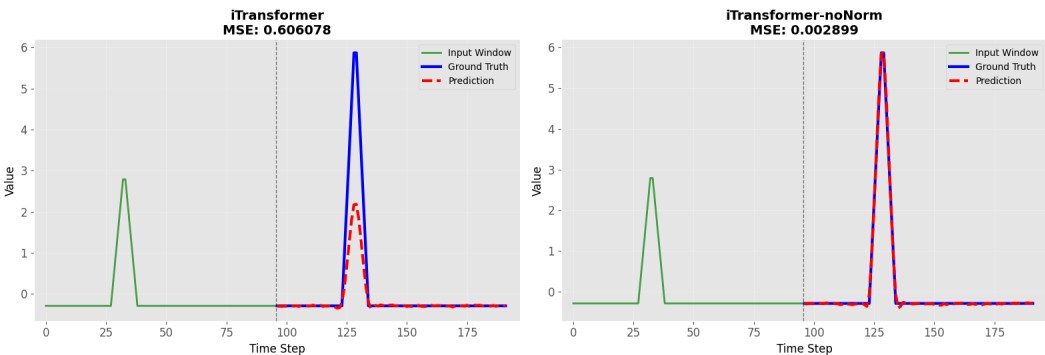

Figure 18: Left: iTransformer with normalization layers. Right: iTransformer-noNorm without normalization layers.

Our experimental results, visualized in Figures 16 and 17, reveal a striking pattern: among all tested models, only TSMixer demonstrates superior performance. Notably, of all models tested, only Seg-RNN, DLinear, and TSMixer do not employ normalization layers, with TSMixer showing the most consistent accuracy.

In the pulse pattern forecasts (Figure 16), TSMixer correctly predicts both the timing and magnitude of the upcoming pulse. In contrast, models with normalization layers like Autoformer show erratic behavior, while others like PaiFilter, TexFilter, and TimeMixer predict pulses with significantly reduced amplitudes. This suggests that normalization is stripping away critical amplitude information.

For the triangle wave pattern (Figure 17), the impact of normalization is even more evident. TSMixer accurately follows the downward trend and correctly predicts the exact point where the trend reverses, perfectly aligning with the ground truth throughout the entire prediction window. Models

using normalization layers largely fail to identify the correct trend reversal point, with most predicting either a continued downward trajectory or an incorrect reversal pattern.

To further validate our hypothesis about normalization layers, we conducted a comparative experiment with iTransformer, creating a variant (iTransformer-noNorm) with all normalization layers removed. As shown in Figure 18, the normalization-free variant dramatically outperforms the original model on these patterns, confirming that normalization layers are indeed the limiting factor.

These findings highlight a fundamental limitation of normalization in time series forecasting: the normalization process inherently discards critical magnitude information. When normalization standardizes features to have zero mean and unit variance, it effectively removes the absolute scale information that is often crucial for time series patterns. While this standardization is beneficial for stable training and handling distribution shifts in many deep learning applications, it proves detrimental when absolute magnitude carries important predictive information.

### B.5 Experimental Supplementary Content

This section provides additional experimental results and visualizations that complement the main findings presented in the paper. We include detailed model performance comparisons across different pattern types, cross-variable relationship analysis, and comprehensive anomaly impact assessment.

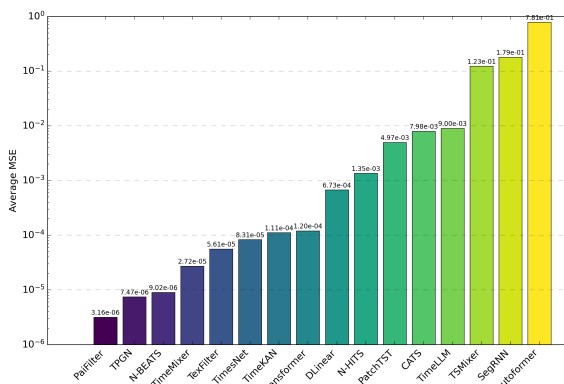

Figure 19: Average MSE comparison for different models on trend data.

Figure 20: Radar chart visualizing the prediction difficulty of different trend patterns. Lower values indicate more challenging patterns to predict.

Figure 19 illustrates model performance on trend data, where PaiFilter consistently achieves the lowest error rates, demonstrating superior capability in capturing trend relationships compared to other architectures. Figure 20 presents a radar chart of trend pattern difficulty. To construct this visualization, we first calculated the average MSE for each signal type across all models, then applied logarithmic transformation (log10) to compress the range of values. The transformed values were then normalized to a 0-1 scale using the formula: $normalized\_value = 1 - (val - min\_val)/(max\_val - min\_val)$, where lower values in the radar chart indicate more challenging patterns. This inverse mapping ensures that patterns with higher average prediction error (more difficult) appear closer to the center of the radar chart. The visualization reveals that patterns with accelerating growth rates, specifically Quadratic-Function and Exponential-Trend, pose the greatest challenge for forecasting models, as their rapid rate changes become increasingly difficult to predict accurately over longer horizons.

Figure 21 presents model performance on periodic data, where N-HiTS, TimesNet and DLinear demonstrate exceptional capabilities in modeling oscillatory patterns. The difficulty radar chart in Figure 22 was calculated using the same methodology as for trend patterns: first computing average MSE across all models for each pattern type, applying logarithmic transformation to compress the range, and finally normalizing and inverting the values to create the 0-1 scale where lower values indicate more challenging patterns. The radar chart clearly indicates that complex multi-frequency patterns (Ten-Sin) and discontinuous patterns (Square-Wave) present the greatest challenges. Square

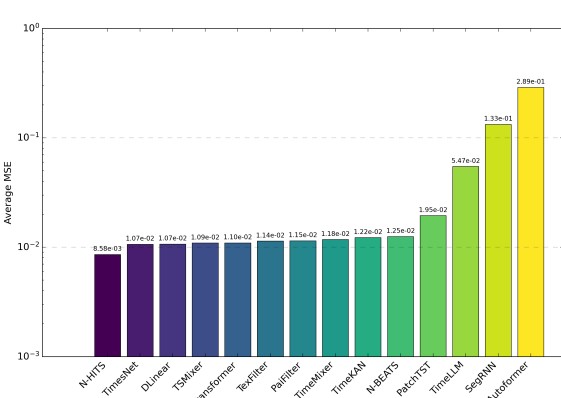

Figure 21: Average MSE comparison for different models on periodic data.

Figure 22: Radar chart visualizing the prediction difficulty of different periodic patterns. Lower values indicate more challenging patterns to predict.

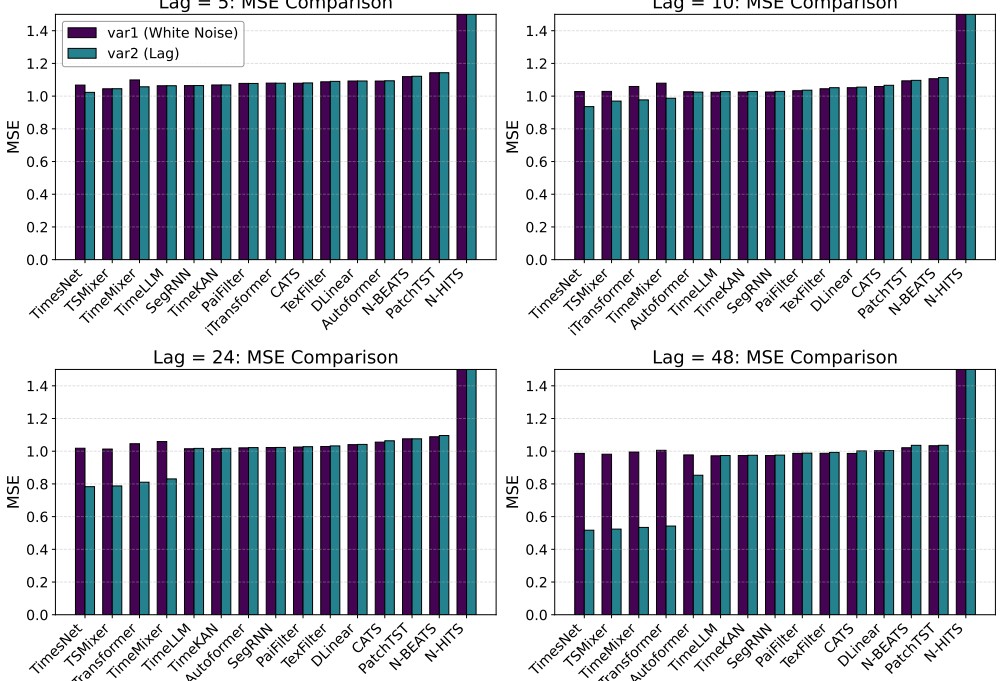

Figure 23: MSE comparison for lag relationship detection with different lag values (5, 10, 24, and 48). Blue bars represent MSE for white noise prediction (var1), while red bars represent MSE for the lagged variable prediction (var2). Lower MSE for the lagged variable indicates better ability to detect temporal relationships.

waves, with their abrupt value transitions, are particularly difficult for models to predict accurately, as most architectures struggle to capture these discontinuities without introducing smoothing effects or oscillatory artifacts.

Figure 23 illustrates model performance in detecting temporal lag relationships, with both input window and forecasting horizon set to 96 time steps. The experiment involved predicting two variables: var1 (random noise) and var2 (var1 lagged by different time steps). As the lag value increases from 5 to 48, channel-dependent architectures (TimesNet, TSMixer, TimeMixer, and iTransformer)

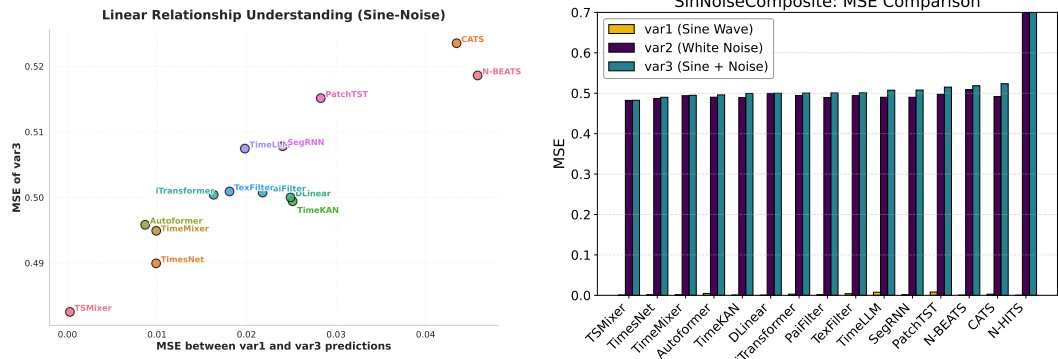

Figure 24: Test results on the Sine-Noise dataset for evaluating models' ability to capture relationships between variables. This dataset consists of var1 (sine wave), var2 (white noise), and var3 (sum of var1 and var2). Left: Scatter plot showing prediction performance for understanding linear relationships, where x-axis represents MSE between var1 and var3 predictions, while y-axis shows MSE of var3 predictions vs ground truth. Models in the bottom-left corner better understand the additive relationship between variables. Right: Per-model MSE values for the three variables, where lower values for var1 and var3 compared to var2 indicate the model effectively captures the relationship between the sine component and the composite signal.

show progressively decreasing MSE values for var2, indicating these models successfully capture the temporal relationships despite increasing lag distances. This capability reveals their effectiveness in leveraging cross-channel information to identify and model temporal dependencies between variables.

Figure 24 evaluates models on the Sine-Noise dataset, which comprises three variables: var1 (sine wave), var2 (white noise), and var3 (sum of var1 and var2). This dataset is specifically designed to test models' ability to learn relationships between variables. If models predict var3 independently (treating each channel separately), they will be affected by the noise component from var2, resulting in higher prediction errors. However, if models successfully learn the additive relationship var3 = var1 + var2, they can recognize that the true signal component of var3 is simply var1, since var2 is unpredictable noise. In this case, models should achieve low MSE for var3 predictions by effectively filtering out the noise. Since var2 is white noise, models typically predict its value as 0, and consequently, if the relationship is properly learned, the predicted values for var3 should closely match those for var1. The left scatter plot reveals a positive correlation between MSE of var3 predictions and MSE between predicted var1 and var3. Notably, TSMixer achieves near-zero MSE between its var1 and var3 predictions, resulting in the lowest overall var3 prediction error. The right bar chart confirms TSMixer's superior performance across all three variables, demonstrating its exceptional ability to capture and leverage the additive relationship between variables.

Table 11 presents model performance under various anomaly conditions, including point anomalies (isolated outliers) at different densities (0.5% to 10%), pulse anomalies (sustained deviations) with varying frequencies (1, 3, or 5 pulses), mean shifts (abrupt changes in the baseline level at different magnitudes), and trend shifts (sudden changes in the growth rate or direction). The results demonstrate significant variance in model robustness against data irregularities. TimeKAN and TimeMixer exhibit strong resilience against point anomalies, maintaining consistent performance as anomaly density increases. For pulse anomalies, TimesNet and TexFilter perform best with fewer pulses, while PaiFilter and TimeKAN demonstrate superior handling of cases with multiple pulses. Notably, TSMixer shows considerable performance degradation as anomaly density increases, highlighting a vulnerability that could be critical in real-world applications where data quality is variable.

Table 12 presents a systematic evaluation of model robustness under various noise distributions beyond the standard Gaussian assumption. The results reveal distinct performance patterns across different noise characteristics. For heavy-tailed noise distributions (t-distribution with df=3 and Lévy with $\alpha$=1.2, 1.8), PaiFilter consistently achieves the best performance, demonstrating superior robustness to extreme outliers. Under more moderate noise conditions (uniform and Laplace distributions), FilterNet-based models (PaiFilter and TexFilter) maintain their advantage alongside Times-

Table 11: Comprehensive Comparison of MSE and MAE for Different Anomaly Types and Scenarios (Forecasting Horizons: $H = 96$)

| Anomaly Type | Metric | Model | | | | | | | | | | | | | | |
|---|---|---|---|---|---|---|---|---|---|---|---|---|---|---|---|---|
| | | Autoformer | CATS | DLinear | N-BEATS | N-HiTS | PaiFilter | PatchTST | SegRNN | TSMixer | TexFilter | TimeKAN | TimeLLM | TimeMixer | TimesNet | iTransformer |
| No Anomaly | MSE | 0.0935 | 0.0231 | 0.0390 | 0.0225 | 0.0410 | 0.0205 | 0.0257 | 0.0863 | 0.0277 | 0.0207 | 0.0207 | 0.0447 | 0.0206 | 0.0203 | 0.0209 |
| | MAE | 0.1871 | 0.1169 | 0.1392 | 0.1159 | 0.1400 | 0.1086 | 0.1260 | 0.1866 | 0.1227 | 0.1091 | 0.1097 | 0.1596 | 0.1094 | 0.1087 | 0.1098 |
| point-anomaly-0.5pct | MSE | 0.1619 | 0.0297 | 0.0280 | 0.0216 | 0.1908 | 0.0246 | 0.0300 | 0.0914 | 0.1344 | 0.0238 | 0.0204 | 0.0416 | 0.0204 | 0.0238 | 0.0254 |
| | MAE | 0.2544 | 0.1354 | 0.1254 | 0.1108 | 0.2498 | 0.1197 | 0.1361 | 0.1919 | 0.2700 | 0.1157 | 0.1079 | 0.1585 | 0.1078 | 0.1174 | 0.1216 |
| point-anomaly-1pct | MSE | 0.1795 | 0.0344 | 0.0209 | 0.0207 | 0.2174 | 0.0210 | 0.0233 | 0.0841 | 0.0663 | 0.0222 | 0.0186 | 0.0383 | 0.0175 | 0.0220 | 0.0223 |
| | MAE | 0.2681 | 0.1466 | 0.1073 | 0.1061 | 0.2969 | 0.1092 | 0.1185 | 0.1810 | 0.1873 | 0.1142 | 0.1041 | 0.1470 | 0.1007 | 0.1136 | 0.1150 |
| point-anomaly-5pct | MSE | 0.1090 | 0.0710 | 0.0198 | 0.0195 | 0.3430 | 0.0214 | 0.0304 | 0.0766 | 0.1519 | 0.0309 | 0.0143 | 0.0692 | 0.0193 | 0.0322 | 0.0293 |
| | MAE | 0.2221 | 0.1946 | 0.1053 | 0.1024 | 0.4281 | 0.1078 | 0.1329 | 0.1803 | 0.2805 | 0.1345 | 0.0846 | 0.1927 | 0.1024 | 0.1390 | 0.1301 |
| point-anomaly-10pct | MSE | 0.0726 | 0.0672 | 0.0579 | 0.0191 | 0.4161 | 0.0209 | 0.0280 | 0.0669 | 0.2046 | 0.0314 | 0.0163 | 0.0793 | 0.0206 | 0.0542 | 0.0361 |
| | MAE | 0.1897 | 0.1879 | 0.1419 | 0.0998 | 0.4963 | 0.1064 | 0.1265 | 0.1787 | 0.2833 | 0.1351 | 0.0950 | 0.1964 | 0.1060 | 0.1686 | 0.1436 |
| pulse-anomaly-1 | MSE | 0.2019 | 0.0210 | 0.0625 | 0.0201 | 0.0366 | 0.0240 | 0.0303 | 0.0991 | 0.0540 | 0.0212 | 0.0211 | 0.0834 | 0.0260 | 0.0198 | 0.0219 |
| | MAE | 0.2775 | 0.1125 | 0.1872 | 0.1099 | 0.1294 | 0.1109 | 0.1306 | 0.1986 | 0.1673 | 0.1093 | 0.1107 | 0.1916 | 0.1143 | 0.1067 | 0.1125 |
| pulse-anomaly-3 | MSE | 0.1271 | 0.0344 | 0.1513 | 0.0208 | 42.4089 | 0.0226 | 0.0381 | 0.1078 | 0.2714 | 0.0213 | 0.0358 | 0.0489 | 0.0255 | 0.0285 | 0.0387 |
| | MAE | 0.2371 | 0.1364 | 0.2501 | 0.1147 | 2.9509 | 0.0926 | 0.1139 | 0.2062 | 0.3640 | 0.0931 | 0.0945 | 0.1470 | 0.0959 | 0.0966 | 0.1068 |
| pulse-anomaly-5 | MSE | 0.1271 | 0.0692 | 0.1169 | 0.0195 | 3.8999 | 0.0145 | 0.0218 | 0.0994 | 0.1764 | 0.0221 | 0.0137 | 0.0368 | 0.0160 | 0.0180 | 0.0189 |
| | MAE | 0.2266 | 0.1772 | 0.2655 | 0.1024 | 0.8587 | 0.0950 | 0.1158 | 0.2066 | 0.2968 | 0.1085 | 0.0921 | 0.1380 | 0.0989 | 0.1065 | 0.1071 |
| mean-shift-1 | MSE | 0.1817 | 0.0205 | 0.0414 | 0.0208 | 0.0215 | 0.0190 | 0.0247 | 0.0904 | 0.0304 | 0.0197 | 0.0202 | 0.0268 | 0.0205 | 0.0187 | 0.0186 |
| | MAE | 0.2390 | 0.1118 | 0.1491 | 0.1122 | 0.1115 | 0.1064 | 0.1250 | 0.1877 | 0.1299 | 0.1082 | 0.1099 | 0.1294 | 0.1061 | 0.1061 | 0.1057 |
| mean-shift-3 | MSE | 0.1125 | 0.0227 | 0.0469 | 0.0212 | 0.0227 | 0.0194 | 0.0260 | 0.0891 | 0.0484 | 0.0200 | 0.0209 | 0.0295 | 0.0194 | 0.0206 | 0.0206 |
| | MAE | 0.1993 | 0.1145 | 0.1599 | 0.1119 | 0.1141 | 0.1066 | 0.1273 | 0.1880 | 0.1579 | 0.1080 | 0.1102 | 0.1338 | 0.1076 | 0.1094 | 0.1106 |
| trend-shift-1 | MSE | 0.1823 | 0.0233 | 0.0500 | 0.0221 | 0.0642 | 0.0210 | 0.0261 | 0.0878 | 0.0378 | 0.0212 | 0.0212 | 0.0306 | 0.0212 | 0.0207 | 0.0214 |
| | MAE | 0.2432 | 0.1167 | 0.1607 | 0.1145 | 0.1876 | 0.1107 | 0.1270 | 0.1883 | 0.1488 | 0.1113 | 0.1118 | 0.1391 | 0.1117 | 0.1105 | 0.1119 |
| trend-shift-2 | MSE | 0.1578 | 0.0382 | 0.0413 | 0.0363 | 0.0622 | 0.0349 | 0.0400 | 0.0998 | 0.0485 | 0.0353 | 0.0351 | 0.0388 | 0.0356 | 0.0339 | 0.0350 |
| | MAE | 0.2443 | 0.1385 | 0.1454 | 0.1359 | 0.1584 | 0.1314 | 0.1475 | 0.2066 | 0.1535 | 0.1320 | 0.1328 | 0.1461 | 0.1333 | 0.1304 | 0.1321 |

Table 12: Comparison of MSE and MAE for Different Noise Types (Forecasting Horizons: $H = 96$)

| Noise Type | Metric | Model | | | | | | | | | | | | | | |
|---|---|---|---|---|---|---|---|---|---|---|---|---|---|---|---|---|
| | | Autoformer | CATS | DLinear | N-BEATS | N-HiTS | PaiFilter | PatchTST | SegRNN | TSMixer | TexFilter | TimeKAN | TimeLLM | TimeMixer | TimesNet | iTransformer |
| Uniform Noise | MSE | 0.1352 | 0.0227 | 0.0401 | 0.0216 | 0.0327 | 0.0198 | 0.0261 | 0.0864 | 0.0509 | 0.0202 | 0.0205 | 0.0267 | 0.0202 | 0.0199 | 0.0202 |
| | MAE | 0.2136 | 0.1207 | 0.1486 | 0.1189 | 0.1349 | 0.1147 | 0.1311 | 0.1915 | 0.1566 | 0.1155 | 0.1162 | 0.1323 | 0.1159 | 0.1150 | 0.1155 |
| Laplace Noise | MSE | 0.1107 | 0.0223 | 0.0276 | 0.0233 | 0.0355 | 0.0204 | 0.0253 | 0.0870 | 0.0276 | 0.0201 | 0.0205 | 0.0267 | 0.0208 | 0.0204 | 0.0208 |
| | MAE | 0.2011 | 0.1067 | 0.1207 | 0.1097 | 0.1316 | 0.0999 | 0.1191 | 0.1801 | 0.1145 | 0.0996 | 0.1015 | 0.1230 | 0.1015 | 0.1004 | 0.1017 |
| t-distribution (df=3) | MSE | 0.2257 | 0.0206 | 0.0385 | 0.0199 | 0.0284 | 0.0178 | 0.0227 | 0.0841 | 0.0275 | 0.0184 | 0.0179 | 0.0275 | 0.0183 | 0.0179 | 0.0183 |
| | MAE | 0.2669 | 0.1002 | 0.1350 | 0.0975 | 0.1119 | 0.0902 | 0.1105 | 0.1702 | 0.1170 | 0.0921 | 0.0912 | 0.1236 | 0.0925 | 0.0906 | 0.0920 |
| t-distribution (df=10) | MSE | 0.2072 | 0.0223 | 0.0274 | 0.0211 | 0.0432 | 0.0199 | 0.0251 | 0.0864 | 0.0282 | 0.0202 | 0.0200 | 0.0282 | 0.0200 | 0.0198 | 0.0199 |
| | MAE | 0.2578 | 0.1121 | 0.1249 | 0.1095 | 0.1397 | 0.1048 | 0.1219 | 0.1832 | 0.1207 | 0.1055 | 0.1059 | 0.1301 | 0.1055 | 0.1046 | 0.1051 |
| Lévy ($\alpha$=1.2) | MSE | 0.0942 | 0.0108 | 0.0181 | 0.0101 | 0.0123 | 0.0099 | 0.0145 | 0.0754 | 0.0293 | 0.0100 | 0.0099 | 0.0154 | 0.0099 | 0.0099 | 0.0100 |
| | MAE | 0.1346 | 0.0365 | 0.0688 | 0.0266 | 0.0411 | 0.0235 | 0.0586 | 0.1144 | 0.0822 | 0.0243 | 0.0270 | 0.0659 | 0.0266 | 0.0250 | 0.0264 |
| Lévy ($\alpha$=1.8) | MSE | 0.1911 | 0.0036 | 0.0069 | 0.0018 | 0.0058 | 0.0016 | 0.0062 | 0.0673 | 0.0041 | 0.0017 | 0.0017 | 0.0084 | 0.0017 | 0.0018 | 0.0019 |
| | MAE | 0.2293 | 0.0408 | 0.0488 | 0.0233 | 0.0400 | 0.0206 | 0.0556 | 0.1164 | 0.0352 | 0.0213 | 0.0239 | 0.0637 | 0.0233 | 0.0225 | 0.0240 |

Table 13: MSE Results on Real-World Datasets. All results are averaged from four different forecasting horizons: $H \in \{24, 48, 96, 192\}$. Avg (Synthetic) is calculated as the average MSE across all Complex Real-world Pattern Simulations datasets (Tables 8 and 9) to provide a comprehensive comparison between synthetic and real-world performance.

| Dataset | Autoformer | CATS | DLinear | N-BEATS | PaiFilter | PatchTST | SegRNN | TSMixer | TexFilter | TimeKAN | TimeMixer | TimesNet | iTransformer |
|---|---|---|---|---|---|---|---|---|---|---|---|---|---|
| ETTh1 | 0.5792 | 0.4945 | 0.5122 | 0.4845 | 0.4916 | 0.4952 | 0.5270 | 0.5656 | 0.4997 | 0.4795 | 0.4858 | 0.5425 | 0.5050 |
| ETTh2 | 0.4569 | 0.4030 | 0.4228 | 0.3978 | 0.3991 | 0.3929 | 0.3939 | 1.7774 | 0.4032 | 0.3884 | 0.3904 | 0.4112 | 0.4061 |
| ETTm1 | 0.5971 | 0.4393 | 0.4545 | 0.4217 | 0.4178 | 0.4310 | 0.4583 | 0.5303 | 0.4396 | 0.4228 | 0.4216 | 0.4771 | 0.4404 |
| ETTm2 | 0.3229 | 0.3018 | 0.3277 | 0.2866 | 0.2828 | 0.2894 | 0.2834 | 0.4727 | 0.2881 | 0.2810 | 0.2799 | 0.2933 | 0.3026 |
| Electricity | 0.3555 | 0.3311 | 0.3141 | 0.2866 | 0.2771 | 0.2949 | 0.3471 | 0.3023 | 0.2572 | 0.3024 | 0.2701 | 0.2983 | 0.2607 |
| Exchange-Rate | 0.3040 | 0.1869 | 0.1765 | 0.1830 | 0.1868 | 0.1774 | 0.1817 | 0.4876 | 0.1816 | 0.1755 | 0.1805 | 0.2192 | 0.1903 |
| Traffic | 0.7140 | 0.6277 | 0.7335 | 0.5862 | 0.5779 | 0.5744 | 0.8679 | 0.6276 | 0.5201 | 0.6903 | 0.5780 | 0.6928 | 0.5155 |
| Weather | 0.4565 | 0.2885 | 0.3060 | 0.2705 | 0.2691 | 0.2761 | 0.2654 | 0.2805 | 0.2700 | 0.2688 | 0.2679 | 0.2862 | 0.3107 |
| Avg (Real-World) | 0.4733 | 0.3841 | 0.4059 | 0.3646 | 0.3628 | 0.3664 | 0.4156 | 0.6305 | 0.3575 | 0.3761 | 0.3593 | 0.4026 | 0.3664 |
| Rank (Real-World) | 12 | 8 | 10 | 4 | 3 | 5 | 11 | 13 | 1 | 7 | 2 | 9 | 6 |
| Avg (Synthetic) | 0.8956 | 0.3775 | 0.3379 | 0.3137 | 0.3004 | 0.3119 | 0.6951 | 0.4903 | 0.2940 | 0.3081 | 0.2956 | 0.3016 | 0.3165 |
| Rank (Synthetic) | 13 | 10 | 9 | 7 | 3 | 6 | 12 | 11 | 1 | 5 | 2 | 4 | 8 |

Net. Notably, filter-based architectures show remarkable stability across all noise types, suggesting their frequency-domain processing provides inherent resilience to diverse noise characteristics.

## B.6 Real World Dataset

To empirically validate the effectiveness of our synthetic benchmark framework, we conducted extensive experiments on eight widely-used real-world time series datasets: ETTh1, ETTh2, ETTm1, ETTm2, Electricity, Exchange-Rate, Traffic, and Weather. These datasets represent diverse domains including energy, finance, transportation, and meteorology, providing a comprehensive testbed for evaluating whether insights gained from our synthetic benchmark translate to practical forecasting scenarios.

Table 13 presents the comprehensive comparison results, where all MSE values are averaged across four different forecasting horizons ($H \in \{24, 48, 96, 192\}$) with a fixed input window of 96 time steps. The results reveal a remarkably strong correlation between model performance on our synthetic datasets and real-world data. Most notably, the top three performing models in our synthetic benchmark—TexFilter (rank 1), TimeMixer (rank 2), and PaiFilter (rank 3)—maintain identical

rankings when evaluated on real-world datasets, with average MSE values of 0.3575, 0.3593, and 0.3628 respectively.

Furthermore, the results expose consistent weaknesses across both evaluation paradigms. TSMixer, which showed vulnerability to anomalies in our synthetic robustness tests, exhibits the poorest average performance (0.6305 MSE, rank 13) on real-world datasets, with particularly catastrophic failure on ETTh2 (1.7774 MSE). Similarly, Autoformer and SegRNN, which struggled with certain synthetic patterns, maintain lower rankings on real-world data.

These findings provide compelling evidence that our synthetic benchmark is not an isolated academic exercise, but rather a reliable and effective tool for predicting real-world model performance. The framework successfully identifies robust, high-performing architectures while exposing fundamental limitations—capabilities that are essential for advancing time series forecasting research. The strong empirical validation demonstrates that insights derived from our controlled synthetic experiments generalize effectively to the complexities of real-world forecasting tasks, thereby establishing the practical value and significance of SynTSBench as a unified benchmark for the time series community.

### B.7 Model Capability Analysis and Architectural Trade-offs

Table 14 summarizes the diverse set of forecasting models evaluated in our benchmark, spanning multiple architectural paradigms from transformers to linear models.

Table 14: Model Overview Table

| Model Name | Framework Type | Key Features |
|---|---|---|
| PatchTST [6] | Transformer | Long sequence forecasting, sub-sequence input |
| TimeLLM [21] | Transformer | Based on LLM, textual original input |
| Autoformer [4] | Transformer | Decomposes Transformer, self-adaptive mechanism |
| iTransformer [12] | Transformer | Inverted Transformer, improved forecasting |
| CATS [20] | Transformer | Cross-attention only |
| TimeMixer [10] | MLP | Multi-dimensional fusion, high efficiency |
| TSMixer [8] | MLP | Fully MLP-based, scalable forecasting |
| N-BEATS [32] | MLP | Neural basis expansion, interpretable decomposition |
| N-HiTS [24] | MLP | Neural hierarchical interpolation, multi-rate sampling |
| TimesNet [11] | CNN | 2D convolution kernel, periodic/trend learning |
| SegRNN [7] | RNN | Segment-based RNN, long sequence forecasting |
| DLinear [5] | Linear Model | Simple linear baseline |
| TimeKAN [9] | KAN | Interpretable, parameterized KAN |
| Paifilter [22] | FilterNet+MLP | Frequency-domain filtering |
| Texfilter [22] | FilterNet+MLP | Frequency-domain transformation |
| Chronos [25] | Transformer | Tokenizes series, probabilistic sampling |
| TimeMoE [26] | Transformer (MoE) | Sparse mixture-of-experts, billion-scale |
| Moirai [27] | Masked Encoder Transformer | Masked-encoder pretraining, multi-patch |

To provide a comprehensive understanding of model capabilities and their practical implications, we synthesize the extensive experimental results from our benchmark into a structured capability analysis. Table 15 summarizes the key strengths and weaknesses of each evaluated model, revealing fundamental trade-offs inherent in different architectural designs.

## C  Detailed Introduction of the Generated Dataset

This section provides a comprehensive overview of the synthetic time series datasets generated within our SynTSBench framework. We systematically introduce the building blocks and methodology behind our dataset generation process, which forms the foundation for rigorous time series forecasting evaluation. The section is organized into five key components: Appendix C.1 presents the fundamental signal types that serve as building blocks for more complex patterns; Appendix C.2 describes how we systematically introduce controlled noise and anomalies to test model robustness; Appendix C.3 explains our approach to evaluating models across varying temporal scales; Appendix C.4 details the generation of complex univariate time series that simulate real-world phenomena; and Appendix C.5 introduces our multivariate datasets with explicit causal relationships

Table 15: Comprehensive Model Capability Analysis

| Model Name | Key Strengths | Main Weaknesses |
|---|---|---|
| DLinear | Strong modeling capability for simple trend and periodic signals. | Performance drops significantly when patterns become complex. |
| PaiFilter & TexFilter | Extremely strong fitting capability for trend and periodic signals; stable performance under noise and anomalies; strong modeling capability on complex datasets. | Limited capability for modeling cross-variable dependencies. |
| TimeMixer | Comprehensive performance with good results in modeling trend/periodic signals, learning multivariate relationships, and complex patterns. | Mediocre performance in modeling short-term dependencies. |
| TimeKAN | Extremely strong robustness to noise and anomalies; outstanding capability in modeling complex signals. | Limited capability for modeling cross-variable dependencies. |
| TSMixer | Strong capability for modeling multivariate relationships. | Poor robustness to anomalies with errors increasing exponentially at high anomaly rates; weak fitting capability for complex patterns. |
| TimesNet | Strong capability for modeling multivariate relationships; outstanding in modeling complex inter-variable dependencies; performs well in noisy scenarios. | Weak in modeling short-range dependencies; performs moderately on some trend signals. |
| SegRNN | Good at modeling short-range dependencies. | Poor performance in modeling long-range dependencies; mediocre performance on trend/periodic signals; limited capability for modeling variable relationships. |
| PatchTST | Good modeling capability for periodic and trend signals; strong performance on complex patterns. | Limited capability for modeling cross-variable relationships; poor capability for modeling short-term dependencies. |
| iTransformer | Comprehensive performance with good results in modeling some trend/periodic signals, learning multivariate relationships, and complex patterns. | Poor performance in modeling short-term dependencies. |
| N-BEATS | Strong baseline performance across various pattern types; interpretable decomposition into trend and seasonality components; stable performance under noise and anomalies. | Limited capability for multivariate modeling. |
| N-HiTS | Hierarchical architecture enables multi-rate temporal modeling; efficient for long-horizon forecasting. | Catastrophic performance under anomalies (worst among all models); prone to overfitting noise; struggles with complex multivariate relationships. |
| CATS | Cross-attention mechanism provides good balance between computational efficiency and performance; | Poor performance in modeling long-range dependencies; relatively mediocre performance in all aspects. |
| Autoformer | Decomposition mechanism provides interpretability. | Poor performance across most evaluation metrics; struggles with both simple and complex patterns; high sensitivity to noise and anomalies. |
| TimeLLM | Potential for leveraging pre-trained language model knowledge. | Mediocre performance in all aspects; poor at modeling cross-variable dependencies; extremely high training costs. |

and interdependencies. Together, these components enable comprehensive evaluation of time series forecasting models across diverse temporal patterns, noise conditions, and complexity levels.

## C.1 Basic Signals

To establish a foundation for systematically evaluating time series forecasting models, we generate a comprehensive set of synthetic signals that simulate real-world temporal patterns. Table 16 presents the basic building blocks used in our framework **SynTSBench**,ategorized into three fundamental classes: trend functions, periodic functions, and other signal types. Trend functions simulate directional movements commonly observed in economic growth, population changes, and technology adoption curves. Periodic functions represent cyclical behaviors found in seasonal fluctuations, biological rhythms, and industrial processes. The third category encompasses stochastic processes

and composite signals that capture the complexity of financial markets, sensor readings, and natural phenomena. By generating controlled combinations of these signals with programmable parameters, we create synthetic datasets that isolate specific temporal features, enabling us to precisely map model capabilities to pattern types and establish theoretical performance boundaries for rigorous evaluation.

## C.2  Signals with Noise and Anomalies

To systematically evaluate model robustness against common data irregularities, we generated synthetic datasets incorporating controlled noise and anomalies. We selected representative signals from both trend patterns (linear trend, logistic trend) and periodic patterns (sin0.05, Double-Sin, five-Sin) as base signals, and applied varying levels of Gaussian noise (30dB, 20dB, 10dB, 0dB, -10dB, and no noise). This approach enables quantitative assessment of model performance degradation across the noise spectrum.

Beyond Gaussian noise, we also evaluated model robustness under diverse noise distributions to comprehensively assess their resilience to different types of disturbances commonly encountered in real-world scenarios. These include: (1) **Uniform noise**, representing measurement errors with bounded uncertainty; (2) **Laplace noise**, modeling disturbances with heavier tails than Gaussian noise; (3) **t-distribution noise** with different degrees of freedom (df=3 for heavy-tailed, df=10 for moderate-tailed), simulating outlier-prone environments; and (4) **Lévy stable distributions** with stability parameters $\alpha$=1.2 and $\alpha$=1.8, representing extreme heavy-tailed noise characteristic of financial markets and network traffic. These diverse noise types enable evaluation of model performance under various realistic noise characteristics beyond the idealized Gaussian assumption.

For anomaly testing, we injected multiple types of irregularities to simulate different real-world data quality issues. The anomaly types include: (1) **Point anomalies** (random spikes) at varying densities (0.5%, 1%, 5%, 10%), representing isolated outliers from sensor malfunctions or data entry errors; (2) **Pulse anomalies** (sustained deviations with 1, 3, or 5 pulses), simulating temporary system disruptions or external interventions; (3) **Mean shifts** at different magnitudes (shift-1, shift-3), modeling abrupt changes in baseline levels due to policy changes or market regime shifts; and (4) **Trend shifts** at different severities (shift-1, shift-2), representing sudden changes in growth rates or directional patterns. All anomalies were injected on top of signals with 20dB noise and were only introduced in the first 80% of each time series, keeping the test portion (last 20%) anomaly-free. This design enables direct comparison of forecasting performance across different anomaly severities, revealing each model's sensitivity to historical anomalies and recovery capabilities. For visualization examples of these synthetic datasets, refer to Figure 25 for noise levels and Figure 26 for anomaly types. Comprehensive evaluation results across all noise types and anomaly scenarios are presented in Table 12 and Table 11.

## C.3  Datasets with Different Length

For evaluating model performance across varying time series lengths, we generated synthetic datasets with lengths of 1000, 5000, 10000, and 20000 time steps. We selected six representative signal patterns: three trend functions ("Linear-Trend", "Quadratic-Function", "Exponential-Trend") and three periodic functions ("Sin0.005", "Double-Sin", "Five-Sin"). Each signal type was generated with four distinct noise levels (SNR of 20dB, 0dB, -10dB, and no noise) to provide a comprehensive evaluation framework. This dataset design enables quantitative analysis of each model's ability to handle increasing historical context and their performance scaling properties across different temporal patterns and data volumes.

Table 16: Trend, Periodic, and Other Signals for Time Series Analysis

| Function Name | Mathematical Formula | Time-oriented Scenario & Domain Justification |
|---|---|---|
| **Trend Functions** | | |
| Linear | $y = at + b$ | Constant velocity motion; cumulative costs with fixed unit price. |
| Quadratic | $y = at^2 + bt + c$ | Position under constant acceleration ($s = v_0 t + \frac{1}{2}at^2$); cumulative production cost with accelerating learning. |
| Exponential | $y = ae^{bt}$ | Compound interest, inflation, early epidemic spread ($SIR : I(t) \approx e^{bt}$); capacitor discharge $i(t) = I_0 e^{-t/RC}$. |
| Logarithmic | $y = a\ln(t) + b$ | Wright learning curve; software bug-discovery rate decay; Weber-Fechner perception law. |
| Logistic | $y = \dfrac{L}{1 + e^{-k(t-t_0)}}$ | Resource-limited growth: new-product adoption, biological populations, rumor spreading. |
| Gompertz | $y = ae^{-be^{-kt}}$ | Longitudinal tumor volume growth; actuarial mortality curves; technology-substitution saturation. |
| Power Law | $y = at^b$ | DLA cluster radius $r(t) \propto t^{1/d}$; Heaps law in evolving networks $n_u \propto t^b$; Omori-Utsu aftershock decay $n(t) \propto t^{-p}$. |
| Step Function | $y = \begin{cases} c & t < t_0 \\ d & t \geq t_0 \end{cases}$ | Tick-level price jumps; control-system set-point changes; AWS Step Functions state transitions. |
| Piecewise Linear | $y = \begin{cases} a_1 t + b_1 & t < t_1 \\ a_2 t + b_2 & t_1 \leq t < t_2 \\ \dots & \dots \end{cases}$ | Net-load "duck curve" in daily power systems; macroeconomic regime shifts; CPU utilization under load. |
| Gaussian | $y = ae^{-\frac{(t-t_0)^2}{2\sigma^2}}$ | Product life-cycle peak (Bass diffusion density); one-off promotional shocks. |
| **Periodic Functions** | | |
| Sine Wave | $y = A\sin(2\pi ft + \phi)$ | Models smooth cyclic phenomena like ocean tides, seasonal temperature variations, or sound wave pressure. |
| Triangle Wave | $y = \frac{2A}{\pi}\arcsin(\sin(2\pi ft))$ | Models linear charging/discharging cycles in electronics, or simplified vibrations in mechanical systems. |
| Square Wave | $y = A \cdot \text{sgn}(\sin(2\pi ft))$ | Models on-off switching patterns like digital signals, heartbeats, machinery with two distinct states, or control systems with binary outputs. |
| Sawtooth Wave | $y = 2A(t/T - \lfloor t/T + 1/2 \rfloor)$ | Models rapid return phenomena such as voltage in electrical circuits with capacitors, ramp generators, or instrument sounds like brass. |
| Composite Sine | $y = \sum_{i=1}^{n} A_i \sin(2\pi f_i t + \phi_i)$ | Models complex periodic phenomena like musical tones with harmonics, combined seasonal effects, or multiple overlapping business cycles. |
| Exponential Sine | $y = e^{\sin(t)}$ | Models amplitude-modulated oscillations in signals, or periodic systems with exponentially varying intensity. |
| **Other Signals** | | |
| ARMA Signal | $x_t = \mu + \sum_{i=1}^{p} \phi_i x_{t-i} + \sum_{j=1}^{q} \theta_j \epsilon_{t-j} + \epsilon_t$ | Models systems with memory and feedback, such as financial returns, temperature fluctuations, or industrial production. |
| Random Walk | $x_t = x_{t-1} + \epsilon_t$ | Models unpredictable accumulating processes like stock prices, exchange rates, or particle movements. |
| White Noise | $x_t = \epsilon_t, \epsilon_t \sim N(0, \sigma^2)$ | Models purely random fluctuations like measurement errors, static in communication, or thermal noise in electronics. |
| Composite Signal | $x_t = \sum_{i=1}^{n} \alpha_i x_i(t)$ | Models real-world complex systems combining multiple patterns, such as economic indicators with trends, seasonality and noise. |

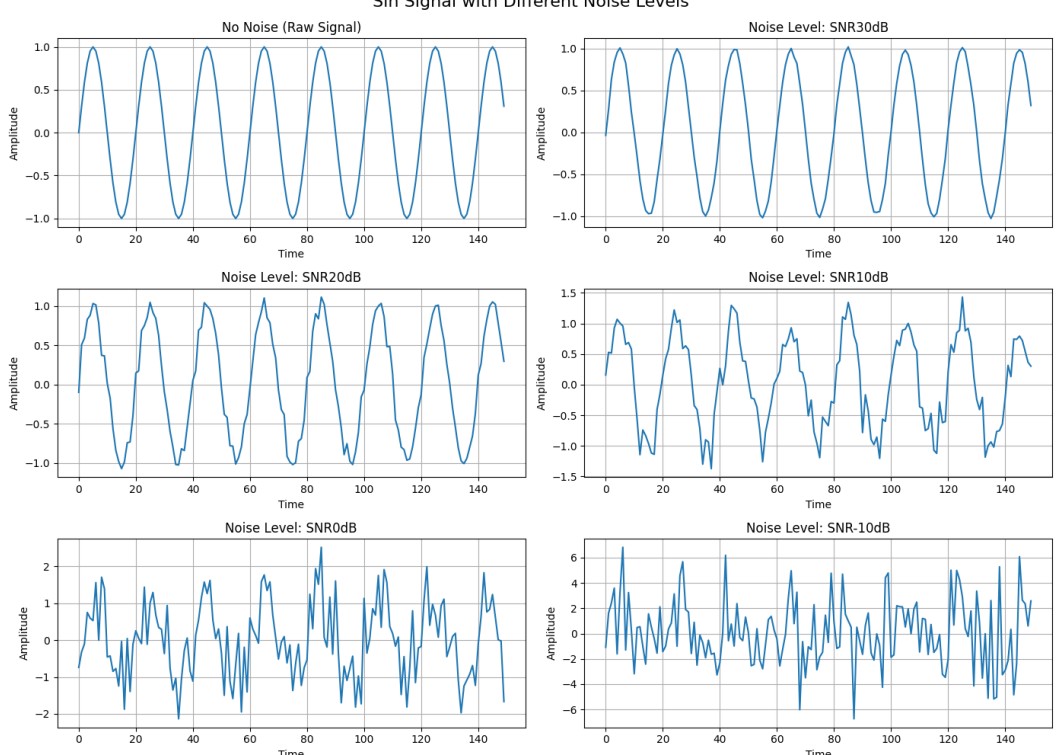

Figure 25: Sine wave signal with progressive noise levels. From top to bottom: clean signal, 30dB, 20dB, 10dB, 0dB, and -10dB SNR. Higher SNR values indicate less noise, while negative SNR values represent cases where noise dominates the signal.

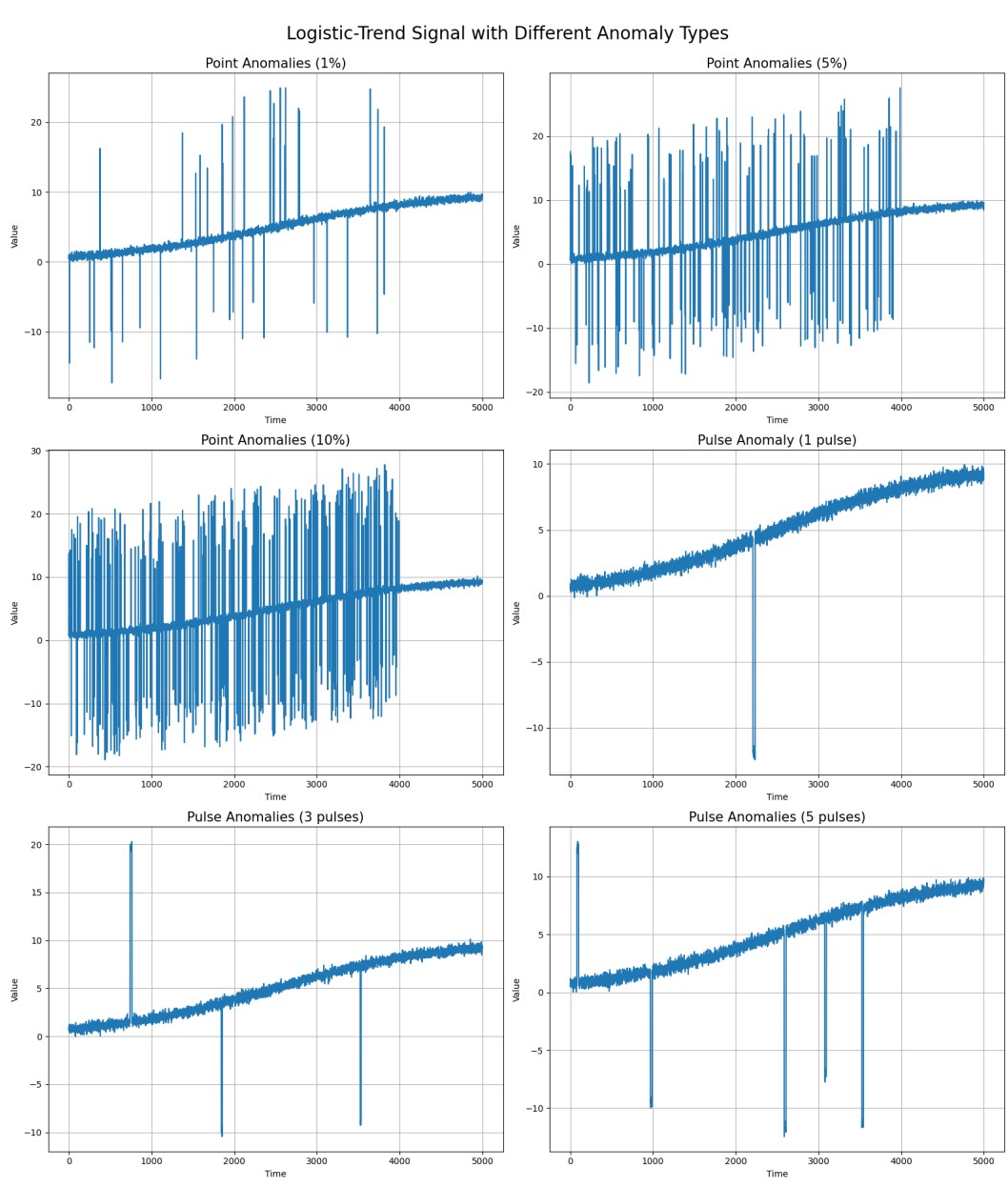

Figure 26: Logistic trend signals with various anomaly types. Top row shows point anomalies at 1% and 5% densities, middle row shows point anomalies at 10% and a single pulse anomaly, bottom row shows multiple pulse anomalies (3 and 5 pulses).

## C.4 Complex Univariate Datasets

To comprehensively evaluate model performance on complex time series patterns, we generated synthetic datasets that combine multiple basic signals to simulate complex temporal behaviors found in various domains. These composite signals capture key characteristics of real-world time series while maintaining fully controlled generation processes, enabling precise evaluation of model performance across diverse pattern types.

For example, our stock price simulator combines five essential components: (1) a slow exponential trend representing long-term market growth, (2) overlapping sine waves with different frequencies simulating market cycles, (3) higher-frequency seasonal patterns, (4) a random walk component capturing market noise according to the efficient market hypothesis, and (5) volatility clusters using GARCH-like behavior to model periods of market stress. Similarly, our temperature sensor simulation combines annual and daily periodic patterns with varying amplitudes, a gradual trend component, and realistic noise.

Other simulated patterns include electricity consumption with daily, weekly, and seasonal components; industrial sensor readings with degradation patterns; network traffic with diurnal and weekly patterns plus random bursts; retail sales with multiple seasonal effects; and economic indicators with trend-cycle-noise components. Each pattern type incorporates domain-specific characteristics essential for realistic forecasting challenges. Figure 27 provides visualization examples of these synthesized time series, demonstrating their diversity and complexity. The complete generation details are available in our open-source codebase.

## C.5 Complex Multivariate Datasets

We generated a diverse set of multivariate time series datasets to evaluate model performance on realistic and interdependent temporal patterns. These datasets include both classical dynamic systems and synthetic scenarios with explicit causal relationships. For example, we used differential equation models such as the Lotka-Volterra equations (predator-prey dynamics) and the SIR epidemic model to create time series where variables interact according to well-defined mathematical laws. Additionally, we constructed datasets with clear causal structures, such as the Weather-Sales dataset (where temperature and rainfall influence ice cream, umbrella, and beverage sales), as well as scenarios modeling advertising-sales effects, macroeconomic indicators, supply-demand-price interactions, and intervention effects. The relationships among variables in these datasets are designed to reflect real-world dependencies, such as weather driving consumer behavior or supply and demand jointly determining prices. Figures 28 to 30 illustrate some representative examples, while many more multivariate datasets and their generation details can be found in our open-source codebase.

Complex Generated Univariate Time Series (First 1000 Points)

### Network-Traffic Time Series

### Economic-Indicator Time Series

### Temperature-Sensor Time Series

### Website-Traffic Time Series

### Industrial-Sensor Time Series

### Electricity-Consumption Time Series

### Stock-Price Time Series

### Retail-Sales Time Series

Figure 27: Examples of complex univariate time series generated by combining multiple basic signal patterns.

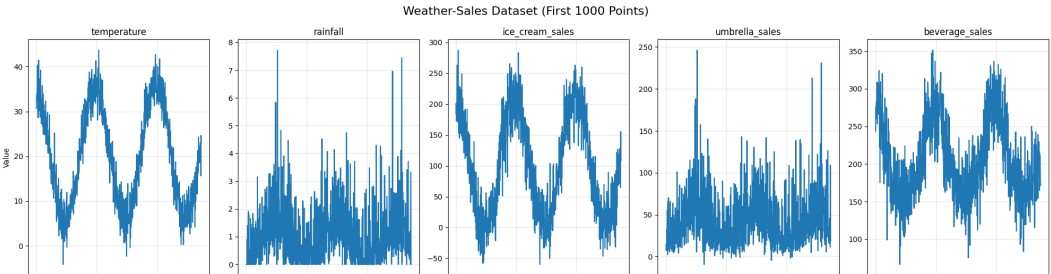

Weather-Sales Dataset (First 1000 Points)

Figure 28: Weather-Sales dataset showing relationships between weather variables (temperature, rainfall) and consumer purchases (ice cream sales, umbrella sales, beverage sales).

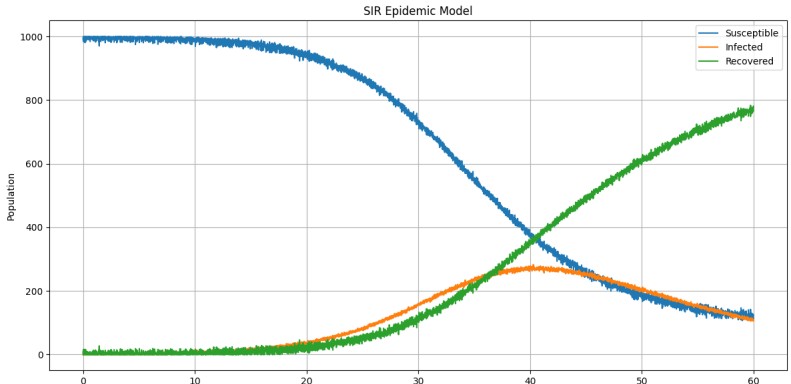

Figure 29: SIR epidemic model simulation showing the progression of an outbreak through Susceptible (blue), Infected (orange), and Recovered (green) populations over time.

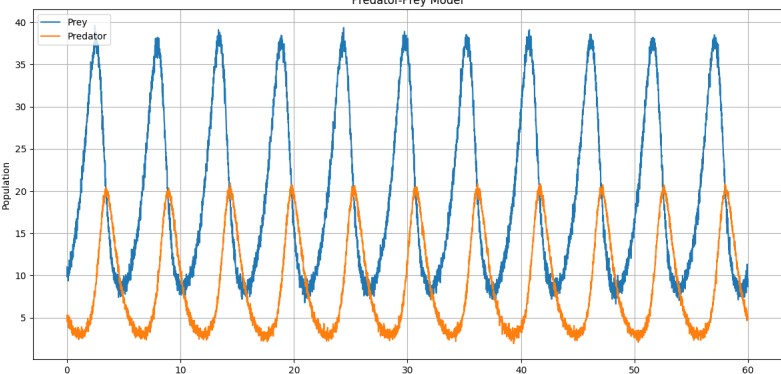

Figure 30: Lotka-Volterra predator-prey model demonstrating the cyclical relationship between prey (blue) and predator (orange) populations, where changes in one population affect the other with a time delay.

# D   Experiment Details

## D.1   Experiment Parameters

Our experiments were conducted on high-performance computing infrastructure consisting of four NVIDIA A800-SXM GPUs, each with approximately 79.3 GB (81251 MiB) of memory, running CUDA version 12.4. For dataset partitioning, we employed different strategies based on model types: deep learning models used a 7:1:2 ratio for training, validation, and testing sets respectively, while traditional models used an 8:2 split for training and validation. All evaluations employed a sliding window prediction approach with a stride of 1, where traditional models utilized the entire historical data available prior to the prediction point.

Table 17 presents the detailed experimental configuration across different evaluation scenarios, including input window sizes, forecast horizons, and dataset lengths used for each experiment type. Table 18 details the hyperparameter configurations for each full-shot model. For zero-shot forecasting models, we used the following model variants: chronos-bolt-base, TimeMoE-200M, and Moirai-small. These models were loaded using the official implementations from their respective GitHub repositories: Chronos (`https://github.com/amazon-science/chronos-forecasting`), TimeMoE (`https://github.com/Time-MoE/Time-MoE`), and Moirai (`https://github.com/SalesforceAIResearch/uni2ts`).

Table 17: Experiment Parameters for Different Dataset Parts. *Note*: "Original Scale Evaluation" indicates whether evaluation metrics were calculated after inverting normalized values back to their original scale (Yes) or directly on the normalized values (No).

| Experiment Type | Input Window | Forecast Horizons | Dataset Length | Original Scale Evaluation | Running Time (hours) |
|---|---|---|---|---|---|
| Temporal Pattern Learning (Trend Period Testing) | 96 | 24, 48, 96, 192 | 5000 | No | 30 |
| Robustness Against Noise & Anomalies | 96 | 96 | 5000 | No | 20 |
| Short/Long-range Dependencies | 96 | 10 | 5000 | Yes | 50 |
| Cross-Variable Learning | 96 | 96 | 5000 | No | 3 |
| Dataset Scale Sensitivity | 96 | 96 | 1000, 5000, 10000, 20000 | No | 40 |
| Complex Real-World Pattern Simulation | 96 | 96 | 5000 | No | 10 |
| Zero-shot Model Testing | 512 | 96 | 5000 | No | 3 |

Table 18: Hyperparameter Settings for Time Series Models

| Model | Architecture Parameters | d_model | d_ff | learning_rate | training_epoch | Other Parameters |
|---|---|---|---|---|---|---|
| TimeKAN | e_layers=2 | 16 | 32 | 0.01 | 30 | down_sampling_layers=2, down_sampling_window=2, begin_order=0 |
| TimeMixer | e_layers=2 | 16 | 32 | 0.01 | 30 | down_sampling_layers=2, down_sampling_method="avg", down_sampling_window=2 |
| TimesNet | e_layers=2 | 16 | 32 | 0.001 | 30 | top_k=5, num_kernels=6 |
| TSMixer | e_layers=2 | 512 | 2048 | 0.001 | 30 | factor=3 |
| SegRNN | seg_len=2 | 512 | - | 0.0001 | 30 | - |
| PatchTST | e_layers=3 | 512 | 2048 | 0.0001 | 30 | patch_len=16, n_heads=4, factor=3 |
| iTransformer | e_layers=3 | 512 | 512 | 0.0005 | 30 | n_heads=8, factor=3 |
| DLinear | - | - | - | 0.01 | 30 | label_len=10, moving_avg=25 |
| Autoformer | e_layers=2, d_layers=1 | 512 | 2048 | 0.001 | 30 | label_len=10, factor=3, moving_avg=25, n_heads=8 |
| PaiFilter | - | - | - | 0.01 | 30 | hidden_size=256 |
| TexFilter | - | - | - | 0.001 | 30 | embed_size=512, hidden_size=512 |
| TimeLLM | llm_layers=32 | 32 | 128 | 0.01 | 5 | patch_len=16, stride=8, label_len=10, factor=3, n_heads=8 |
| CATS | d_layers=3 | 256 | 512 | 0.01 | 30 | patch_len=48, stride=48, n_heads=32, QAM_start=0.1, QAM_end=0.2 |
| N-BEATS | n_blocks=[3,3,3] | - | - | 0.001 | 30 | stack_types=["seasonality", "trend", "identity"], mlp_units=[[512,512],[512,512],[512,512]], n_harmonics=10 |
| N-HiTS | n_blocks=[1,1,1] | - | - | 0.001 | max_steps=1500 | stack_types=["identity", "identity", "identity"], mlp_units=[[512,512],[512,512],[512,512]] |

### D.2 Evaluation Framework

#### D.2.1 Evaluation Metrics

**Mean Absolute Error (MAE)** The Mean Absolute Error (MAE) quantifies the average absolute deviation between the predicted values and the actual values. It treats all errors with equal weight regardless of their magnitude, making it robust to outliers. MAE is particularly useful when the cost of errors increases linearly with their size. Its mathematical formula is given by:

$$\text{MAE} = \frac{1}{n} \sum_{t=1}^{n} |y_t - \hat{y}_t| \tag{3}$$

where $y_t$ represents the actual value at time $t$, $\hat{y}_t$ represents the predicted value at time $t$, and $n$ is the total number of observations.

**Mean Squared Error (MSE)** The Mean Squared Error (MSE) measures the average of the squares of the errors between predicted and actual values. By squaring the errors before averaging, MSE penalizes larger errors more heavily than smaller ones, making it especially sensitive to outliers. It is widely used when large errors are particularly undesirable. The mathematical representation is:

$$\text{MSE} = \frac{1}{n} \sum_{t=1}^{n} (y_t - \hat{y}_t)^2 \tag{4}$$

**Root Mean Squared Error (RMSE)** The Root Mean Squared Error (RMSE) is the square root of MSE, bringing the error metric back to the same unit as the original data. This makes RMSE more interpretable than MSE while still maintaining the property of penalizing large errors more heavily. RMSE is commonly used in regression problems and time series forecasting. It can be calculated as:

$$\text{RMSE} = \sqrt{\frac{1}{n} \sum_{t=1}^{n} (y_t - \hat{y}_t)^2} \tag{5}$$

**Mean Absolute Percentage Error (MAPE)** The Mean Absolute Percentage Error (MAPE) expresses the prediction error as a percentage of the actual value, providing a scale-independent measure of accuracy. This makes it useful for comparing forecast performance across different datasets with varying scales. However, MAPE is undefined for zero values and can give misleadingly high errors for small actual values. Its formula is:

$$\text{MAPE} = \frac{1}{n} \sum_{t=1}^{n} \left| \frac{y_t - \hat{y}_t}{y_t} \right| \times 100\% \tag{6}$$

**Symmetric Mean Absolute Percentage Error (SMAPE)** The Symmetric Mean Absolute Percentage Error (SMAPE) is a modified version of MAPE that addresses some of its limitations. By using the average of actual and predicted values in the denominator, SMAPE avoids division by zero and provides a more balanced measure that's bounded between 0% and 100%. It treats positive and negative errors symmetrically, making it suitable for evaluations where both over-forecasting and under-forecasting are equally important. The formula is:

$$\text{SMAPE} = \frac{100\%}{n} \sum_{t=1}^{n} \frac{|y_t - \hat{y}_t|}{\frac{|y_t| + |\hat{y}_t|}{2}} \tag{7}$$

#### D.2.2 Theoretical Performance Boundaries

One significant advantage of using synthetic datasets is that we can precisely determine the theoretical optimal prediction for each time series type. In real-world datasets, the underlying data generation process is typically unknown, making it impossible to establish true performance boundaries.

With our synthetic approach, we can separate each time series into its predictable and unpredictable components, allowing us to calculate the theoretical minimum error achievable by any forecasting model.

**Core Logic**    Our calculation of the theoretical optimum is based on time series decomposition:

$$\text{Time Series} = \text{Predictable Part} + \text{Unpredictable Part} \tag{8}$$

This represents the performance limit of an ideal model that has full knowledge of the Data Generating Process (DGP). The **Predictable Part** consists of deterministic functions (e.g., trend, seasonality), which an ideal model can perfectly predict. The **Unpredictable Part** is generated by stochastic processes (e.g., white noise, random walk), for which the best forecast is its mathematical expectation.

Therefore, our **Theoretical Optimal Forecast** is defined as:

$$\text{Optimal Forecast}(t + h) = \text{Predictable\_Part}(t + h) + \mathbb{E}[\text{Unpredictable\_Part}(t + h)] \tag{9}$$

The error between this forecast and the actual value, which includes the stochastic components, stems entirely from the unavoidable randomness we designed into the data. This provides a reliable theoretical lower bound for model performance. Specifically:

- For **white noise** $\epsilon(t) \sim \mathcal{N}(0, \sigma^2)$: The expectation is 0, so $\mathbb{E}[\epsilon(t + h)] = 0$.
- For **random walk** $p(t) = p(t-1) + \epsilon(t)$: The best forecast for the future is the last observed value $p(t)$. In this context, the unpredictable component is the actual future realization of $\epsilon(t + h)$, whose expectation is 0.
- For **ARMA processes**: The optimal forecast comes from the correctly specified ARMA model with true parameters, which captures all predictable temporal dependencies through $\mathbb{E}[X_{t+h}|X_t, X_{t-1}, \ldots]$.

**Trend and Periodic Datasets**    For trend and periodic datasets without noise, the time series are generated by deterministic mathematical functions without any stochastic components. Since these patterns are entirely predictable given sufficient historical data, the theoretical optimal MSE and MAE values are exactly zero. This provides a clear benchmark against which to measure model performance on pure pattern learning.

**Stochastic Processes**    For stochastic processes, the optimal prediction strategy depends on the specific process characteristics:

- For random walk processes, the optimal prediction is the naive forecast (i.e., $\hat{y}_{t+1} = y_t$), since by definition $y_{t+1} = y_t + \epsilon_t$ where $\epsilon_t$ is unpredictable noise. Therefore, the optimal MSE/MAE is calculated using the naive forecasting method.
- For white noise processes, the optimal prediction is the mean of the process (assuming prior knowledge that the series is white noise but without knowing its mean), as each observation is independent of previous values.
- For ARMA processes, the optimal prediction comes from the correctly specified ARMA model with the true parameter values used in data generation, as this captures all predictable temporal dependencies.

**Datasets with Noise**    For datasets with added noise, we know both the observed values and the true underlying signal. The optimal prediction is based on the true signal component without noise. The optimal MSE/MAE is calculated by comparing this true signal with the observed values, representing the theoretical minimum error achievable in the presence of observation noise.

**Complex Time Series Datasets**    For complex time series like Network-Traffic or Stock-Price that combine multiple components (trend, seasonality, event-driven spikes, and noise), we identify which components are theoretically predictable from historical data (typically trend and seasonality) and which are not (random events and noise). The optimal prediction includes only the predictable

components, and the optimal MSE/MAE is calculated by comparing these predictable components with the full observed series.

To ensure fair comparison, all optimal calculations are performed using the same multi-step forecast horizons as those used by the deep learning models in our experiments. This approach allows us to quantify precisely how close each forecasting model comes to the theoretical performance ceiling for each type of time series pattern.

# E    Models Forecasting Visualization

In this subsection, we provide visual comparisons of model forecasting performance across a diverse range of time series patterns. Figure 31 to Figure 34 show the prediction results of various forecasting models on representative datasets. We visualize model behavior on fundamental trend patterns (linear and exponential trends in Figure 31), periodic signals (Square-Wave and Triple-Sin patterns in Figure 32), and the impact of varying noise levels on both periodic and trend patterns (Double-Sin and Linear-Trend in Figure 35 and Figure 36). Additionally, we showcase forecasting capabilities on time series with temporal dependency characteristics (ARMA(1,1) process and Random Walk in Figure 33) and on more complex simulated real-world patterns (Network Traffic and Temperature Sensor data in Figure 34). These visualizations complement our quantitative analyses by providing intuitive representations of how different architectures handle various temporal patterns.

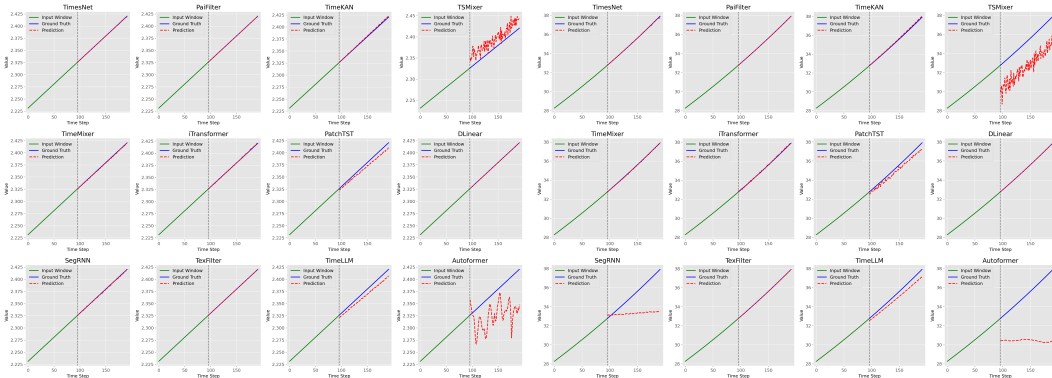

Figure 31: Comparison of model predictions for different trend patterns. The left panels show predictions for linear trend, while the right panels show predictions for exponential trend. Each subplot represents a different model's prediction (red dashed line) compared to ground truth (blue solid line) and input window (green solid line).

# F    Limitations

While our synthetic benchmark framework offers valuable insights into model capabilities, several limitations should be acknowledged. First, our generated time series, despite their complexity, may not fully capture the intricacies of real-world data where multiple factors interact through sophisticated, non-linear mechanisms rather than simple superposition of components. Second, our current framework does not address time-varying patterns where the underlying dynamics evolve over time—a common characteristic in many real-world systems that we plan to explore in future work. Third, due to computational constraints, we conducted all experiments with a fixed input window size of 96 time steps, which, although standard in the field, limits our understanding of how different temporal contexts might affect model performance.

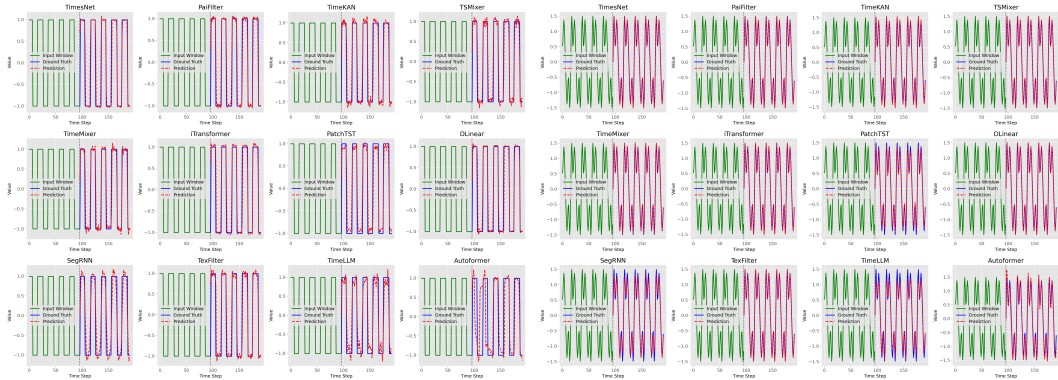

Figure 32: Comparison of model predictions for different periodic patterns. The left panels show predictions for Square-Wave patterns, while the right panels show predictions for Triple-Sin patterns. Each subplot represents a different model's prediction (red dashed line) compared to ground truth (blue solid line) and input window (green solid line).

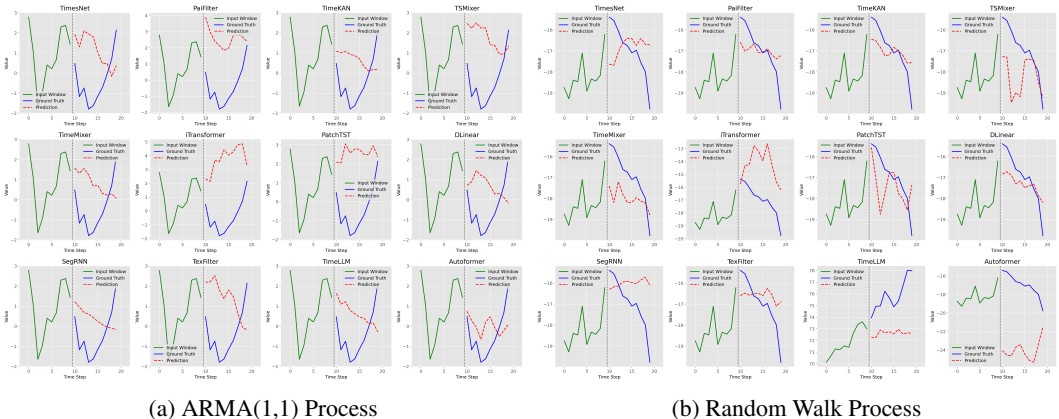

(a) ARMA(1,1) Process            (b) Random Walk Process

Figure 33: Model forecasting comparison on time series with temporal dependency characteristics. Left panels show predictions for ARMA(1,1) process with autocorrelation; right panels show predictions for Random Walk pattern commonly observed in financial data. Each subplot displays a model's prediction (red dashed line) against the ground truth (blue solid line) with the input window (green solid line).

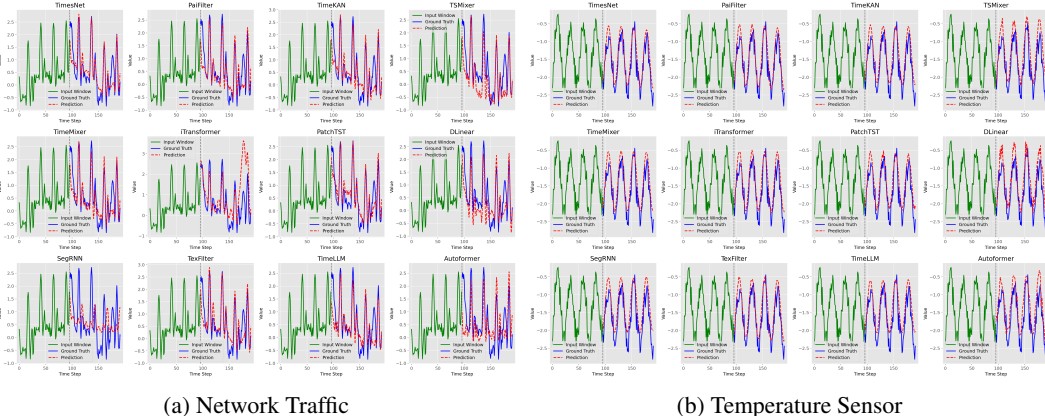

(a) Network Traffic            (b) Temperature Sensor

Figure 34: Model forecasting performance on simulated real-world time series data. Left panels show predictions for Network Traffic; right panels show predictions for Temperature Sensor. Each subplot displays a model's prediction (red dashed line) against the ground truth (blue solid line) with the input window (green solid line).

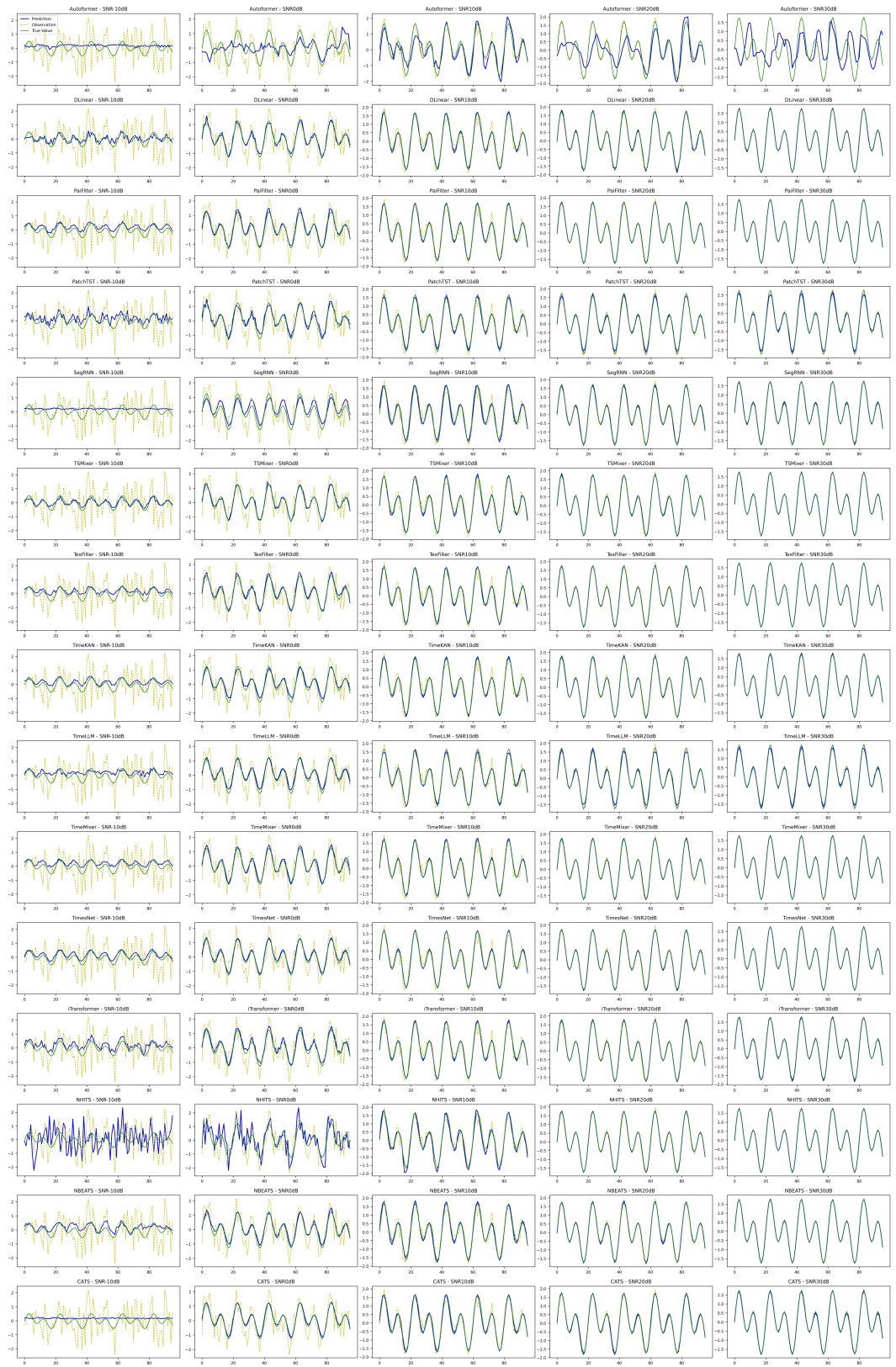

Figure 35: Comparison of model predictions under different noise levels for all models (Double-Sin). Blue lines represent model predictions, green lines show the true underlying signal, and yellow dashed lines represent the noisy observations.

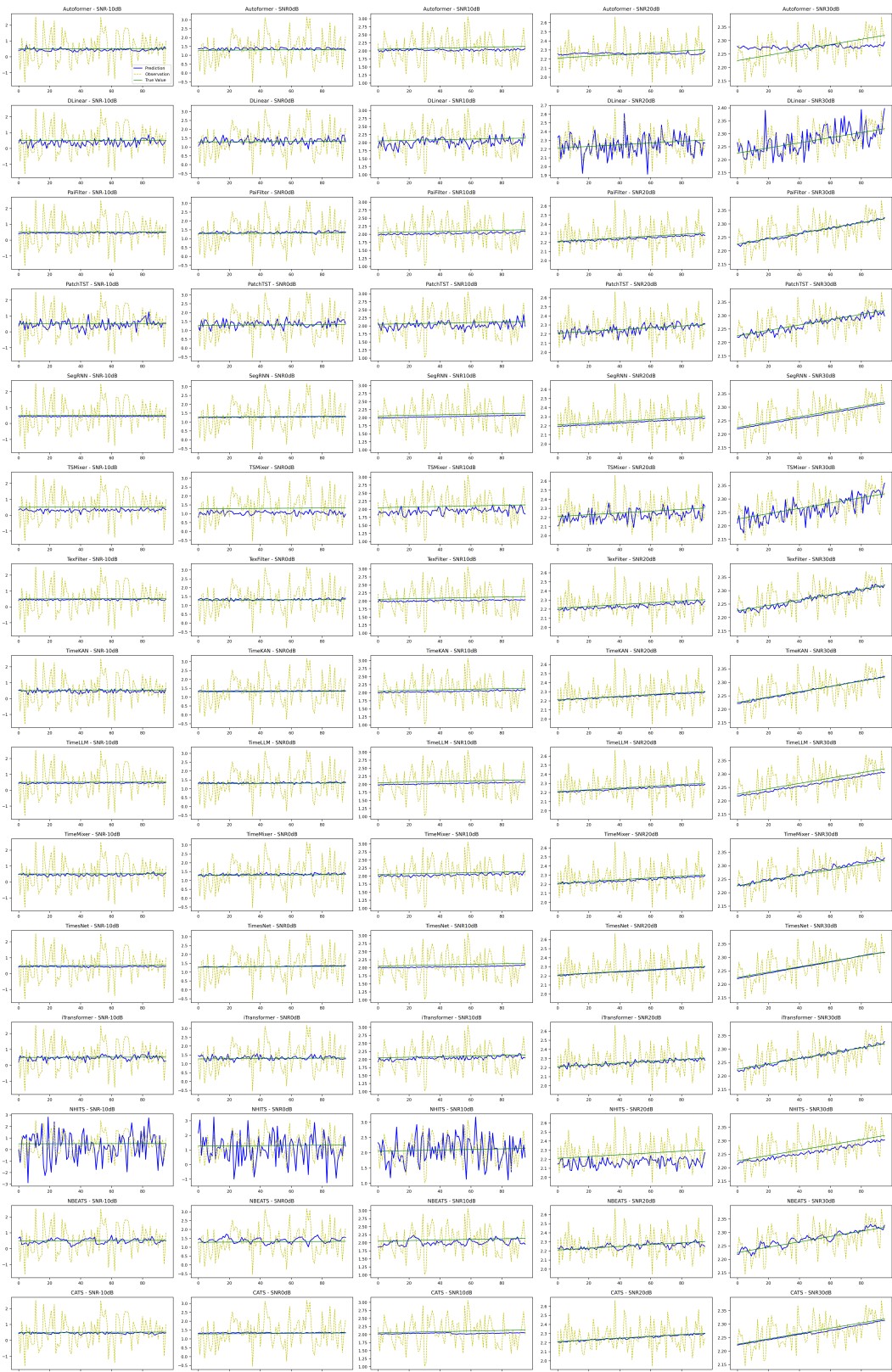

Figure 36: Comparison of model predictions under different noise levels for all models (Linear-Trend). Blue lines represent model predictions, green lines show the true underlying signal, and yellow dashed lines represent the noisy observations.

