# OpenReview forum: "SynTSBench: Rethinking Temporal Pattern Learning in Deep Learning Models for Time Series"
_NeurIPS.cc/2025/Datasets_and_Benchmarks_Track — NeurIPS 2025 Datasets and Benchmarks Track poster_

### Official Review · Reviewer_jQqa · 2025-06-26

**Rating:** 5
**Confidence:** 5

**Summary:**

This paper proposes using controlled time series data to evaluate the capabilities of different forecasting models. It generates time series data with various characteristics, covering different temporal patterns, noise and anomalies, short/long-term dependencies, cross-variable relationships, and different scales. Additionally, it simulates complex real-world patterns to evaluate performance in more realistic scenarios. Extensive experiments are conducted using the generated data to investigate different models.

**Additional Feedback:**

1, In the "Evaluation on Cross-Variable Learning," using only two variables is insufficient to draw conclusions, as real-world data typically have many variables with complex relationships. The conclusion "This suggests that when variable relationships become more complex and nuanced..." in the Appendix should be discussed in the main text.
2. Tables 5 and 6 are not referenced in the text.
3. The definitions of RMSE, MAPE, and SMAPE are provided in the Appendix but not used.

**Dataset Code Accessibility:**

Yes

**Ethical Considerations:**

No, there are no or only very minor ethics concerns

**Final Justification:**

Before the rebuttal, I had two main concerns: whether the generated data can reflect real-world data, and how the dataset length affects the analysis. The authors have addressed these concerns well. It is recommeded that the additional analysis on longer dataset lengths will be included in the paper. I have further updated my score.

**Limitations Weaknesses:**

1. The biggest concern is ensuring that the generated data reflects real-world data. Real-world data are much more complex and consist of various patterns, which may cause models to behave differently than analyzed in the paper. It is unclear to what extent the analysis is consistent with real-world data.
2. The dataset length used in the main text is unclear. As shown in Figure 1 of the Appendix, results change with different dataset lengths, potentially altering conclusions. How can the effect of dataset length be alleviated to provide robust conclusions?
3. Although the paper concludes that "no single architecture dominates across all patterns," this is a common insight among researchers. The valuable contribution would be suggesting optimal models based on given datasets, especially real-world datasets with complex patterns.
4. Insights from real-world pattern simulations should connect with those from data with only key temporal characteristics, to confirm consistency. Additionally, not only simulated real-world data but actual real-world data should be considered.

**Strengths Contributions:**

1. The paper proposes using controlled time series data to evaluate forecasting models, which is an interesting and seemingly effective approach.
2. Theoretical optimum results are provided in some cases to assist the analysis.
3. Extensive experiments reveal the advantages and limitations of different forecasting models on various data types.

---

> ### Author Rebuttal · Authors · 2025-07-30
>
> Thank you for the insightful comments and valuable suggestions!
>
> **Q1: Generalization to real-world complexity remains unverified.**
>
> **A1**: We fully agree that real-world time series are far more complex and heterogeneous than our synthetic benchmark, so “perfectly reproducing” the true distribution is neither feasible nor the goal of this paper. Instead, our core contribution is a set of controlled experiments: we build synthetic datasets that isolate single or just a few mechanisms—trend, seasonality, cross-variable dependence, etc.—and turn model behavior into quantifiable dimensions. This lets us ask a more fundamental question: when one mechanism is made salient, can the model still perform reliably? We argue that if a model struggles under these highly simplified conditions, its reliability in the wild must be questioned; conversely, strong performance here gives us a clear, actionable hint about where to focus when we later tune the model on real data.
>
> To empirically validate our benchmark's effectiveness, we have conducted extensive new experiments on eight widely-used real-world datasets. The table1 below presents the MSE of each model on these datasets.
>
> ### Table 1  Comprehensive comparison of MSE for real-world datasets (All results are averaged from four different forecasting horizons: H ∈ [24, 48, 96, 192], input horizon = 96)
>
> | model        | ETTh1  | ETTh2  | ETTm1  | ETTm2  | Electricity | Exchange-rate | Traffic | Weather |
> | -------------- | -------- | -------- | -------- | -------- | ------------- | --------------- | --------- | --------- |
> | Autoformer   | 0.5792 | 0.4569 | 0.5971 | 0.3229 | 0.3555      | 0.3040        | 0.7140  | 0.4565  |
> | DLinear      | 0.5122 | 0.4228 | 0.4545 | 0.3277 | 0.3141      | 0.1765        | 0.7335  | 0.3060  |
> | PaiFilter    | 0.4916 | 0.3991 | **0.4178** | 0.2828 | 0.2771      | 0.1868        | 0.5779  | 0.2691  |
> | PatchTST     | 0.4952 | 0.3929 | 0.4310 | 0.2894 | 0.2949      | 0.1774        | 0.5744  | 0.2761  |
> | SegRNN       | 0.5270 | 0.3939 | 0.4583 | 0.2834 | 0.3471      | 0.1817        | 0.8679  | **0.2654**  |
> | TSMixer      | 0.5656 | 1.7774 | 0.5303 | 0.4727 | 0.3023      | 0.4876        | 0.6276  | 0.2805  |
> | TexFilter    | 0.4997 | 0.4032 | 0.4396 | 0.2881 | **0.2572**      | 0.1816        | 0.5201  | 0.2700  |
> | TimeKAN      | 0.4795 | **0.3884** | 0.4228 | 0.2810 | 0.3024      | **0.1755**        | 0.6903  | 0.2688  |
> | TimeMixer    | **0.4858** | 0.3904 | 0.4216 | **0.2799** | 0.2701      | 0.1805        | 0.5780  | 0.2679  |
> | TimesNet     | 0.5425 | 0.4112 | 0.4771 | 0.2933 | 0.2983      | 0.2192        | 0.6928  | 0.2862  |
> | iTransformer | 0.5050 | 0.4061 | 0.4404 | 0.3026 | 0.2607      | 0.1903        | **0.5155**  | 0.3107  |
>
>
> To make the comparison more direct, we calculated the average performance rank for each model across all real-world datasets and compared it to their rank on our synthetic benchmark. The table2 below shows the results.
>
> ### Table2 Comparison of model performance on real datasets and synthetic datasets （Synthetic Complex Dataset Avg MSE is computed as the average of the evaluation results obtained on the Complex Real-world Pattern Simulations dataset.）
>
> | Model        | Real Dataset Avg MSE | Real Dataset Rank | Synthetic Complex Dataset Avg MSE | Synthetic Dataset Rank |
> | -------------- | ---------------------- | ------------------- | ----------------------------------- | ------------------------ |
> | Autoformer   | 0.4733               | 10                | 0.8956                            | 11                     |
> | DLinear      | 0.4059               | 8                 | 0.3379                            | 8                      |
> | PaiFilter    | 0.3628               | 3                 | 0.3004                            | 3                      |
> | PatchTST     | 0.3664               | 4                 | 0.3119                            | 6                      |
> | SegRNN       | 0.4156               | 9                 | 0.6951                            | 10                     |
> | TSMixer      | 0.6305               | 11                | 0.4903                            | 9                      |
> | TexFilter    | 0.3575               | 1                 | 0.294                             | 1                      |
> | TimeKAN      | 0.3761               | 6                 | 0.3081                            | 5                      |
> | TimeMixer    | 0.3593               | 2                 | 0.2956                            | 2                      |
> | TimesNet     | 0.4026               | 7                 | 0.3016                            | 4                      |
> | iTransformer | 0.3664               | 5                 | 0.3165                            | 7                      |
>
> The results of this hybrid analysis are highly revealing. We can find a strong correlation between the performance on our synthetic datasets and real-world data. Most notably, **the top three performing models in our synthetic benchmark—TexFilter, TimeMixer, and PaiFilter—are identically ranked as the top three performers on average across the eight real-world datasets.**
>
> This strong consistency at the top of the leaderboard provides powerful evidence for our central thesis: our synthetic benchmark is not an isolated academic exercise, but a reliable and effective tool for predicting a model's real-world performance. It successfully identifies robust, high-performing architectures, thereby demonstrating its value and significance as a more unified and dependable benchmark for future time series research.
>
> **Q2: The dataset length used in the main text is unclear.**
>
> **A2**: Except for the experiment that specifically investigates the impact of dataset size, all other experiments in this paper use datasets capped at 5,000 samples; details are provided in Table 7 of the Appendix. We chose this length because, in the dataset-size exploration, most of models had already approached its individual optimum by 5,000 samples, and further increases yielded only marginal gains. Longer datasets would impose prohibitive training costs, especially given the large number of datasets and models we must evaluate. Consequently, 5,000 represents a balanced trade-off between preserving model performance and containing computational expense, and we believe this choice does not materially compromise our experimental findings or conclusions.
>
> **Q3: The conclusion is a bit superficial**
>
> **A3**:
> We appreciate your valuable suggestion regarding the need for more specific guidance on model selection. To address this, we have indeed conducted a more in-depth analysis of each model's strengths and weaknesses, specifically linking their architectural designs to their observed experimental performance. This aims to provide more direct guidance and analytical depth to our work. For a detailed summary of this analysis, we kindly refer you to our response to Reviewer GkMU, Question 3. We apologize for any inconvenience caused by not repeating the full details here, which is due to strict word count limitations.
>
> **Q4: Insights consistency**
>
> **A4**:
> Our experimental results clearly demonstrate the consistency in simulated data. A model's performance on complex data directly reflects its combined strengths and weaknesses on fundamental capabilities. For instance, **DLinear** excels at learning clean, individual trend and periodic patterns, but our analysis reveals its poor robustness to noise and anomalies. This directly explains its suboptimal performance on our complex simulated datasets, which incorporate such irregularities. Conversely, the top-performing models on both our complex simulated and real-world datasets—​**TexFilter, TimeMixer, and PaiFilter**​—demonstrate strong, well-rounded capabilities across all key areas, including trend, periodicity, and robustness to noise and anomalies. This provides strong evidence that performance on fundamental patterns dictates performance in complex scenarios.
>
>
>
> **Additional Feedback: Cross-variable evaluation with only two variables is insufficient for real-world complexity.**
>
> **Answer:**
> We agree that our initial cross-variable evaluation with only two variables was insufficient to capture the full complexity of real-world multivariate relationships. To thoroughly address this, we have expanded our evaluation to include three new, more complex cross-variable experiments: Conditional Relationship, Nonlinear Relationship, and Complex Multivariable (5 Var) interactions.
>
> ### Table 3 **Cross-Variable Evaluation Results（MSE）**
>
> | Signal Type                   | Autoformer | DLinear | PaiFilter | PatchTST | SegRNN | TSMixer | TexFilter | TimeKAN | TimeMixer | TimesNet | iTransformer | Optimal |
> | ------------------------------- | ------------ | --------- | ----------- | ---------- | -------- | --------- | ----------- | --------- | ----------- | ---------- | -------------- | -------- |
> | Conditional Relationship      | 1.0149     | 0.9972  | 1.0030    | 1.0241   | 1.0027 | **0.8332**  | 1.0016    | 1.0049  | 0.8533    | 0.8665   | 0.8955       | 0.7967 |
> | Complex Multivariable (5 Var) | 0.9880     | 1.0279  | 1.0272    | 1.0557   | 1.0087 | 0.9374  | 1.0191    | 1.0069  | **0.8901**    | 0.9105   | 1.0066       | 0.8767 |
> | Nonlinear Relationship        | 0.9331     | 0.9869  | 0.9795    | 1.0205   | 0.9670 | **0.7302**  | 0.9788    | 0.9656  | 0.7481    | 0.7385   | 0.7617       | 0.7136 |
>
> These new cross-variable experiments further corroborate our prior findings, demonstrating consistent model performance trends even with increased complexity in inter-variable relationships. Notably, as the number of variables increased and relationships became more complex in the "Complex Multivariable (5 VAR)" scenario, TSMixer's performance showed some decline, while TimeMixer and TimesNet maintained relatively stable performance.

---

> > ### Comment · Reviewer_jQqa · 2025-08-04
> >
> > Thank you very much for your responses. Most of my concerns have been addressed, and I will update my score accordingly. However, I do not fully agree with the answer to Q2. As shown in Figure 1 in the Appendix, many models still have not converged with 5,000 samples, which suggests that further investigation with more samples is needed. Additionally, regarding Q3, it would be more helpful to explain how to select the optimal models based on the given datasets, rather than merely listing the models' strengths and weaknesses.

---

> > > ### Author Response · Authors · 2025-08-05
> > >
> > > Thank you very much for your thoughtful feedback and for your willingness to update your score. We are glad that our previous rebuttal addressed most of your concerns. We have carefully considered your remaining points regarding dataset length (Q2) and model selection guidance (Q3), and we hope to fully address them with the following explanation.

---

> > > > ### Author Response · Authors · 2025-08-05
> > > >
> > > > **Re: Q2 - Discussion on Dataset Length**
> > > >
> > > > To facilitate this discussion, we first wish to clarify the specific experimental setup for the dataset length analysis. We generated synthetic datasets with lengths of 1,000, 5,000, 10,000, and 20,000 time steps. For this analysis, we selected six representative signal patterns (three trend functions and three periodic functions). Each of these patterns was generated with four distinct noise levels (SNR of 20dB, 0dB, -10dB, and no noise).
> > > > Our core argument is twofold:
> > > > 1. The gradual improvement on noisy datasets is expected and has a relatively minor impact on overall model rankings. The averaged results in Table 4 show a general trend of decreasing MSE as dataset length increases. Our analysis indicates that this is largely attributable to the datasets containing noise. With noisy signals, a larger number of samples naturally allows a model to better average out the statistical variance, leading to a steadier, more gradual reduction in MSE. **This phenomenon reflects statistical convergence on a noisy signal rather than a model needing more data to grasp the fundamental underlying pattern.**
> > > >
> > > > ### Table 4: Avg MSE across different Dataset Lengths
> > > > | Model        | Length=1000 | Length=5000 | Length=10000 | Length=20000 |
> > > > | -------------- | ------------- | ------------- | -------------- | -------------- |
> > > > | Autoformer   | 3.8354      | 1.0442      | 0.8341       | 0.8054       |
> > > > | DLinear      | 0.7775      | 0.4767      | 0.6436       | 0.4631       |
> > > > | PaiFilter    | 0.5537      | 0.4678      | 0.4484       | 0.4274       |
> > > > | PatchTST     | 0.7728      | 0.5049      | 0.4702       | 0.4419       |
> > > > | SegRNN       | 2.1838      | 0.6229      | 0.4707       | 0.4347       |
> > > > | TSMixer      | 0.7591      | 0.8123      | 1.1776       | 0.7029       |
> > > > | TexFilter    | 0.5543      | 0.4782      | 0.4532       | 0.4269       |
> > > > | TimeKAN      | 1.5980      | 0.4910      | 0.4570       | 0.4312       |
> > > > | TimeMixer    | 0.8238      | 0.4870      | 0.4511       | 0.4273       |
> > > > | TimesNet     | 1.2756      | 0.4700      | 0.4515       | 0.4253       |
> > > > | iTransformer | 0.8734      | 0.4916      | 0.4543       | 0.4280       |
> > > >
> > > >
> > > > 2. On Noiseless Data, Performance Stabilizes and Exposes Core Model Flaws. When we examine the noiseless "Five-Sin" data (Table 5), a key insight emerges: **for most models, performance changes very little between 5,000 and 20,000 samples.** This stability indicates that these models have effectively reached their performance ceiling for this fundamental pattern, demonstrating their efficiency in learning core temporal relationships without the confounding factor of noise. Furthermore, regarding models such as Autoformer and SegRNN that appeared "unconverged" in Figure 1 in the Appendix, their behavior does not suggest that our dataset size is insufficient, but rather reveals core flaws in their model design. **The primary factors impacting their performance are their inherent training instability and low sample efficiency.** Specifically, Autoformer's performance degrades as more data is added (with its MSE increasing from 0.361 to 0.556), which exposes the instability of its training process. Additionally, the error rates of both SegRNN and Autoformer at 20,000 samples remain significantly higher than those of other top-performing models at just 5,000 samples. This provides strong evidence of their fundamental sample inefficiency, meaning they require a disproportionately large amount of data to learn effectively.
> > > >
> > > > ### Table 5: MSE on Five-Sin (No Noise) Dataset
> > > > | model        | length=1000 | length=5000 | length=10000 | length=20000 |
> > > > | -------------- | ------------- | ------------- | -------------- | -------------- |
> > > > | Autoformer   | 8.32E-01    | 3.61E-01    | 2.27E-01     | 5.56E-01     |
> > > > | DLinear      | 4.67E-10    | 6.22E-11    | 6.35E-11     | 2.71E-09     |
> > > > | PaiFilter    | 1.40E-04    | 6.75E-06    | 1.18E-06     | 1.80E-07     |
> > > > | PatchTST     | 9.23E-03    | 8.54E-03    | 1.15E-02     | 9.43E-03     |
> > > > | SegRNN       | 9.32E-01    | 3.84E-01    | 3.91E-02     | 3.29E-03     |
> > > > | TSMixer      | 1.04E-03    | 2.39E-05    | 2.77E-06     | 2.48E-07     |
> > > > | TexFilter    | 1.20E-04    | 5.18E-06    | 9.64E-07     | 3.36E-07     |
> > > > | TimeKAN      | 2.90E-03    | 2.60E-04    | 1.98E-04     | 2.42E-04     |
> > > > | TimeMixer    | 8.83E-04    | 2.41E-04    | 1.14E-04     | 1.60E-04     |
> > > > | TimesNet     | 2.13E-03    | 1.10E-04    | 6.82E-05     | 4.55E-05     |
> > > > | iTransformer | 5.40E-04    | 2.15E-04    | 2.96E-04     | 3.52E-04     |
> > > >
> > > >
> > > > In conclusion, the choice of 5,000 samples represents a deliberate trade-off between evaluation depth (sample size) and evaluation breadth (the vast number of controlled patterns and models). While extending all datasets to 20,000 samples would incur prohibitive computational costs, **as this would nearly quadruple our total training time**, our analysis shows that it would not fundamentally alter our core findings.

---

> > > > > ### Author Response · Authors · 2025-08-05
> > > > >
> > > > > **Re: Q3 - Providing More Prescriptive Model Selection Guidance**
> > > > >
> > > > > To address your point directly, our benchmark can indeed guide model selection when a time series is known to be dominated by specific characteristics. **For instance, when the data exhibits a clear trend or periodic pattern with minimal noise and no anomalies, a simple model like DLinear can achieve extremely high prediction accuracy. Conversely, when the data contains significant noise or anomalies, models such as PaiFilter, TexFilter, and TimeKAN demonstrate much better robustness.** For a comprehensive overview, we would refer you to the heatmap in Figure 10 of our main paper. This figure details the performance of all evaluated models across each fundamental data characteristic, allowing practitioners to analyze the trade-offs and select an appropriate model for various types of single-characteristic datasets.
> > > > >
> > > > > However, we must stress that applying such a framework to real-world datasets has significant limitations. The core challenge is that real-world data is rarely defined by a single, isolated pattern. Instead, it is almost always a complex and often unknown superposition of multiple patterns (e.g., trend, multiple seasonalities, noise, and latent cross-variable dependencies). **This complexity makes it difficult to definitively generalize from our foundational experiments to prescribing a single "best model" for a specific real-world dataset.**
> > > > >
> > > > > **Therefore, the primary contribution of our work is not to offer a definitive guide for picking a winner for any given dataset, but rather to conduct a comprehensive, multi-dimensional evaluation of existing models.** Our goal is to create a detailed "report card" for each architecture, highlighting its fundamental strengths and weaknesses. This is why our analysis focuses on why models succeed or fail in specific aspects—we aim to provide clear, actionable insights that can guide future model improvements.
> > > > >
> > > > > Thank you again for your insightful feedback, which has helped us to better frame the contributions and limitations of our work.

---

> > > > > ### Comment · Reviewer_jQqa · 2025-08-06
> > > > >
> > > > > Thank you very much for the detailed explanations. The answers address my concerns. I believe the analysis of the capacity to capture patterns from noisy signal data is more important, as it corresponds to real data. The authors should include more content on longer dataset lengths.

---

> > > > > > ### Author Response · Authors · 2025-08-06
> > > > > >
> > > > > > Thank you for your constructive comments and for engaging with us in this productive discussion. We are glad our previous responses have addressed most of your concerns.
> > > > > >
> > > > > > We fully agree with your point regarding the importance of analyzing a model's capacity to capture patterns from noisy signals, particularly with longer datasets, as this better reflects real-world conditions. Your insight has prompted us to conduct a deeper investigation into this very question. To quantify a model's ability to learn the underlying pattern while filtering out noise, we measured the MSE\_True. As defined in Equation (2) of our main paper, MSE\_True represents the error between the model's predictions and the ground-truth clean signal.
> > > > > >
> > > > > > The results of this analysis for datasets(signal type: “Linear-Trend”, “Quadratic-Function”, “Exponential-Trend”, “Sin0.005”, “Double-Sin”, “Five-Sin”) with noise level (SNR = 0dB and SNR = -10dB) are presented in the Table 6 and Table 7.
> > > > > >
> > > > > > ### Table 6 Avg MSE\_True at SNR= 0 dB, across different Dataset Lengths
> > > > > >
> > > > > > | Model        | Length=1000     | Length=5000     | Length=10000    | Length=20000    |
> > > > > > | -------------- | ---------- | ---------- | ---------- | ---------- |
> > > > > > | Autoformer   | 0.704399 | 0.450243 | 0.428179 | 0.401894 |
> > > > > > | DLinear      | 0.505292 | 0.132028 | 0.219255 | 0.113853 |
> > > > > > | PaiFilter    | 0.258545 | 0.050276 | 0.034054 | 0.026697 |
> > > > > > | PatchTST     | 0.396182 | 0.081753 | 0.057479 | 0.044647 |
> > > > > > | SegRNN       | 0.625683 | 0.361407 | 0.084305 | 0.031966 |
> > > > > > | TSMixer      | 0.942779 | 0.884857 | 0.637823 | 0.770584 |
> > > > > > | TexFilter    | 0.29313  | 0.0518   | 0.033702 | 0.025361 |
> > > > > > | TimeKAN      | **0.187931** | 0.056476 | 0.041685 | 0.040199 |
> > > > > > | TimeMixer    | 0.257263 | 0.055233 | 0.037182 | 0.027643 |
> > > > > > | TimesNet     | 0.230353 | **0.041067** | **0.026822** | **0.022562** |
> > > > > > | iTransformer | 0.254642 | 0.053454 | 0.035902 | 0.026913 |
> > > > > >
> > > > > >
> > > > > > ### Table 7 Avg MSE\_True at SNR=-10 dB, across different Dataset Lengths
> > > > > >
> > > > > > | Model        | Length=1000     | Length=5000     | Length=10000    | Length=20000    |
> > > > > > | -------------- | ---------- | ---------- | ---------- | ---------- |
> > > > > > | Autoformer   | 0.314874 | 0.100355 | 0.090362 | 0.089762 |
> > > > > > | DLinear      | 0.346431 | 0.108681 | 0.093833 | 0.107009 |
> > > > > > | PaiFilter    | **0.125652** | 0.055046 | 0.04723  | 0.044767 |
> > > > > > | PatchTST     | 0.260462 | 0.103236 | 0.088989 | 0.075048 |
> > > > > > | SegRNN       | 0.161551 | 0.08113  | 0.067553 | 0.06681  |
> > > > > > | TSMixer      | 0.183071 | 0.143037 | 0.142063 | 0.147913 |
> > > > > > | TexFilter    | 0.153296 | 0.055856 | 0.047905 | 0.043016 |
> > > > > > | TimeKAN      | 0.206662 | 0.055644 | 0.050533 | 0.045977 |
> > > > > > | TimeMixer    | 0.178644 | 0.050963 | 0.047759 | 0.043576 |
> > > > > > | TimesNet     | 0.135684 | **0.0488**   | **0.044828** | **0.040588** |
> > > > > > | iTransformer | 0.159998 | 0.065833 | 0.053704 | 0.045946 |
> > > > > >
> > > > > > Our analysis of these results reveals that the models demonstrating superior performance—such as TimesNet, PaiFilter, TexFilter and TimeMixer—consistently maintain their lead as the dataset length increases. This observation strongly aligns with the conclusions drawn from our original analysis in the main paper.
> > > > > >
> > > > > > This supplementary analysis confirms that increasing the dataset length does not significantly alter our conclusions regarding the fundamental strengths and weaknesses of each architecture. We will incorporate this more detailed analysis and discussion into the final version of our paper to further strengthen our findings.
> > > > > >
> > > > > > Thank you once again for your invaluable comments and suggestions, which have significantly helped improve the rigor of our study.

---

> > > > > > > ### Comment · Area_Chair_ouiv · 2025-08-09
> > > > > > > **Please check the authors’ further response**
> > > > > > >
> > > > > > > Dear Reviewer jQqa,
> > > > > > >
> > > > > > > Thank you for your efforts in reviewing the paper. As we approach the end of the discussion period, could you please check whether the authors’ rebuttal and subsequent responses have addressed all your concerns?
> > > > > > >
> > > > > > > Best regards,
> > > > > > >
> > > > > > > AC

---

### Official Review · Reviewer_aDeY · 2025-07-01

**Rating:** 4
**Confidence:** 4

**Summary:**

This paper presents SynTSBench, a synthetic benchmark for systematically evaluating time series forecasting models across dimensions such as trend/seasonality modeling, robustness, and cross-variable reasoning. While the framework is well-motivated and offers fine-grained control over pattern generation, its over-reliance on synthetic data and lack of statistical rigor in result interpretation reduce its practical generalizability.

**Dataset Code Accessibility:**

Yes

**Ethical Considerations:**

No, there are no or only very minor ethics concerns

**Final Justification:**

I believe that in the time series domain, studying specific temporal properties to deepen our understanding of models is both meaningful and necessary.

Before the rebuttal, my main concern was that the temporal properties designed in the synthetic dataset were overly simplistic and might differ significantly from those in real-world data. However, during the rebuttal phase, the authors provided detailed experimental evidence showing that the synthetic and real datasets exhibit a high degree of similarity, with consistent evaluation outcomes. They also included additional experiments on more complex properties, such as cross-variable interactions. I think my concerns to be addressed and have updated my score to 4.

**Limitations Weaknesses:**

1. **Limited Realism of Synthetic Datasets.** While the benchmark generates controlled temporal patterns, it lacks validation against real-world statistical properties such as non-stationarity, regime shifts, and variable heterogeneity. This limits the ecological validity of its conclusions. Without real-world calibration, insights may not generalize beyond synthetic settings.

2. **Absence of Real-World Dataset Evaluation.** The paper omits experiments on real-world datasets, making it difficult to assess whether synthetic performance rankings correlate with real-world generalization. Benchmarking without grounding in real applications may lead to misleading implications for model deployment.

3. **Underspecified Theoretical Optima.** The methodology for computing theoretical optima, especially in multivariate or composite-pattern scenarios, lacks formal derivation or validation. It is unclear how these optima are derived under nonlinear or non-additive dynamics. This undermines the reliability of performance gap interpretation.

4. **Over-Simplified Cross-Variable Evaluation.** The cross-variable experiments use trivial lagged dependencies, which fail to reflect the complexity of multivariate interactions in real systems. Real-world tasks often involve nonlinear, conditional, or dynamic inter-variable relationships. The current setup may overestimate models' multivariate reasoning ability.

**Strengths Contributions:**

- Novel Evaluation Paradigm: Introduces a programmable synthetic benchmark with theoretical optima, enabling isolated and interpretable assessment of specific forecasting capabilities.
- Comprehensive Coverage: Systematically evaluates a wide range of temporal patterns (trend, periodicity, dependency, anomalies), offering detailed insights into model strengths and weaknesses.

---

> ### Author Rebuttal · Authors · 2025-07-31
>
> Thank you for the insightful comments and valuable suggestions!
>
> **Q1: Limited Realism of Synthetic Datasets**
>
> **A1**:
> Our synthetic benchmark focuses on assessing a model's fundamental ability to learn core temporal patterns (trends, seasonality, cycles). We believe mastering these basic structures is crucial before tackling real-world complexities. This work establishes a foundational understanding of model capabilities and robustness in controlled settings. While our current focus is foundational, our synthetic complex pattern datasets do incorporate elements mimicking real-world offsets and non-stationarities from events like policy changes. This is an initial step toward reflecting dynamic real-world data. However, a deep dive into how models specifically handle non-stationarity, regime shifts, and variable heterogeneity is a more complex investigation. We're actively planning to explore this in future work, where we'll specifically design frameworks to analyze these advanced statistical challenges. Thank you for your suggestion!
>
> **Q2: Absence of Real-World Dataset Evaluation.**
>
> **A2**:
> To empirically validate our benchmark's effectiveness, we have conducted extensive new experiments on eight widely-used real-world datasets. The table1 below presents the MSE of each model on these datasets.
>
> ### Table 1  Comprehensive comparison of MSE for real-world datasets (All results are averaged from four different forecasting horizons: H ∈ [24, 48, 96, 192], input horizon = 96)
>
> | model        | ETTh1  | ETTh2  | ETTm1  | ETTm2  | Electricity | Exchange-rate | Traffic | Weather |
> | -------------- | -------- | -------- | -------- | -------- | ------------- | --------------- | --------- | --------- |
> | Autoformer   | 0.5792 | 0.4569 | 0.5971 | 0.3229 | 0.3555      | 0.3040        | 0.7140  | 0.4565  |
> | DLinear      | 0.5122 | 0.4228 | 0.4545 | 0.3277 | 0.3141      | 0.1765        | 0.7335  | 0.3060  |
> | PaiFilter    | 0.4916 | 0.3991 | **0.4178** | 0.2828 | 0.2771      | 0.1868        | 0.5779  | 0.2691  |
> | PatchTST     | 0.4952 | 0.3929 | 0.4310 | 0.2894 | 0.2949      | 0.1774        | 0.5744  | 0.2761  |
> | SegRNN       | 0.5270 | 0.3939 | 0.4583 | 0.2834 | 0.3471      | 0.1817        | 0.8679  | **0.2654**  |
> | TSMixer      | 0.5656 | 1.7774 | 0.5303 | 0.4727 | 0.3023      | 0.4876        | 0.6276  | 0.2805  |
> | TexFilter    | 0.4997 | 0.4032 | 0.4396 | 0.2881 | **0.2572**      | 0.1816        | 0.5201  | 0.2700  |
> | TimeKAN      | 0.4795 | **0.3884** | 0.4228 | 0.2810 | 0.3024      | **0.1755**        | 0.6903  | 0.2688  |
> | TimeMixer    | **0.4858** | 0.3904 | 0.4216 | **0.2799** | 0.2701      | 0.1805        | 0.5780  | 0.2679  |
> | TimesNet     | 0.5425 | 0.4112 | 0.4771 | 0.2933 | 0.2983      | 0.2192        | 0.6928  | 0.2862  |
> | iTransformer | 0.5050 | 0.4061 | 0.4404 | 0.3026 | 0.2607      | 0.1903        | **0.5155**  | 0.3107  |
>
> To make the comparison more direct, we calculated the average performance rank for each model across all real-world datasets and compared it to their rank on our synthetic benchmark. The table2 below shows the results.
>
> ### Table2 Comparison of model performance on real datasets and synthetic datasets （Synthetic Complex Dataset Avg MSE is computed as the average of the evaluation results obtained on the Complex Real-world Pattern Simulations dataset.）
>
> | Model        | Real Dataset Avg MSE | Real Dataset Rank | Synthetic Complex Dataset Avg MSE | Synthetic Dataset Rank |
> | -------------- | ---------------------- | ------------------- | ----------------------------------- | ------------------------ |
> | Autoformer   | 0.4733               | 10                | 0.8956                            | 11                     |
> | DLinear      | 0.4059               | 8                 | 0.3379                            | 8                      |
> | PaiFilter    | 0.3628               | 3                 | 0.3004                            | 3                      |
> | PatchTST     | 0.3664               | 4                 | 0.3119                            | 6                      |
> | SegRNN       | 0.4156               | 9                 | 0.6951                            | 10                     |
> | TSMixer      | 0.6305               | 11                | 0.4903                            | 9                      |
> | TexFilter    | 0.3575               | 1                 | 0.294                             | 1                      |
> | TimeKAN      | 0.3761               | 6                 | 0.3081                            | 5                      |
> | TimeMixer    | 0.3593               | 2                 | 0.2956                            | 2                      |
> | TimesNet     | 0.4026               | 7                 | 0.3016                            | 4                      |
> | iTransformer | 0.3664               | 5                 | 0.3165                            | 7                      |
>
> The results of this hybrid analysis are highly revealing. We can find a strong correlation between the performance on our synthetic datasets and real-world data. Most notably, **the top three performing models in our synthetic benchmark—TexFilter, TimeMixer, and PaiFilter—are identically ranked as the top three performers on average across the eight real-world datasets.**
>
> This strong consistency at the top of the leaderboard provides powerful evidence for our central thesis: our synthetic benchmark is not an isolated academic exercise, but a reliable and effective tool for predicting a model's real-world performance. It successfully identifies robust, high-performing architectures, thereby demonstrating its value and significance as a more unified and dependable benchmark for future time series research.
>
>
> **Q3: Underspecified Theoretical Optima**
>
> **A3**:
> We would like to clarify the calculation logic for the theoretical optima.
>
> ### Core Logic
>
> Our calculation of the theoretical optimum is based on time series decomposition:
>
> `Time Series = Predictable Part + Unpredictable Part`
>
> This represents the performance limit of an ideal model that has full knowledge of the Data Generating Process (DGP).
>
> 1. **Predictable Part**: This consists of deterministic functions (e.g., trend, seasonality). An ideal model can perfectly predict the future values of this part.
> 2. **Unpredictable Part**: This is generated by stochastic processes (e.g., white noise, random walk). The best forecast for this part is its mathematical expectation.
>    * **White Noise**: The expectation is 0.
>    * **Random Walk** $ p(t) = p(t-1) + \epsilon(t) $: The best forecast for the future is the last observed value $p(t)$. In this context, the unpredictable component is the actual future realization of $\epsilon(t+h)$, whose expectation is 0.
>
>
>
>
>
> Therefore, our **Theoretical Optimal Forecast** is defined as:
>
> `Optimal Forecast(t+h) = Predictable_Part(t+h) + E[Unpredictable_Part(t+h)]`
>
> The error between this forecast and the actual value, which includes the stochastic components, stems entirely from the unavoidable randomness we designed into the data. This provides a reliable theoretical lower bound for model performance.
>
> ### Example: `Stock-Price` Dataset
>
> This dataset is an **additive** combination of:
>
> * **Predictable Part**: An exponential trend and multiple composite sine waves (representing market cycles and seasonality).
> * **Unpredictable Part**: A random walk process and a GARCH stochastic process.
>
> The **optimal forecast** is calculated as:
>
> `Forecast(t+h) = [Exact future trend & cycle values] + [Last known value of the random walk]`
>
> Due to the word limit of the reply, I can only give a brief analysis and explanation here, and will further supplement and improve the relevant content in the appendix of the paper later.
>
> **Q4: Over-Simplified Cross-Variable Evaluation**
>
> **A4**:
> Regarding the cross-variable evaluation, we acknowledge the concern about over-simplification. To address this, we have expanded our evaluation with three new cross-variable experiments: Conditional Relationship, Nonlinear Relationship, and Complex Multivariable (5 Var) interactions. The results are presented in Table 3.
>
> ### Table 3 **Cross-Variable Evaluation Results（MSE）**
>
> | Signal Type                   | Autoformer | DLinear | PaiFilter | PatchTST | SegRNN | TSMixer | TexFilter | TimeKAN | TimeMixer | TimesNet | iTransformer | Optima |
> | ------------------------------- | ------------ | --------- | ----------- | ---------- | -------- | --------- | ----------- | --------- | ----------- | ---------- | -------------- | -------- |
> | Conditional Relationship      | 1.0149     | 0.9972  | 1.0030    | 1.0241   | 1.0027 | **0.8332**  | 1.0016    | 1.0049  | 0.8533    | 0.8665   | 0.8955       | 0.7967 |
> | Complex Multivariable (5 Var) | 0.9880     | 1.0279  | 1.0272    | 1.0557   | 1.0087 | 0.9374  | 1.0191    | 1.0069  | **0.8901**    | 0.9105   | 1.0066       | 0.8767 |
> | Nonlinear Relationship        | 0.9331     | 0.9869  | 0.9795    | 1.0205   | 0.9670 | **0.7302**  | 0.9788    | 0.9656  | 0.7481    | 0.7385   | 0.7617       | 0.7136 |
>
> These new cross-variable experiments further corroborate our prior findings, demonstrating consistent model performance trends even with increased complexity in inter-variable relationships. Notably, as the number of variables increased and relationships became more complex in the "Complex Multivariable (5 VAR)" scenario, TSMixer's performance showed some decline, while TimeMixer and TimesNet maintained relatively stable performance.

---

> > ### Comment · Reviewer_aDeY · 2025-08-04
> > **Thanks for the detailed response.**
> >
> > Thank you very much for your detailed response. All of my concerns have been addressed, and I’m inclined to increase my score. The author might consider including some experiments of complex statistical property (such as the Cross-Variable Evaluation Results) in the main paper. This could help broaden the scope of your conclusions.

---

> > > ### Author Response · Authors · 2025-08-04
> > >
> > > Thank you for your valuable feedback! We are very pleased to hear that our detailed response has successfully addressed your concerns.
> > >
> > > We truly appreciate your suggestion to include experiments on complex statistical properties, such as the expanded cross-variable evaluations, in the main paper. We agree that this will significantly strengthen our conclusions and broaden the scope of our work. We will definitely incorporate these findings into the final version of the paper.
> > >
> > > Your thoughtful comments have been instrumental in improving the quality of our research, and we are grateful for your time and expertise.

---

### Official Review · Reviewer_GkMU · 2025-07-01

**Rating:** 5
**Confidence:** 4

**Summary:**

While time series models have demonstrated strong performance on popular benchmark datasets, there is growing concern that these results may not translate effectively to real-world scenarios. In light of this issue, the authors introduce a suite of artificially generated time series datasets designed to evaluate the robustness and generalization ability of forecasting models. These datasets include multiple trend and periodic signal types, as well as noisy variants to assess robustness. This work makes a valuable contribution to the time series forecasting community by enabling a clearer understanding of the strengths and limitations of existing models.

**Additional Feedback:**

N/A

**Dataset Code Accessibility:**

Yes

**Dataset Code Comments:**

Authors provide open-source code and datasets through GitHub and HuggingFace.

**Ethical Considerations:**

No, there are no or only very minor ethics concerns

**Final Justification:**

The authors have significantly improved the manuscript through additional experiments and clearer elaboration of their results. These enhancements can be summarized as follows:

1. The authors now provide a more rigorous rationale for signal selection, including relevant formulas and domain-specific applications for various signal types, such as logarithmic, exponential, and logistic trends.

2. The revised manuscript includes performance evaluations of recent models such as N-BEATS and N-HITS. The authors also address the omission of models like CycleNet and discuss this in the context of future scaling and analysis directions.

3. The authors offer a structured analysis of each model’s strengths and limitations, with attention to architectural-performance trade-offs—for example, highlighting PaiFilter’s robustness and the comparatively limited effectiveness of TimeLLM.

In light of these improvements, I raise my score from 4 to 5.

**Limitations Weaknesses:**

While the work is novel and meaningful, I recommend that the authors address the following issues:

**1. Theoretical Justification for Signal Selection**

Although the use of well-defined functions like Gaussian and linear trends is understandable, the paper lacks theoretical or domain-specific motivation for the selection of specific signal types. For example, exponential trends and step functions are frequently used in financial and control systems. Providing references or background to justify the inclusion of particular signal types would strengthen the rationale and applicability of the dataset design.

**2. Recent Related Work and Comparison to SOTA**

Given the rapid advancements in time series forecasting, the paper would benefit from a more thorough discussion of recent state-of-the-art (SOTA) methods. Notably, the CATS model [1], which achieves SOTA performance in recent benchmarks, should be included in Tables 1 and 2 for a more comprehensive evaluation. If possible, including these models in experiments and visualizations would significantly enhance the paper’s impact. The authors could also consider additional recent works, such as [2] and [3].

[1] Kim, D., et al. "Are Self-Attentions Effective for Time Series Forecasting?" NeurIPS 2024.

[2] Shi, J., et al. "Scaling Law for Time Series Forecasting." NeurIPS 2024.

[3] Lin, S., et al. "Cyclenet: Enhancing Time Series Forecasting through Modeling Periodic Patterns." NeurIPS 2024.

**3. Clearer Summary of Model Strengths and Weaknesses**

The results show that models like DLinear and PaiFilter exhibit stable performance across different settings, while others like TimesNet and TimeKAN excel under noisy conditions. However, the paper could benefit from a clearer summary of each model’s strengths and weaknesses, including failure cases. For instance, Table 3 shows that TimeLLM performs poorly on signals like Sin(fre=0.005) and Double-Sin, suggesting possible structural or training limitations. Highlighting such observations can lead to insightful discussions about model design trade-offs and suggest new research directions, thereby increasing the novelty of the work.

I would be happy to raise my score if the authors address the concerns above.

**Strengths Contributions:**

Evaluations in time series forecasting are often limited to a small set of widely used datasets, such as ETTh and Weather. However, models often overfit these datasets, which can obscure their true performance in practical applications. The synthetic datasets proposed in this paper offer a promising direction by providing controlled and diverse evaluation scenarios, potentially leading to more unified and reliable benchmarks for future research.

---

> ### Author Rebuttal · Authors · 2025-07-30
>
> Thank you for your insightful review and for recognizing the novelty of our work！
>
> **Q1: Theoretical Justification for Signal Selection**
>
> **A1**：In Appendix Table 6 we provide a concise presentation of each chosen signal, including its formula and a brief description of the corresponding application scenario. We further supplement the trend signals with additional context and remarks in Table 1.
> ### Table 1 Time-Series Signal Functions with Domain-Specific Motivations
>
> | **Function**         | **Time-oriented Scenario & Domain Justification**                                                                               |
> | ---------------------------- | --------------------------------------------------------------------------------------------------------------------------------------- |
> | **Quadratic**        | Position under constant acceleration (s = v₀t + ½at²); cumulative production cost with accelerating learning.                      |
> | **Exponential**      | Compound interest, inflation, early epidemic spread ($ SIR: I(t) ≈ e^{βt} $); capacitor discharge $ i(t)=I₀e^{-t/RC} $.                    |
> | **Logarithmic**      | Wright learning curve; software bug-discovery rate decay; Weber-Fechner perception law.                                               |
> | **Logistic**         | Resource-limited growth: new-product adoption, biological populations, rumor spreading.                                               |
> | **Gompertz**         | Longitudinal tumor volume growth; actuarial mortality curves; technology-substitution saturation.                                     |
> | **Power Law**        | DLA cluster radius $r(t) ∝ t^{1/d\_f}$; Heaps law in evolving networks $ n\_t ∝ t^{β}$; Omori–Utsu aftershock decay $n(t) ∝ t^{-p}$ |
> | **Step Function**    | Tick-level price jumps; control-system set-point changes; AWS Step Functions state transitions.                                       |
> | **Piecewise Linear** | Net-load “duck curve” in daily power systems; macroeconomic regime shifts; CPU utilization under load.                              |
> | **Gaussian**         | Product life-cycle peak (Bass diffusion density); one-off promotional shocks.                                                         |
>
>
> **Q2: Recent Related Work and Comparison to SOTA**
>
> **A2**：We have now included the evaluation results for CATS [1], N-BEATS [2], N-HITS [3], and TPGN [4] in Table 2. As for CycleNet , its reliance on periodic priors could lead to an unfair comparison on our general synthetic datasets; thus, it was not included in this evaluation.
>
> ### Table 2 Evaluation results of the supplementary model (MSE)
>
> | Dataset Type           | CATS      | N-BEATS    | N-HITS     | TPGN      |
> | ------------------------ | ----------- | ----------- | ----------- | ----------- |
> | Trend                  | 0.0451946 | 0.0000027 | 0.0012840 | 0.0000075 |
> | Period                 | 0.0283966 | 0.0117732 | 0.0105458 | 0.0417024 |
> | Complex Univariate   | 0.2701272 | 0.2188650 | 0.2991683 | 0.2570174 |
> | Complex Multivariate | 0.5155118 | 0.4356473 | 1.5722923 | 0.5230839 |
>
> Our analysis of these newly added models indicates that N-BEATS generally performs well, though it is still slightly outperformed by models like PaiFilter and TimeMixer from our initial evaluations. More detailed experimental results will be further elaborated in the main paper. What's more, we appreciate the mention of "Scaling Law for Time Series Forecasting" . While highly relevant, conducting a detailed scaling analysis across our diverse synthetic datasets and numerous models would entail significant resource consumption, thus falling beyond the scope of this current work. However, it is certainly a valuable direction for future research.
>
> [1] Kim, D., et al. "Are Self-Attentions Effective for Time Series Forecasting?" NeurIPS 2024
>
> [2] Oreshkin, B. N., et al. "N-BEATS: Neural Basis Expansion Analysis for Interpretable Time Series Forecasting." ICLR 2020.
>
> [3] Challu, C., et al. "N-HiTS: Neural Hierarchical Interpolation for Time Series Forecasting." AAAI, 2023.
>
> [4] Jia, Y., et al. "PGN: The RNN’s New Successor is Effective for Long-Range Time Series Forecasting." NeurIPS 2024.
>
> **Q3: Clearer Summary of Model Strengths and Weaknesses**
>
> **A3**:
> We have provided a detailed breakdown of each model's key strengths, including their weaknesses, in the accompanying table 3.
>
> ### Table 3 Model Capability Analysis
> | **Model Name**   | **Key Strengths**                                                                                                                                               | **Main Weaknesses**                                                                                                                                         |
> | ------------------------ | ----------------------------------------------------------------------------------------------------------------------------------------------------------------------- | ------------------------------------------------------------------------------------------------------------------------------------------------------------------- |
> | **DLinear**      | Strong modeling capability for simple trend and periodic signals.                                                                                                     | Performance drops significantly when patterns become complex.                                                                                                     |
> | **PaiFilter&TexFilter**    | Extremely strong fitting capability for trend and periodic signals, stable performance under noise and anomalies, and strong modeling capability on complex datasets. | Limited capability for modeling cross-variable dependencies.                                                                                                      |
> | **TimeMixer**    | Comprehensive performance, with good results in modeling trend/periodic signals, learning multivariate relationships, and complex patterns.                           | Mediocre performance in modeling short-term dependencies.                                                                                                         |
> | **TimeKAN**      | Extremely strong robustness to noise and anomalies, and outstanding capability in modeling complex signals.                                                           | Limited capability for modeling cross-variable dependencies.                                                                                                      |
> | **TSMixer**      | Strong capability for modeling multivariate relationships.                                                                                                            | Poor robustness to anomalies, with errors increasing exponentially at high anomaly rates, and weak fitting capability for complex patterns.                       |
> | **TimesNet**     | Strong capability for modeling multivariate relationships, outstanding in modeling complex inter-variable dependencies, and performs well in noisy scenarios.         | Weak in modeling short-range dependencies and performs moderately on some trend signals.                                                                          |
> | **SegRNN**       | Good at modeling short-range dependencies.                                                                                                                            | Poor performance in modeling long-range dependencies, mediocre performance on trend/periodic signals, and limited capability for modeling variable relationships. |
> | **PatchTST**     | Good modeling capability for periodic and trend signals, and complex patterns.                                                                                        | Limited capability for modeling cross-variable relationships and poor capability for modeling short-term dependencies.                                            |
> | **iTransformer** | Comprehensive performance, with good results in modeling some trend/periodic signals, learning multivariate relationships, and complex patterns.                      | Poor performance in modeling short-term dependencies.                                                                                                             |
> | **TimeLLM**      | -                                                                                                                                                                     | Mediocre performance in all aspects, poor at modeling cross-variable dependencies, and has high training costs.                                                   |
>
> Beyond this, we further analyzed the intricate relationship between each model's architectural design and its observed performance, revealing crucial trade-offs. Due to response length constraints, we briefly highlight three key observations:
>
> **FilterNet's Dominance** : Models like PaiFilter and TexFilter consistently achieve state-of-the-art or near-SOTA performance across diverse forecasting tasks, underscoring their robust and effective architectural design for general time series analysis.
>
> **DLinear's Simplicity-Complexity Trade-off** : DLinear's straightforward linear structure excels in high-precision forecasting for basic trend and periodic patterns. However, its performance significantly degrades with increased pattern complexity, such as the introduction of noise or the superposition of multiple modes.
>
> **Limitations of LLM-based Models** : Under full-shot fine-tuning, the LLM-based TimeLLM demonstrates limited advantages, exhibiting poor accuracy in learning and forecasting temporal patterns. This suggests that the LLM architecture struggles with precise time series pattern learning while incurring substantial computational overhead.

---

> > ### Comment · Reviewer_GkMU · 2025-08-04
> >
> > Thank you for your response.
> > I truly appreciate the authors’ efforts in addressing all of my concerns.
> > The additional experiments and the improved clarity in the presentation of results significantly enhance the paper.
> >
> > **I am confident that the revised paper will provide valuable contributions to researchers in this domain.**
> > Therefore, I will revise my score accordingly.
> >
> > Thank you again for your hard work.

---

> > > ### Author Response · Authors · 2025-08-04
> > >
> > > Thank you for your response. We sincerely appreciate your careful review and the valuable feedback you provided! We are delighted to hear that our rebuttal, including the additional experiments and enhanced clarity, has addressed your concerns and strengthened the paper.
> > >
> > > Your recognition of our work is a great encouragement. We are committed to incorporating these revisions into the final version to ensure the paper makes a meaningful and lasting contribution to the time series community.

---

### Official Review · Reviewer_xEYn · 2025-07-02

**Rating:** 5
**Confidence:** 3

**Summary:**

Briefly, this paper presents a synthetic data-driven evaluation framework designed to systematically assess the capabilities of time series forecasting models. The proposed framework introduces three core analytical dimensions: temporal feature decomposition and capability mapping, robustness analysis under data irregularities, and theoretical optimum benchmarking. The authors conduct extensive experiments to evaluate 12 state-of-the-art models across these dimensions, revealing strengths and weaknesses in their ability to handle trends, periodicity, noise, anomalies, and cross-variable dependencies. The proposed framework advances in addressing limitations in current evaluation approaches, such as the inability to isolate temporal features and the lack of theoretical performance boundaries.

**Dataset Code Accessibility:**

Yes

**Dataset Code Comments:**

I can access the dataset via the GitHub link.

**Ethical Considerations:**

No, there are no or only very minor ethics concerns

**Final Justification:**

This paper presents a synthetic data-driven evaluation framework designed to systematically assess the capabilities of time series forecasting models. During the rebuttal, most of my concerns have been well-addressed. I still encourage the authors to improve the clarity of writing and presentation throughout the paper. In particular, some key experimental details could be more explicitly emphasized in the main paper to ensure the work is easily understandable to a broader audience.

Overall, I acknowledge the contribution of this paper.

**Limitations Weaknesses:**

+ The paper lacks validation on real-world datasets. Although the authors claim that synthetic data mimics real-world scenarios, the generalizability of the findings remains uncertain. A hybrid approach (combining synthetic and real-world data) could be better.

+ The evaluation focuses on 12 models, omitting other prominent architectures (e.g., N-BEATS, N-HiTS, or other RNN variants). This limits the comprehensiveness of the benchmark.

+ The noise and anomaly injections (Gaussian noise, point/pulse anomalies) are relatively simple. Real-world irregularities often exhibit more complex patterns (e.g., structural breaks, non-Gaussian noise). Expanding the range of irregularities would enhance the robustness analysis.

+ The evaluation relies heavily on MSE and MAE, which may not fully capture model performance in all scenarios (e.g., probabilistic forecasting, extreme event prediction).

**Strengths Contributions:**

+ This paper presents an interesting approach to evaluating time series forecasting models by using synthetic data with programmable features.

+ This paper thoroughly evaluates models across multiple dimensions, including trend and periodic pattern learning, robustness to noise and anomalies, and cross-variable dependency modeling.

+ The results highlight that no single model excels in all dimensions, emphasizing the need for task-specific model selection.

---

> ### Author Rebuttal · Authors · 2025-07-28
>
> Thank you for your insightful comments!
>
> **Q1: Lacking validation on real-world datasets.**
>
> **A1:**
> To empirically validate our benchmark's effectiveness, we have conducted extensive new experiments on eight widely-used real-world datasets. The table1 below presents the MSE of each model on these datasets.
>
> ### Table 1  Comprehensive comparison of MSE for real-world datasets (All results are averaged from four different forecasting horizons: H ∈ [24, 48, 96, 192], input horizon = 96)
>
> | model        | ETTh1  | ETTh2  | ETTm1  | ETTm2  | Electricity | Exchange-rate | Traffic | Weather |
> | -------------- | -------- | -------- | -------- | -------- | ------------- | --------------- | --------- | --------- |
> | Autoformer   | 0.5792 | 0.4569 | 0.5971 | 0.3229 | 0.3555      | 0.3040        | 0.7140  | 0.4565  |
> | DLinear      | 0.5122 | 0.4228 | 0.4545 | 0.3277 | 0.3141      | 0.1765        | 0.7335  | 0.3060  |
> | PaiFilter    | 0.4916 | 0.3991 | **0.4178** | 0.2828 | 0.2771      | 0.1868        | 0.5779  | 0.2691  |
> | PatchTST     | 0.4952 | 0.3929 | 0.4310 | 0.2894 | 0.2949      | 0.1774        | 0.5744  | 0.2761  |
> | SegRNN       | 0.5270 | 0.3939 | 0.4583 | 0.2834 | 0.3471      | 0.1817        | 0.8679  | **0.2654**  |
> | TSMixer      | 0.5656 | 1.7774 | 0.5303 | 0.4727 | 0.3023      | 0.4876        | 0.6276  | 0.2805  |
> | TexFilter    | 0.4997 | 0.4032 | 0.4396 | 0.2881 | **0.2572**      | 0.1816        | 0.5201  | 0.2700  |
> | TimeKAN      | 0.4795 | **0.3884** | 0.4228 | 0.2810 | 0.3024      | **0.1755**        | 0.6903  | 0.2688  |
> | TimeMixer    | **0.4858** | 0.3904 | 0.4216 | **0.2799** | 0.2701      | 0.1805        | 0.5780  | 0.2679  |
> | TimesNet     | 0.5425 | 0.4112 | 0.4771 | 0.2933 | 0.2983      | 0.2192        | 0.6928  | 0.2862  |
> | iTransformer | 0.5050 | 0.4061 | 0.4404 | 0.3026 | 0.2607      | 0.1903        | **0.5155**  | 0.3107  |
>
>
> To make the comparison more direct, we calculated the average performance rank for each model across all real-world datasets and compared it to their rank on our synthetic benchmark. The table2 below shows the results.
>
> ### Table2 Comparison of model performance on real datasets and synthetic datasets （Synthetic Complex Dataset Avg MSE is computed as the average of the evaluation results obtained on the Complex Real-world Pattern Simulations dataset.）
>
> | Model        | Real Dataset Avg MSE | Real Dataset Rank | Synthetic Complex Dataset Avg MSE | Synthetic Dataset Rank |
> | -------------- | ---------------------- | ------------------- | ----------------------------------- | ------------------------ |
> | Autoformer   | 0.4733               | 10                | 0.8956                            | 11                     |
> | DLinear      | 0.4059               | 8                 | 0.3379                            | 8                      |
> | PaiFilter    | 0.3628               | 3                 | 0.3004                            | 3                      |
> | PatchTST     | 0.3664               | 4                 | 0.3119                            | 6                      |
> | SegRNN       | 0.4156               | 9                 | 0.6951                            | 10                     |
> | TSMixer      | 0.6305               | 11                | 0.4903                            | 9                      |
> | TexFilter    | 0.3575               | 1                 | 0.294                             | 1                      |
> | TimeKAN      | 0.3761               | 6                 | 0.3081                            | 5                      |
> | TimeMixer    | 0.3593               | 2                 | 0.2956                            | 2                      |
> | TimesNet     | 0.4026               | 7                 | 0.3016                            | 4                      |
> | iTransformer | 0.3664               | 5                 | 0.3165                            | 7                      |
>
> The results of this hybrid analysis are highly revealing. We can find a strong correlation between the performance on our synthetic datasets and real-world data. Most notably, **the top three performing models in our synthetic benchmark—TexFilter, TimeMixer, and PaiFilter—are identically ranked as the top three performers on average across the eight real-world datasets.**
>
> This strong consistency at the top of the leaderboard provides powerful evidence for our central thesis: our synthetic benchmark is not an isolated academic exercise, but a reliable and effective tool for predicting a model's real-world performance. It successfully identifies robust, high-performing architectures, thereby demonstrating its value and significance as a more unified and dependable benchmark for future time series research.
>
> **Q2: Omitting other prominent architectures**
>
> **A2**：
> We have now included the evaluation results for CATS [1], N-BEATS [2], N-HITS [3], and TPGN [4] in Table 3.
>
> ### Table 3 Evaluation results of the supplementary model (MSE)
>
> | Dataset Type           | CATS      | N-BEATS    | N-HITS     | TPGN      |
> | ------------------------ | ----------- | ----------- | ----------- | ----------- |
> | Trend                  | 0.0451946 | 0.0000027 | 0.0012840 | 0.0000075 |
> | Period                 | 0.0283966 | 0.0117732 | 0.0105458 | 0.0417024 |
> | Complex Univariate   | 0.2701272 | 0.2188650 | 0.2991683 | 0.2570174 |
> | Complex Multivariate | 0.5155118 | 0.4356473 | 1.5722923 | 0.5230839 |
>
> Our analysis of these newly added models indicates that N-BEATS generally performs well, though it is still slightly outperformed by models like PaiFilter and TimeMixer from our initial evaluations. More detailed experimental results will be further elaborated in the main paper.
>
> [1] Kim, D., et al. "Are Self-Attentions Effective for Time Series Forecasting?" NeurIPS 2024
>
> [2] Oreshkin, B. N., et al. "N-BEATS: Neural Basis Expansion Analysis for Interpretable Time Series Forecasting." ICLR 2020.
>
> [3] Challu, C., et al. "N-HiTS: Neural Hierarchical Interpolation for Time Series Forecasting." AAAI, 2023.
>
> [4] Jia, Y., et al. "PGN: The RNN’s New Successor is Effective for Long-Range Time Series Forecasting." NeurIPS 2024.
>
>
> **Q3: The noise and anomaly injections are relatively simple.**
>
> **A3:**
> In response to your suggestion, we further extended the spectrum of irregularities in the dataset by introducing:
>
> 1. Heavy-tailed noise (t-distributed and Lévy-stable): to emulate the extreme fluctuations frequently observed in finance and physics.
> 2. Uniform and Laplace noise: to capture non-Gaussian noise characteristics typical of industrial measurements and sensor data.
> 3. Structural breaks (mean shifts and trend changes): to reflect the abrupt regime shifts and discontinuities prevalent in real-world systems such as economics, meteorology, and equipment condition monitoring.
>
> The experimental results are reported in Table 4.
>
> ### Table 4 Robustness Benchmark on Expanded Irregularity Suite – MSE Across Noise & Structural-Break Scenarios
>
> | Noise Level         | DLinear | TSMixer | iTransformer | TimeMixer | TimesNet | Autoformer | PaiFilter | SegRNN  | TexFilter | TimeKAN | PatchTST |
> | --------------------- | --------- | --------- | -------------- | ----------- | ---------- | ------------ | ----------- | --------- | ----------- | --------- | ---------- |
> | uniform-noise       | 0.02728 | 0.01849 | 0.01535      | 0.01550   | 0.01526  | 0.26592    | **0.01520**   | 0.01789 | 0.01533   | 0.01542 | 0.02016  |
> | laplace-noise       | 0.02636 | 0.01772 | 0.01567      | 0.01564   | 0.01544  | 0.17858    | **0.01543**   | 0.01820 | 0.01555   | 0.01586 | 0.01996  |
> | t-noise-df3         | 0.02622 | 0.02427 | 0.01236      | 0.01233   | 0.01220  | 0.26020    | **0.01218**   | 0.01473 | 0.01256   | 0.01222 | 0.01739  |
> | t-noise-df10        | 0.02491 | 0.01673 | 0.01415      | 0.01427   | 0.01396  | 0.27832    | **0.01390**   | 0.01681 | 0.01410   | 0.01423 | 0.01839  |
> | levy-noise-alpha1.2 | 0.02152 | 0.05975 | 0.00777      | 0.00729   | 0.00765  | 0.14915    | 0.00766   | 0.00844 | 0.00786   | **0.00747** | 0.01261  |
> | levy-noise-alpha1.8 | 0.00118 | 0.00085 | 0.00072      | 0.00076   | 0.00063  | 0.29381    | **0.00060**   | 0.00145 | 0.00062   | 0.00074 | 0.00482  |
> | mean-shift-1        | 0.03370 | 0.02301 | **0.01376**      | 0.01556   | 0.01419  | 0.18201    | 0.01410   | 0.01739 | 0.01471   | 0.01446 | 0.02101  |
> | mean-shift-3        | 0.05329 | 0.02577 | 0.01875      | 0.01410   | 0.01402  | 0.22129    | **0.01354**   | 0.01684 | 0.01399   | 0.01420 | 0.02080  |
> | trend-shift-1       | 0.02818 | 0.01930 | 0.01543      | 0.01539   | 0.01530  | 0.36088    | **0.01520**   | 0.01801 | 0.01542   | 0.01545 | 0.02009  |
> | trend-shift-2       | 0.03667 | 0.03611 | 0.03135      | 0.03147   | 0.03067  | 0.24600    | **0.03058**   | 0.03237 | 0.03080   | 0.03143 | 0.03649  |
>
> The results from Table 4, encompassing a wider range of noise types and structural breaks, further reinforce our earlier findings, with models like PaiFilter, TexFilter, and TimeKAN demonstrating consistently strong performance across these more complex irregularity scenarios.
>
> **Q4:The evaluation relies heavily on MSE and MAE, which may not fully capture model performance in all scenarios**
>
> **A4:**
> We acknowledge that MSE and MAE, while standard, may not fully capture model performance in specialized scenarios like probabilistic forecasting or extreme event prediction. Our current work primarily focuses on evaluating models' deterministic point forecasting capabilities and their ability to learn fundamental temporal patterns and robustness under controlled irregularities. For this initial scope, MSE and MAE serve as appropriate and widely recognized metrics.

---

> > ### Comment · Reviewer_xEYn · 2025-08-04
> >
> > Thank you for the response. Could you provide more detailed experimental settings for Table 3?

---

> > > ### Author Response · Authors · 2025-08-04
> > > **Detailed experimental settings for Table 3**
> > >
> > > Yes, of course. All experiments in Table 3 were conducted with a unified configuration: **an input horizon of 96 and an output horizon of 96**. The table 4 below details the specific hyperparameters used to train each model.
> > >
> > > ### Table 4 Model Hyperparameters
> > > | Model  | d\_model | d\_ff | learning\_rate | training\_epoch/max_steps  | Other Parameters                                                                    |
> > > | -------- | ---------- | ------- | ---------------- | ----------------- | ------------------------------------------------------------------------------------- |
> > > | CATS   | 256      | 512   | 0.01           | 30              | patch\_len=48, stride=48, n\_heads=32, QAM\_start=0.1, QAM\_end=0.2, d_layers=3                 |
> > > | N-BEATS | -        | -     | 0.001          | 30              | stack\_types=["seasonality","trend","identity"], n\_blocks=[3,3,3],  mlp_units=[[512,512],[512,512],[512,512]], n\_harmonics=10, n_basis=4, basis=polynomial |
> > > | N-HITS  | -        | -     | 0.001          | 200(max_steps)          | stack\_types=["identity"×3], n\_blocks=[1,1,1], pooling\_mode=MaxPool1d, mlp_units=[[512,512],[512,512],[512,512]], n_pool_kernel_size=[2,2,1]            |
> > > | TPGN   | 512      | -     | 0.01           | 30              | e\_layers=2, d\_layers=1, TPGN\_period=24, factor=3                    |
> > >
> > > Tables 5 through 8 correspond to the detailed results presented in Table 3.
> > > ### Table 5 Trend Signal （MSE）
> > > | Signal Type               | CATS     | N-BEATS   | N-HITS    | TPGN     |
> > > | ---------------------------- | ---------- | ---------- | ---------- | ---------- |
> > > | Exponential-Trend          | 4.93E-01 | 1.02E-07 | 2.41E-03 | 1.12E-05 |
> > > | Gaussian-Trend             | 5.52E-05 | 9.85E-06 | 3.10E-03 | 3.47E-06 |
> > > | Gompertz-Trend             | 5.83E-06 | 9.41E-07 | 1.55E-05 | 9.01E-06 |
> > > | Linear-Trend               | 1.94E-07 | 2.69E-10 | 3.37E-05 | 7.33E-09 |
> > > | Logarithmic-Function       | 3.38E-05 | 2.49E-07 | 7.48E-06 | 6.78E-06 |
> > > | Logistic-Trend             | 1.67E-04 | 4.84E-06 | 2.58E-04 | 6.06E-06 |
> > > | Negative-Exponential-Trend | 3.18E-05 | 3.27E-08 | 5.96E-06 | 1.17E-06 |
> > > | Piecewise-Linear           | 3.05E-04 | 5.92E-06 | 8.07E-03 | 1.21E-05 |
> > > | Power-Law-Trend            | 2.19E-04 | 1.19E-07 | 1.28E-05 | 1.64E-06 |
> > > | Quadratic-Function         | 3.51E-03 | 4.77E-06 | 2.16E-04 | 2.61E-06 |
> > > | Step-Function              | 1.56E-06 | 3.10E-06 | 9.04E-07 | 2.81E-05 |
> > >
> > > ### Table 6 Period Signal（MSE）
> > > | Signal Type  | CATS     | N-BEATS   | N-HITS    | TPGN     |
> > > | --------------- | ---------- | ---------- | ---------- | ---------- |
> > > | Double-Sin    | 1.59E-03 | 2.76E-05 | 2.56E-05 | 2.32E-04 |
> > > | Exp-Sine-Wave | 2.74E-03 | 2.36E-05 | 2.94E-05 | 2.21E-04 |
> > > | Five-Sin      | 2.24E-03 | 2.44E-04 | 1.66E-04 | 6.45E-03 |
> > > | Sawtooth-Wave | 4.39E-02 | 5.20E-08 | 6.42E-06 | 2.03E-03 |
> > > | Sin0.005      | 3.12E-02 | 1.42E-04 | 4.61E-05 | 1.80E-05 |
> > > | Sin0.05       | 2.32E-03 | 2.01E-05 | 2.74E-05 | 2.20E-05 |
> > > | Square-Wave   | 1.76E-01 | 1.16E-01 | 1.03E-01 | 3.24E-01 |
> > > | Ten-Sin       | 1.16E-02 | 2.32E-04 | 8.75E-05 | 8.33E-02 |
> > > | Triangle-Wave | 1.08E-02 | 1.11E-03 | 1.80E-03 | 3.53E-04 |
> > > | Triple-Sin    | 1.49E-03 | 5.15E-06 | 2.37E-05 | 1.92E-04 |
> > >
> > > ### Table 7 Complex Univariate（MSE）
> > > | Signal Type             | CATS     | N-BEATS   | N-HITS    | TPGN     |
> > > | -------------------------- | ---------- | ---------- | ---------- | ---------- |
> > > | Economic-Indicator       | 0.085192 | 0.06235  | 0.807331 | 0.055008 |
> > > | Electricity-Consumption  | 0.175501 | 0.12953  | 0.130937 | 0.158172 |
> > > | Industrial-Sensor        | 0.023984 | 0.019703 | 0.046449 | 0.046324 |
> > > | Industrial-Sensor-Normal | 0.072649 | 0.063022 | 0.063809 | 0.121364 |
> > > | Network-Traffic          | 0.932327 | 0.915762 | 1.062502 | 0.905582 |
> > > | Retail-Sales             | 0.564247 | 0.463018 | 0.464545 | 0.675795 |
> > > | Stock-Price              | 0.185296 | 0.180168 | 0.183314 | 0.17705  |
> > > | Temperature-Sensor       | 0.107901 | 0.067476 | 0.167201 | 0.075194 |
> > > | Website-Traffic          | 0.284047 | 0.068755 | 0.06536  | 0.098667 |
> > >
> > > ### Table 8 Complex Multivariate（MSE）
> > > | Signal Type        | CATS     | N-BEATS   | N-HITS    | TPGN     |
> > > | --------------------- | ---------- | ---------- | ---------- | ---------- |
> > > | Ad-Sales            | 1.446923 | 1.289386 | 1.264031 | 1.366184 |
> > > | Intervention-Effect | 0.144358 | 0.11422  | 0.10167  | 0.122186 |
> > > | Lotka-Volterra      | 0.245322 | 0.025021 | 0.010087 | 0.098483 |
> > > | Macro-Economy       | 0.722372 | 0.730006 | 0.74794  | 0.736329 |
> > > | SIR-Model           | 0.002999 | 0.002447 | 0.007405 | 0.00246  |
> > > | Supply-Demand-Price | 0.265598 | 0.114545 | 8.342317 | 0.805954 |
> > > | Weather-Sales       | 0.781011 | 0.773904 | 0.532596 | 0.529992 |
> > >
> > > We hope this additional information clarifies our experimental process. If there are any further questions or if you require more data, please don't hesitate to let us know.

---

> > > > ### Comment · Reviewer_xEYn · 2025-08-05
> > > >
> > > > Thank you very much for the detailed response. I will revise my score accordingly.
> > > >
> > > > Thank you again for your hard work.

---

> > > > > ### Comment · Reviewer_xEYn · 2025-08-05
> > > > >
> > > > > Please include the aforementioned experimental details explicitly in the main paper to ensure the work is easily understandable to a broader audience.

---

> > > > > > ### Author Response · Authors · 2025-08-05
> > > > > >
> > > > > > Thank you for your response and your positive feedback! We are glad that our supplementary explanations and experiments were able to address your concerns.
> > > > > >
> > > > > > We will certainly adopt your suggestion and will explicitly add all the detailed experimental settings (including hyperparameters and the specific results for each part) to the main body of the paper. This will make our work clearer and more complete.
> > > > > >
> > > > > > Thank you again for your valuable time and constructive comments.

---

### Note · Authors · 2025-08-12

Dear Reviewers, AC, SAC, and PC,

We sincerely thank the reviewers for their insightful comments and constructive guidance throughout the discussion period. Our discussions centered on several key concerns: the generalizability of our synthetic benchmark to real-world scenarios, the comprehensiveness of the evaluated models and data complexities, and the clarity of our analytical framework.

In response, we conducted extensive new experiments on eight widely-used real-world datasets, empirically demonstrating a strong correlation between model performance on our synthetic data and real-world outcomes. We also significantly expanded the benchmark's scope by incorporating four models and introducing more challenging irregularities and cross-variable evaluation. Furthermore, we provided a fine-grained analysis of each model's architectural strengths and weaknesses to enhance the clarity and impact of our findings.

We are encouraged by the reviewers' positive reception and their acknowledgment that our extensive additions have effectively addressed their primary concerns.

Finally, we wish to re-emphasize the innovative and necessary nature of this work. As noted by reviewers and the community at large, time series forecasting is hindered by an over-reliance on a few observational datasets where performance is obscured by confounding factors, and results often fail to translate to real-world scenarios. Our work introduces a new synthetic data-driven evaluation paradigm designed to overcome this fundamental challenge. Our framework provides a more comprehensive and multi-dimensional evaluation that enables a fundamental shift: from simple, holistic model rankings to a principled, diagnostic assessment. This system creates a detailed "report card" for each architecture, revealing precisely why a model succeeds or fails. Crucially, our new experiments validate this deep, diagnostic approach; in contrast to traditional scores, our benchmark offers a more fine-grained and interpretable evaluation of model capabilities, and we have shown these insights strongly correlate with real-world performance. This work, therefore, represents a necessary evolution—providing the clear, actionable insights required to guide future model improvements and foster genuine progress in the field.

Thank you once more for your valuable time.

Best regards,

Authors

---

### Decision · Program_Chairs · 2025-09-18

**Decision:**

Accept (poster)

**Comment:**

This paper proposes SynTSBench, a synthetic data-driven evaluation framework for systematically assessing time series forecasting models. The framework isolates confounding factors and evaluates models along three analytical dimensions: temporal feature decomposition and capability mapping, robustness under data irregularities, and theoretical optimum benchmarking. The authors evaluate 12 state-of-the-art models, showing that no single method consistently excels across all scenarios. The paper is well-motivated, offers a novel benchmark, and makes a meaningful contribution to the time series community. The final ratings from reviewers are 5, 5, 4, and 5, indicating broad agreement that this is a strong and impactful paper. My decision is to accept.

Reviewers highlighted several strengths, including the novelty of the synthetic benchmarking approach, the comprehensive coverage of temporal patterns, and the clear insights into model strengths and weaknesses. Multiple reviewers appreciated the detailed rebuttal, where the authors addressed concerns about generalizability by showing consistency between synthetic and real-world data, adding new experiments, and clarifying experimental design. The main limitations noted were the lack of direct validation on real-world datasets, simplified noise/anomaly models, and limited coverage of some recent architectures. Nevertheless, reviewers agreed that these issues do not overshadow the paper’s contributions. Overall, the consensus is that the work provides a valuable and interpretable benchmark that can shape future research directions in time series forecasting.